# Anomaly of $(2+1)$-dimensional symmetry-enriched topological order from $(3+1)$-dimensional topological quantum field theory

**Weicheng Ye[1,2] and Liujun Zou[1]**

**1** Perimeter Institute for Theoretical Physics, Waterloo, Ontario, Canada N2L 2Y5
**2** Department of Physics and Astronomy, University of Waterloo,
Waterloo, Ontario, Canada N2L 3G1

## Abstract

Symmetry acting on a (2+1)$D$ topological order can be anomalous in the sense that they possess an obstruction to being realized as a purely (2+1)$D$ on-site symmetry. In this paper, we develop a (3+1)$D$ topological quantum field theory to calculate the anomaly indicators of a (2+1)$D$ topological order with a general symmetry group $G$, which may be discrete or continuous, Abelian or non-Abelian, contain anti-unitary elements or not, and permute anyons or not. These anomaly indicators are partition functions of the (3+1)$D$ topological quantum field theory on a specific manifold equipped with some $G$-bundle, and they are expressed using the data characterizing the topological order and the symmetry actions. Our framework is applied to derive the anomaly indicators for various symmetry groups, including $\mathbb{Z}_2 \times \mathbb{Z}_2$, $\mathbb{Z}_2^T \times \mathbb{Z}_2^T$, $SO(N)$, $O(N)^T$, $SO(N) \times \mathbb{Z}_2^T$, etc, where $\mathbb{Z}_2$ and $\mathbb{Z}_2^T$ denote a unitary and anti-unitary order-2 group, respectively, and $O(N)^T$ denotes a symmetry group $O(N)$ such that elements in $O(N)$ with determinant $-1$ are anti-unitary. In particular, we demonstrate that some anomaly of $O(N)^T$ and $SO(N) \times \mathbb{Z}_2^T$ exhibit symmetry-enforced gaplessness, i.e., they cannot be realized by any symmetry-enriched topological order. As a byproduct, for $SO(N)$ symmetric topological orders, we derive their $SO(N)$ Hall conductance.

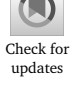

# 1 Introduction

Topological orders are interesting gapped quantum phases of matter beyond the conventional paradigm, and their discovery is one of the main forces that revolutionized modern quantum many-body physics [1]. Instead of being characterized by local order parameters associated with symmetries, in $(2+1)D$ they are characterized by *anyons*, quasiparticle excitations with nontrivial statistics that may be neither bosonic nor fermionic. The physical properties of a topological order are nicely summarized using the language of tensor category [2–5], and in particular in $(2+1)D$ bosonic systems the data of anyons forms an elegant mathematical structure called *unitary modular tensor category* (UMTC). In this paper, we focus on bosonic topological orders in $(2+1)D$, and will use the terms topological order and UMTC interchangeably.

There is rich interplay between topological order and symmetry.[1] Two topological orders that have the same set of anyon excitations but cannot be smoothly connected to each other in the presence of some symmetry are referred to as different *symmetry-enriched topological orders* (SETs). A non-trivial aspect of symmetry actions on a topological order is *symmetry fractionalization,* in the sense that symmetry actions on anyons may not form a representation of the symmetry group, but a projective representation. So we sometimes say that anyons carry "fractional" quantum numbers. Different symmetry actions on anyons, reflected in how anyons are permuted by symmetries and symmetry fractionalization patterns, differentiate different SETs.

Interestingly, some SETs are anomalous, in the sense that their symmetry fractionalization patterns cannot be realized in a purely $(2+1)D$ style with on-site symmetry actions. On the contrary, it has to be realized on the boundary of a $(3+1)D$ *symmetry-protected topological phase* (SPT), so that the symmetry actions can be on-site. This is believed to be equivalent to the notion of a 't Hooft anomaly [6]. Given a symmetry group $G$, possible anomalies are classified by *group cohomology* or *cobordism*, and these different classes are in one-to-one correspondence with the SPT states in the $(3+1)D$ bulk that can potentially cancel the anomaly and host this anomalous SET on its boundary [7–9].

Understanding the anomaly of SETs, or general quantum many-body systems, is very important because the anomaly constrains the low-energy dynamics in a powerful way. If the system has some 't Hooft anomaly, then its ground state cannot be trivial, i.e., either the symmetries are spontaneously broken, or the ground state is gapless or topologically ordered. Going one step further, even more powerful constraint comes from anomaly matching. Since the anomaly can be viewed as a property of the higher dimensional bulk, it is an invariant under deformations of the original system. In particular, it is an invariant under renormalization group that should be the same in the UV and IR. For strongly interacting field theories, we do not have too many handles on their low-energy dynamics so far, and understanding their 't Hooft anomalies and considering anomaly matching serve as a powerful approach [10–17]. More specifically, similar to topological orders, in a general strongly interacting field theory with one-form symmetries, the action of $G$ on charged line operators is specified by symmetry fractionalization and serves as a further constraint on the IR phase of these theories [13,16,17]. Therefore, understanding the anomaly of SETs can definitely shed light into the understanding of a general theory with one-form symmetries.

In the context of symmetry-enriched topological orders in condensed matter systems, it is also of paramount importance to understand the anomaly of SETs under similar veins. A fundamental task of condensed matter physics is to understand whether a certain quantum phase or phase transition can emerge from a many-body system. For this purpose, an emergibility hypothesis based on matching the Lieb-Schultz-Mattis-type anomaly of a lattice system and the

---

[1]Unless otherwise stated, all symmetries in this paper are 0-form invertible internal symmetries.

anomaly of a quantum phase or transition is proposed [11, 18]. In particular, the Lieb-Schultz-Mattis-type anomalies of a large class of lattice systems relevant to experimental and numerical studies are worked out in Ref. [18]. To apply the emergibility hypothesis to an SET, we need to know its anomaly. Furthermore, although there is great progress in understanding the characterization and classification of SETs (especially in (2+1)$D$), such understanding mostly applies to topological orders with internal symmetries, i.e., symmetries that do not change the spatial locations of the degrees of freedom. However, lattice symmetries are important in condensed matter systems, yet the characterization and classification of topological orders with lattice symmetries are relatively less understood, despite the partial progress [19–25]. In the spirit of Refs. [11, 18], understanding the anomalies of a topological order with lattice symmetries (and possibly also with internal symmetries) and applying the emergibility hypothesis provide a route to classify such symmetry-enriched topological orders in condensed matter systems.

The main goals of this paper are two folds. First, for any SET with any symmetry group $G$ (which may be discrete or continuous, Abelian or non-Abelian, contain anti-unitary elements and/or permute anyons), we develop a (3+1)$D$ topological quantum field theory (TQFT) defined on manifolds with a $G$-bundle structure, which describes the SPT state whose boundary can host this SET. Based on this TQFT, we establish a framework to calculate the anomaly of a (2+1)$D$ topological order with symmetry group $G$, by calculating the partition function of the corresponding TQFT on certain manifolds with some $G$-bundle structure. This procedure is spelled out in great detail for finite group symmetries and connected Lie groups. For disconnected Lie groups, we also have a formal construction of the partition function, although we have not rigorously proved that it satisfies all consistency conditions of a TQFT. Second, we apply this framework to specific examples. In particular, we calculate the *anomaly indicators* of various symmetry groups, including $\mathbb{Z}_2^T$, $\mathbb{Z}_2 \times \mathbb{Z}_2$, $\mathbb{Z}_2^T \times \mathbb{Z}_2^T$, $SO(N)$, $O(N)^T$ and $SO(N) \times \mathbb{Z}_2^T$, where $\mathbb{Z}_2$ and $\mathbb{Z}_2^T$ refer to a unitary and anti-unitary order-2 symmetry group, respectively, and $O(N)^T$ denotes a symmetry group $O(N)$ such that elements in $O(N)$ with determinant $-1$ are anti-unitary. Here anomaly indicators of symmetry group $G$ refer to a family of quantities, expressed in terms of the data characterizing an SET, that can completely determine the anomaly of any topological order enriched by the symmetry group $G$. In addition, a byproduct of our analysis is an explicit formula for the $SO(N)$ Hall conductance of an $SO(N)$ symmetric topological order, expressed in terms of the data characterizing this SET (up to contributions from (2+1)$D$ invertible states). Moreover, for $O(N)^T, N \geqslant 5$ and $SO(N) \times \mathbb{Z}_2^T, N \geqslant 4$, we show that certain anomalies cannot be realized by any SET, demonstrating the phenomenon of "symmetry-enforced gaplessness" [26].

In the rest of this introduction, we first comment on the relation between our work and prior work, and then give an outline and summary of the paper.

## 1.1 Relation to prior work

There are already multiple papers that discuss the anomaly of a topological order from various perspectives. See for example Refs. [27–46]. In particular, based on the idea of $G$-crossed braided fusion categories [27], Refs. [28, 29] derived a formula to calculate the anomaly of a general topological order with a unitary symmetry that does not permute anyons. Ref. [31] considered anomalies of Abelian topological orders with a finite unitary Abelian symmetry that does not permute anyons, by explicitly studying the bulk-boundary correspondence. Later, for reflection symmetry $\mathbb{Z}_2^R$ and time reversal symmetry $\mathbb{Z}_2^T$ that may permute anyons, Refs. [32–34] gave their anomaly indicators which apply to any topological order. The anomaly indicators for $U(1) \rtimes \mathbb{Z}_2^T$ and $U(1) \times \mathbb{Z}_2^T$ symmetries were later given in Ref. [35], with their lattice-symmetry-versions discussed in Ref. [36]. Ref. [36] also gave anomaly indicators for $SO(3) \times \mathbb{Z}_2^T$ and $SU(2) \times \mathbb{Z}_2^T$. Refs. [30, 37] derived a general formula to calculate the relative anomaly between two different symmetry-enriched topological orders, i.e., the difference

between the anomalies of a given topological order with different symmetry fractionalization classes. Ref. [38] gave a state-sum construction to calculate the anomaly of a general bosonic symmetry-enriched topological order with a general finite group symmetry, which may contain anti-unitary elements and/or permute anyons. This work was later generalized to fermonic symmetry-enriched topological orders [39] (see related work in Refs. [41–44]) and to incorporate a U(1) subgroup in the symmetry [40].

In this work, we calculate the anomalies and anomaly indicators via $(3+1)D$ TQFTs, in a similar spirit to Refs. [34, 38–40]. Different from Ref. [38–40], where the TQFTs are based on cellulations of 4-manifolds, we utilize handle decompositions in our construction, following the idea of Ref. [34, 47, 48]. The handle-decomposition-based formulation greatly simplifies the calculations. In this way, we explicitly derive the anomaly indicators for $\mathbb{Z}_2 \times \mathbb{Z}_2$, $\mathbb{Z}_2^T \times \mathbb{Z}_2^T$, $SO(N)$, $O(N)^T$ and $SO(N) \times \mathbb{Z}_2^T$ symmetries (besides reproducing the known anomaly indicators in the literature [32–36]). Our framework has wide applicability, and now the calculation of anomaly indicators for any symmetry group $G$ is equally straightforward.

There is also a vast number of works done regarding constructing a $(3+1)D$ TQFT from the data of a UMTC (or tensor category in general), including Refs. [34, 38, 47–57]. Our work builds on the construction in Refs. [34, 48] to build up our TQFT, and in particular we spell out in detail how to deal with manifolds with a $G$-bundle structure and categories with a $G$-action, for general symmetry group $G$. When $G$ is finite, our work can also be thought of as a handle version of the state sum construction in Ref. [38]. As mentioned before, our formulation makes the calculation much easier and explicit formulae possible. Moreover, our framework generalizes in a straightforward manner to continuous symmetries.

We remark that symmetries considered in this paper are all "exact symmetries", which are supposed to be present in the system microscopically. They are in contrast to "emergent symmetries", which do not exist microscopically but emerge as good approximate symmetries at low energies and long distances, sometimes in the form of generalized symmetries [58–60]. A possible approach to calculate the anomaly associated with an exact symmetry is to first figure out the full emergent symmetry of a theory and its associated anomaly, and then use some "pullback" to get the anomaly of the exact symmetry (see, e.g., Refs. [17, 45, 46]). This approach is certainly elegant. However, as more and more emergent generalized symmetries are discovered, it appears subtle to know whether we obtain the complete set of emergent symmetries and how exactly the anomaly of the exact symmetry is related to the anomaly of the emergent symmetry (see Point 7 of Discussion in Sec. 7). Specifically, one might wonder, within such an approach, if one has to first understand all emergent non-invertible symmetries and their anomalies, which seems complicated. In this paper, we avoid this subtlety by directly working with the exact symmetry of a topological order, without referring to its full emergent symmetry. In particular, the construction of the $(3+1)D$ TQFT does not explicitly take the full emergent symmetry as an input.

## 1.2 Outline and summary

The outline and summary of the rest of the paper are as follows.

- In Sec. 2, we briefly review relevant concepts and notations of UMTC and symmetry fractionalization.

- In Sec. 3, for a finite symmetry group $G$, we present the general construction of the $(3+1)D$ TQFT defined on 4-manifolds equipped with an extra $G$-bundle structure (see Sec. 3.2) and an explicit recipe to calculate its partition function (see Sec. 3.4). This partition function is expressed compactly in Eq. (44).

- In Sec. 4, we apply the general framework to calculate the anomaly indicators of various finite group symmetries. First, we reproduce the anomaly indicators of the $\mathbb{Z}_2^T$ symmetry (see Eqs. (46),(50)), first proposed in Ref. [32] and later proved in Ref. [34] (see also Ref. [33]). We then derive the anomaly indicators of the $\mathbb{Z}_2 \times \mathbb{Z}_2$ (see Eqs. (53),(54)) and $\mathbb{Z}_2^T \times \mathbb{Z}_2^T$ symmetries (see Eqs. (55),(56)), which are unavailable in the prior literature as far as we know. To illustrate the usage of these anomaly indicators, in Sec. 4.4.1 we classify all symmetry fractionalization classes of the all-fermion $\mathbb{Z}_2$ topological order with $\mathbb{Z}_2^T \times \mathbb{Z}_2^T$ symmetry, and calculate the anomalies for all these classes.

- In Sec. 5, we generalize the construction to connected Lie group symmetries, where the expression of the partition function is given by Eq. (65). We then apply it to derive the anomaly indicators of $SO(N)$ (see Eqs. (78)). As a byproduct, we also derive the $SO(N)$ Hall conductance of an $SO(N)$ symmetric topological order (up to contributions from $(2+1)D$ invertible states), expressed in terms of the data characterzing an SET (see Eqs. (80),(81)).

- In Sec. 6, we explain a simple way to use the results we have already derived to obtain the anomaly indicators of many other groups, including $O(N)^T$, $SO(N) \times \mathbb{Z}_2^T$, $\mathbb{Z}_n \times \mathbb{Z}_2^T$, $\mathbb{Z}_n \rtimes \mathbb{Z}_2^T$, $\mathbb{Z}_n \rtimes \mathbb{Z}_2$, $O(N)$, etc. In particular, we derive the anomaly indicators of $O(N)^T$ (see Eqs. (95),(96)) and $SO(N) \times \mathbb{Z}_2^T$ (see Eq. (106)), and demonstrate that certain anomaly of them cannot be realized by any symmetry-enriched topological order, showcasing the phenomenon of "symmetry-enforced gaplessness" [26,61,62].

- We finish with some discussion in Sec. 7.

- The appendices contain further details of our framework and calculations. Appendix A presents the derivation that leads to our main formulae Eq. (44). In Appendix B, for finite group symmetry $G$, we give a more explicit expression of the "$\eta$-factor" that will enter the partition function in Eq. (44). In Appendix C, we explicitly perform various consistency checks for the partition functions, given by Eq. (44) for a finite group symmetry $G$ and Eq. (65) for a connected Lie group symmetry $G$. In Appendix D, we give some introduction about identifying manifolds relevant to calculating the anomaly indicators. In Appendix E, we present more details on the handle decomposition of various manifolds explicitly used in the paper.

## 2 Review of topological order with symmetry $G$

### 2.1 Review of UMTC notation

In this subsection we briefly review relevant concepts and notations that we use to describe UMTCs. For a more comprehensive review of these concepts and notations, see e.g., Refs. [29, 63,64] for a more physics oriented introduction, or Refs. [2,3,65,66] for a more mathematical treatment.

   A category consists of objects and morphisms between those objects. In a UMTC $\mathcal{C}$, there is a finite set of simple objects $a$. They are referred to as (simple) anyons in the context of topological orders. The set of morphisms $\mathrm{Hom}(a, b)$ between two objects $a$ and $b$ in a UMTC $\mathcal{C}$ forms a $\mathbb{C}$-linear vector space. The vector space is referred to as the topological state space in the context of topological order. For example, $\mathrm{Hom}(a, b)$ can be viewed as the Hilbert space of states on a 2-sphere that hosts anyons $a$ and $\bar{b}$ (see Eq. (35)).

Moreover, a UMTC $\mathcal{C}$ has the structure of fusion and braiding. Fusion means that there is a bifunctor $\times$ such that acting it on anyons $a$ and $b$ we have

$$a \times b \cong \sum_c N_{ab}^c c, \tag{1}$$

where $N_{ab}^c$ is interpreted as the dimension of the topological state space of two anyons $a$ and $b$ fusing into a third anyon $c$. There are two related vector spaces, $V_{ab}^c$ and $V_c^{ab}$, referred to as the fusion and splitting vector spaces, respectively. The two vector spaces are dual to each other, and depicted graphically as:

$$(d_c/d_a d_b)^{1/4} \quad \vcenter{\hbox{}} = \langle a, b; c|_\mu \in V_{ab}^c, \tag{2}$$

$$(d_c/d_a d_b)^{1/4} \quad \vcenter{\hbox{}} = |a, b; c\rangle_\mu \in V_c^{ab}, \tag{3}$$

where $\mu = 1, \ldots, N_{ab}^c$, $d_a$ is the *quantum dimension* of $a$, and the factors $\left(\frac{d_c}{d_a d_b}\right)^{1/4}$ are a normalization convention for the diagrams.

In this paper, we will use the convention that the splitting space is referred to as the vector space, corresponding to "ket" in Dirac's notation, while the fusion space is the dual vector space, corresponding to "bra" in Dirac's notation. Diagrammatically, inner products of the vector space are formed by stacking vertices so the fusing/splitting lines connect

$$\vcenter{\hbox{}} = \delta_{cc'} \delta_{\mu\mu'} \sqrt{\frac{d_a d_b}{d_c}} \quad \vcenter{\hbox{}}, \tag{4}$$

which can be applied inside more complicated diagrams.

More generally, for any integer $n$ and $m$ there are vector spaces $V_{b_1, b_2, \ldots, b_m}^{a_1, a_2, \ldots, a_n}$, which are referred to as the fusion space of $m$ anyons into $n$ anyons. These vector spaces have a natural basis in terms of tensor products of the elementary splitting spaces $V_c^{ab}$ and fusion spaces $V_{ab}^c$. For instance, we have

$$V_d^{abc} \cong \sum_e V_e^{ab} \otimes V_d^{ec} \cong \sum_f V_d^{af} \otimes V_f^{bc}. \tag{5}$$

The two vector spaces are related to each other by a basis transformation referred to as $F$-symbols, which is diagrammatically shown as follows

$$\vcenter{\hbox{}} = \sum_{f, \mu, \nu} \left[F_d^{abc}\right]_{(e,\alpha,\beta),(f,\mu,\nu)} \vcenter{\hbox{}}. \tag{6}$$

The basis transformations are required to be unitary transformations, i.e.

$$\left[\left(F_d^{abc}\right)^{-1}\right]_{(f,\mu,\nu)(e,\alpha,\beta)} = \left[\left(F_d^{abc}\right)^\dagger\right]_{(f,\mu,\nu)(e,\alpha,\beta)}$$
$$= \left[F_d^{abc}\right]_{(e,\alpha,\beta)(f,\mu,\nu)}^*. \tag{7}$$

There is also a trivial anyon denoted by 1 such that $1 \times a = a \times 1 = a$. We denote $\bar{a}$ as the anyon conjugate to $a$, for which $N_{a\bar{a}}^1 = 1$, i.e.

$$a \times \bar{a} = 1 + \cdots. \tag{8}$$

Note that $\bar{a}$ is unique for a given $a$.

The *R*-symbols define the braiding properties of the anyons, and are defined via the the following diagram:

$$a \searrow\hspace{-0.3em}\nearrow b \atop c \uparrow_{\mu} = \sum_{\nu} \left[R_c^{ab}\right]_{\mu\nu} \, a \searrow\hspace{-0.3em}\nearrow b \atop c \uparrow_{\nu} . \tag{9}$$

Under a basis transformation, $\Gamma_c^{ab} : V_c^{ab} \to V_c^{ab}$, the $F$ and $R$ symbols change according to:

$$F_{def}^{abc} \to \tilde{F}_d^{abc} = \Gamma_e^{ab} \Gamma_d^{ec} F_{def}^{abc} [\Gamma_f^{bc}]^\dagger [\Gamma_d^{af}]^\dagger ,$$
$$R_c^{ab} \to \tilde{R}_c^{ab} = \Gamma_c^{ba} R_c^{ab} [\Gamma_c^{ab}]^\dagger , \tag{10}$$

where we have suppressed splitting space indices and dropped brackets on the *F*-symbol for shorthand. In this paper, we refer to this basis transformation as a *vertex basis transformation*.

On the other hand, physical quantities, like the topological twist $\theta_a$ and the modular *S*-matrix $S_{ab}$, should always be basis-independent combinations of the data. The *topological twist* $\theta_a$ is defined via the diagram:

$$\theta_a = \theta_{\bar{a}} = \sum_{c,\mu} \frac{d_c}{d_a} \left[R_c^{aa}\right]_{\mu\mu} = \frac{1}{d_a} \; \infty_a . \tag{11}$$

Finally, the *modular S-matrix $S_{ab}$*, is defined as

$$S_{ab} = D^{-1} \sum_c N_{\bar{a}b}^c \frac{\theta_c}{\theta_a \theta_b} d_c = \frac{1}{D} \, a\hspace{-0.2em}\bigcirc\hspace{-0.4em}b\hspace{-0.2em}\bigcirc , \tag{12}$$

where $D = \sqrt{\sum_a d_a^2}$ is the *total dimension* of the UMTC.

## 2.2 Global symmetry

We now consider a UMTC $\mathcal{C}$ which is equipped with a global symmetry group $G$. Mathematically speaking, by definition, $G$ associates a monoidal functor $\rho_{\mathbf{g}}$ modulo natural isomorphism to each $\mathbf{g} \in G$, which should satisfy various consistency conditions. In this subsection we break down the definition and review the concepts and notations related to global symmetry $G$. For a more comprehensive review, see e.g., Refs. [27, 29, 65].

First of all, as a functor, $\rho_{\mathbf{g}}$ acts on the anyon labels and the topological state spaces. For an individual element $\mathbf{g} \in G$, $\mathbf{g}$ can permute the anyons and we use ${}^{\mathbf{g}}a$ to denote the (simple) anyon we get after the $\mathbf{g}$ action on the (simple) anyon labeled by $a$. Moreover, $\mathbf{g}$ also has an action on the topological state space, which is a $\mathbb{C}$-linear or $\mathbb{C}$-anti-linear operator on the fusion space, depending on whether $\mathbf{g}$ is unitary or anti-unitary. We denote this action on individual topological state space as $\rho_{\mathbf{g}}$ as well:

$$\rho_{\mathbf{g}} : \; V_c^{ab} \to V_{{}^{\mathbf{g}}c}^{{}^{\mathbf{g}}a \, {}^{\mathbf{g}}b} . \tag{13}$$

And in particular we have

$$N_{{}^{\mathbf{g}}a \, {}^{\mathbf{g}}b}^{{}^{\mathbf{g}}c} = N_{ab}^c . \tag{14}$$

To account for anti-unitary symmetry, we associate a $\mathbb{Z}_2$ grading $q(\mathbf{g})$ (and related $\sigma(\mathbf{g})$) as follows

$$q(\mathbf{g}) = \begin{cases} 0, & \text{if } \mathbf{g} \text{ is unitary}, \\ 1, & \text{if } \mathbf{g} \text{ is anti-unitary}, \end{cases} \tag{15}$$

$$\sigma(\mathbf{g}) = \begin{cases} 1, & \text{if } \mathbf{g} \text{ is unitary}, \\ *, & \text{if } \mathbf{g} \text{ is anti-unitary}, \end{cases} \tag{16}$$

where $*$ denotes complex conjugation.

Assembling the above information in the component form, we can write the action of $\rho_{\mathbf{g}}$ on the topological state space as a matrix $U_{\mathbf{g}}(^{\mathbf{g}}a, {}^{\mathbf{g}}b; {}^{\mathbf{g}}c)_{\mu\nu}$

$$\rho_{\mathbf{g}}|a, b; c\rangle_{\mu} = \sum_{\nu} U_{\mathbf{g}}(^{\mathbf{g}}a, {}^{\mathbf{g}}b; {}^{\mathbf{g}}c)_{\mu\nu} K^{q(\mathbf{g})} |{}^{\mathbf{g}}a, {}^{\mathbf{g}}b; {}^{\mathbf{g}}c\rangle_{\nu}, \tag{17}$$

where $U_{\mathbf{g}}(^{\mathbf{g}}a, {}^{\mathbf{g}}b; {}^{\mathbf{g}}c)$ is an $N_{ab}^c \times N_{ab}^c$ matrix, and $K$ denotes complex conjugation which appears when $q(\mathbf{g}) = 1$ and the action $\rho_{\mathbf{g}}$ is $\mathbb{C}$-anti-linear. As a convention, we will also use $U_{\mathbf{g}}^{-1}(^{\mathbf{g}}a, {}^{\mathbf{g}}b; {}^{\mathbf{g}}c)$ to denote the matrix inverse of $U_{\mathbf{g}}(^{\mathbf{g}}a, {}^{\mathbf{g}}b; {}^{\mathbf{g}}c)$, even when $\mathbf{g}$ is anti-unitary.

Under a vertex basis transformation, $\Gamma_c^{ab}: V_c^{ab} \to V_c^{ab}$, $U_{\mathbf{g}}(a, b; c)_{\mu\nu}$ transforms to

$$\tilde{U}_{\mathbf{g}}(a, b, c) = \left[ \Gamma_{\bar{\mathbf{g}}c}^{\bar{\mathbf{g}}a \bar{\mathbf{g}}b} \right]^{\sigma(\mathbf{g})} U_{\mathbf{g}}(a, b, c) \left[ (\Gamma_c^{ab})^{-1} \right], \tag{18}$$

with the shorthand $\bar{\mathbf{g}} = \mathbf{g}^{-1}$. Moreover, to preserve the structure of braiding and fusion, under the action of $\rho_{\mathbf{g}}$, the $F$ and $R$ symbols should transform according to the following rules:

$$\rho_{\mathbf{g}}[F_{def}^{abc}] = U_{\mathbf{g}}(^{\mathbf{g}}a, {}^{\mathbf{g}}b; {}^{\mathbf{g}}e) U_{\mathbf{g}}(^{\mathbf{g}}e, {}^{\mathbf{g}}c; {}^{\mathbf{g}}d) F_{{}^{\mathbf{g}}d\,{}^{\mathbf{g}}e\,{}^{\mathbf{g}}f}^{{}^{\mathbf{g}}a\,{}^{\mathbf{g}}b\,{}^{\mathbf{g}}c} U_{\mathbf{g}}^{-1}(^{\mathbf{g}}b, {}^{\mathbf{g}}c; {}^{\mathbf{g}}f) U_{\mathbf{g}}^{-1}(^{\mathbf{g}}a, {}^{\mathbf{g}}f; {}^{\mathbf{g}}d)$$

$$= K^{q(\mathbf{g})} F_{def}^{abc} K^{q(\mathbf{g})},$$

$$\rho_{\mathbf{g}}[R_c^{ab}] = U_{\mathbf{g}}(^{\mathbf{g}}b, {}^{\mathbf{g}}a; {}^{\mathbf{g}}c) R_{{}^{\mathbf{g}}c}^{{}^{\mathbf{g}}a\,{}^{\mathbf{g}}b} U_{\mathbf{g}}(^{\mathbf{g}}a, {}^{\mathbf{g}}b; {}^{\mathbf{g}}c)^{-1} = K^{q(\mathbf{g})} R_c^{ab} K^{q(\mathbf{g})}, \tag{19}$$

where we have suppressed the additional indices that appear when $N_{ab}^c > 1$. Accordingly, the basis-independent quantity, including the topological twist $\theta_a$ and the modular $S$-matrix $S_{ab}$, should be invariant or complex-conjugated under the action of $\rho_{\mathbf{g}}$, i.e.,

$$S_{{}^{\mathbf{g}}a\,{}^{\mathbf{g}}b} = S_{ab}^{\sigma(\mathbf{g})},$$

$$\theta_{{}^{\mathbf{g}}a} = \theta_a^{\sigma(\mathbf{g})}. \tag{20}$$

Finally, we demand that $\rho_{\mathbf{g}}$ satisfy the group multiplication rule up to a natural isomorphism denoted by $\eta(\mathbf{g}, \mathbf{h})$, i.e.,

$$\eta(\mathbf{g}, \mathbf{h}): \quad \rho_{\mathbf{g}} \circ \rho_{\mathbf{h}} \Longrightarrow \rho_{\mathbf{gh}}. \tag{21}$$

By the definition of natural isomorphism, first of all, for every anyon $a$, $\eta(\mathbf{g}, \mathbf{h})$ assigns a morphism $\eta_{{}^{\mathbf{gh}}a}(\mathbf{g}, \mathbf{h}) \in \text{Hom}(^{\mathbf{g}}(^{\mathbf{h}}a), {}^{\mathbf{gh}}a)$ to $^{\mathbf{gh}}a$. In order for this morphism to be an isomorphism, we need to have

$$^{\mathbf{g}}(^{\mathbf{h}}a) = {}^{\mathbf{gh}}a, \tag{22}$$

and accordingly, $\eta_{{}^{\mathbf{gh}}a}(\mathbf{g}, \mathbf{h})$ can be identified with just a $U(1)$ phase for simple anyon $a$. Secondly, the definition of natural isomorphism demands that, on the topological state space $|{}^{\overline{\mathbf{gh}}}a, {}^{\overline{\mathbf{gh}}}b; {}^{\overline{\mathbf{gh}}}c\rangle_{\mu}$, the action of $\rho_{\mathbf{g}} \circ \rho_{\mathbf{h}}$ should be equal to the action of $\rho_{\mathbf{gh}}$ up to a phase $\frac{\eta_a(\mathbf{g}, \mathbf{h}) \eta_b(\mathbf{g}, \mathbf{h})}{\eta_c(\mathbf{g}, \mathbf{h})}$, i.e., we should have

$$\frac{\eta_a(\mathbf{g}, \mathbf{h}) \eta_b(\mathbf{g}, \mathbf{h})}{\eta_c(\mathbf{g}, \mathbf{h})} = U_{\mathbf{g}}(a, b; c)^{-1} K^{q(\mathbf{g})} U_{\mathbf{h}}(\bar{\mathbf{g}}a, \bar{\mathbf{g}}b; \bar{\mathbf{g}}c)^{-1} K^{q(\mathbf{g})} U_{\mathbf{gh}}(a, b; c). \tag{23}$$

This phase is often denoted by $\kappa_{\mathbf{g}, \mathbf{h}}(a, b; c)$ in the literature [29].

We also wish to impose a third constraint on $\eta(\mathbf{g}, \mathbf{h})$ coming from the constraint of associativity of symmetry actions. Namely, we wish that the two different ways of connecting $\rho_{\mathbf{g}} \circ \rho_{\mathbf{h}} \circ \rho_{\mathbf{k}}$ with $\rho_{\mathbf{ghk}}$ through natural isomorphism $\eta$ are identically the same, i.e., we wish to have

$$\eta_a(\mathbf{g}, \mathbf{h})\eta_a(\mathbf{gh}, \mathbf{k}) = \eta_a(\mathbf{g}, \mathbf{hk})\eta_{\bar{\mathbf{g}}_a}(\mathbf{h}, \mathbf{k})^{\sigma(\mathbf{g})}. \tag{24}$$

The action $\rho_{\mathbf{g}}$ above defines an element $\mathfrak{O} \in \mathcal{H}^3_{[\rho]}(G, \mathcal{A})$ [27, 29, 45]. Eq. (24) can be satisfied only when $\mathfrak{O}$ is trivial. If $\mathfrak{O}$ is non-trivial, then $\mathfrak{O}$ is referred to as the *obstruction to symmetry fractionalization*.[2] Different solutions $\eta_a(\mathbf{g}, \mathbf{h})$ of Eq. (23) together with (24) corresponding to the same $\rho_{\mathbf{g}}$ are referred to as different *symmetry fractionalization classes*.

Finally, we identify different choices of $\rho_{\mathbf{g}}$ up to natural isomorphism $\gamma(\mathbf{g})$, i.e., we identify two sets of functors $\rho_{\mathbf{g}}$ and $\tilde{\rho}_{\mathbf{g}}$ if they are connected to each other by some natural isomorphism $\gamma(\mathbf{g})$

$$\gamma(\mathbf{g}): \quad \rho_{\mathbf{g}} \Longrightarrow \tilde{\rho}_{\mathbf{g}}, \tag{25}$$

and this changes $U_{\mathbf{g}}(a, b; c)$ and $\eta_a(\mathbf{g}, \mathbf{h})$ in the following way [29]:

$$U_{\mathbf{g}}(a, b; c) \to \frac{\gamma_a(\mathbf{g})\gamma_b(\mathbf{g})}{\gamma_c(\mathbf{g})} U_{\mathbf{g}}(a, b; c),$$

$$\eta_a(\mathbf{g}, \mathbf{h}) \to \frac{\gamma_a(\mathbf{gh})}{\gamma_a(\mathbf{g})(\gamma_{\bar{\mathbf{g}}_a}(\mathbf{h}))^{\sigma(\mathbf{g})}} \eta_a(\mathbf{g}, \mathbf{h}). \tag{26}$$

In this paper we refer to this transformation as the *symmetry action gauge transformation*. Different gauge inequivalent choices of $\{\eta\}$ and $\{U\}$ characterize distinct symmetry fractionalization classes [29]. In this paper we will always fix the gauge

$$\eta_1(\mathbf{g}, \mathbf{h}) = \eta_a(\mathbf{1}, \mathbf{g}) = \eta_a(\mathbf{g}, \mathbf{1}) = 1,$$
$$U_{\mathbf{g}}(1, b; c) = U_{\mathbf{g}}(a, 1; c) = 1. \tag{27}$$

Moreover, we choose $\rho_1$ to always be the identity functor. When $G$ is continuous, we further choose $\rho_{\mathbf{g}}$ such that $\rho_{\mathbf{g}}$'s for different $\mathbf{g}$'s in the same connected component are the same functor.

One can show that distinct symmetry fractionalization classes form a torsor over $\mathcal{H}^2_\rho(G, \mathcal{A})$. That is, different possible symmetry fractionalization classes can be related to each other by elements of $\mathcal{H}^2_\rho(G, \mathcal{A})$, where $\mathcal{A}$ is an Abelian group whose group elements correspond to the Abelian anyons in this UMTC, and the group multiplication corresponds to the fusion of these Abelian anyons. In particular, given an element $[\mathbf{t}] \in \mathcal{H}^2_\rho(G, \mathcal{A})$, we can go from one symmetry fractionalization class with data $\eta_a(\mathbf{g}, \mathbf{h})$ to another with data $\tilde{\eta}_a(\mathbf{g}, \mathbf{h})$ given by

$$\tilde{\eta}_a(\mathbf{g}, \mathbf{h}) = \eta_a(\mathbf{g}, \mathbf{h})M_{a, \mathbf{t}(\mathbf{g}, \mathbf{h})}, \tag{28}$$

where $\mathbf{t}(\mathbf{g}, \mathbf{h}) \in \mathcal{A}$ is a representative 2-cocyle for the cohomology class $[\mathbf{t}]$ and $M_{a, \mathbf{t}(\mathbf{g}, \mathbf{h})} = \frac{\theta_{a \times \mathbf{t}(\mathbf{g}, \mathbf{h})}}{\theta_a \theta_{\mathbf{t}(\mathbf{g}, \mathbf{h})}}$ is the double braid between $a$ and $\mathbf{t}(\mathbf{g}, \mathbf{h})$ [67].

In the case where the permuation $\rho$ is trivial, there is always a canonical notion of a trivial symmetry fractionalization class, where $\eta_a(\mathbf{g}, \mathbf{h}) = 1$ for all anyon $a$ and all $\mathbf{g}, \mathbf{h} \in G$. In this case, an element of $\mathcal{H}^2(G, \mathcal{A})$ is sufficient to completely characterize the symmetry fractionalization class.

---

[2]In this paper, we will always assume that this obstruction is absent, and it can be straightforwardly checked for specific examples that we consider in the paper.

As the take-home message, the data $\{\rho_{\mathbf{g}}; U_{\mathbf{g}}(a, b; c), \eta_a(\mathbf{g}, \mathbf{h})\}$ defines a categorical $G$ action on $\mathcal{C}$, satisfying various consistency conditions, especially Eqs. (19),(23) and (24).

Sometimes we need to consider the symmetry actions of two different groups $G_1$ and $G_2$ on a UMTC $\mathcal{C}$, with data $\{\rho_{\mathbf{g}}^{(1)}; U_{\mathbf{g}}^{(1)}(a, b; c), \eta_a^{(1)}(\mathbf{g}, \mathbf{h})\}$ and $\{\rho_{\mathbf{g}}^{(2)}; U_{\mathbf{g}}^{(2)}(a, b; c), \eta_a^{(2)}(\mathbf{g}, \mathbf{h})\}$, respectively. We say that a map $f : G_1 \to G_2$ is compatible with these symmetry actions on $\mathcal{C}$ if for any $\mathbf{g}_1 \in G_1$, $\rho_{\mathbf{g}_1}^{(1)}$ and $\rho_{f(\mathbf{g}_1)}^{(2)}$ are two functors connected to each other by a natural isomorphism $\gamma(\mathbf{g}_1)$ as in Eq. (25), i.e.,

$$\gamma(\mathbf{g}_1): \quad \rho_{\mathbf{g}_1}^{(1)} \Longrightarrow \rho_{f(\mathbf{g}_1)}^{(2)}. \tag{29}$$

In particular, $\mathbf{g}_1$ and $f(\mathbf{g}_1)$ are either both unitary or both anti-unitary, and they permute anyons in exactly the same way. Moreover, up to a symmetry action gauge transformation their actions on the topological state space satisfy

$$U_{f(\mathbf{g}_1)}^{(2)}(a, b; c) = U_{\mathbf{g}_1}^{(1)}(a, b; c), \tag{30}$$

for any anyons $a, b, c \in \mathcal{C}$. All maps between symmetries considered in this paper are in fact maps compatible with symmetry actions on some UMTC $\mathcal{C}$ if not stated explicitly.

Given such a map, we say that the symmetry fractionalization class $\eta^{(1)}$ of $G_1$ is the *pullback* of the symmetry fractionalization class $\eta^{(2)}$ of $G_2$, if, under the gauge choice leading to Eq. (30), we have

$$\eta_a^{(1)}(\mathbf{g}, \mathbf{h}) = \eta_a^{(2)}(f(\mathbf{g}), f(\mathbf{h})), \tag{31}$$

for any $\mathbf{g}, \mathbf{h} \in G_1$ and any $a \in \mathcal{C}$. It is straightforward to see that $\eta_a^{(1)}(\mathbf{g}, \mathbf{h})$ defined this way satisfies Eqs. (23) and (24), as long as $\eta_a^{(2)}(\mathbf{g}, \mathbf{h})$ does.

# 3 (3+1)$D$ TQFT with finite group symmetry $G$

A UMTC $\mathcal{C}$ defines a (3+1)$D$ TQFT via a path integral state sum construction due originally to Crane and Yetter [49], and the state sum construction is extended to orientable or nonorientable manifolds with $G$-bundle structure in Ref. [38], where $G$ is a finite group. In this section, after explaining the relation of the partition function to anomaly, we review the approach of Refs. [34, 47, 48] to give a more formal definition of the TQFT along the lines of Refs. [68, 69], and demonstrate how to compute the partition function of this TQFT. In particular, we also extend the approach to allow for an extra $G$-bundle structure, where $G$ is a finite group symmetry that may contain anti-unitary elements, in which case the manifold under consideration can be non-orientable. While Ref. [38] explictly uses a cellulation of a manifold, our approach here utilizes a handle decomposition of a manifold, which is reviewed in Sec. 3.3. As a result, our calculation is simpler and will produce closed-form expressions for partition functions and anomaly indicators.

In this section, when we refer to a manifold $\mathcal{M}$, we assume that there is a $G$-bundle structure $\mathcal{G}$ defined on it as well, and an orientation has been chosen if $\mathcal{M}$ is orientable.[3]

## 3.1 Characterizing the anomaly by bulk-boundary correspondence

In the field theoretic language, a $(d + 1)D$ $G$-symmetric theory is anomalous if it cannot be gauged, i.e., its partition function evaluated on a $(d+1)D$ manifold with a $G$-bundle cannot be

---

[3]Even for non-orientable $\mathcal{M}$, we still need to choose an orientation of $T\mathcal{M} \oplus \xi$, where $T\mathcal{M}$ is the tangent bundle of $\mathcal{M}$ and $\xi$ denotes the 1-dimensional vector bundle associated to the gauge bundle $\mathcal{G}$ according to $q : G \to \mathbb{Z}_2$ [70,71]. See Appendix D.

made gauge invariant by local deformations. However, there exists an appropriate $(d+1+1)D$ $G$-symmetric invertible bulk theory [71, 72] whose boundary can host the original $(d + 1)D$ theory, such that the combined theory is anomaly-free. So we can characterize the anomaly of the boundary utilizing properties of the bulk. Specifically, the topological part of the partition function of the $(d + 1)D$ theory (i.e., the part of the partition function that is insensitive to dynamical details and only concerns the anomaly) on some $(d+1)D$ manifold $\mathcal{N}$ can be defined as the partition function of a $(d+1+1)D$ invertible bulk theory on some $(d+1+1)D$ manifold $\mathcal{M}$ with $\partial \mathcal{M} = \mathcal{N}$, i.e.,

$$\mathcal{Z}_{d+1}(\mathcal{N}) \equiv \mathcal{Z}_{d+1+1}(\mathcal{M}; \partial \mathcal{M} = \mathcal{N}). \tag{32}$$

Yet as an intrinsic $(d + 1)D$ theory the partition function for fixed $\mathcal{N}$ should be independent of the choice of $\mathcal{M}$. Hence on closed $(d + 1 + 1)D$ manifold $\mathcal{M}$ we are supposed to have $\mathcal{Z}_{d+1+1}(\mathcal{M}; \partial \mathcal{M} = \emptyset) = 1$. Therefore, any $\mathcal{Z}_{d+1+1}(\mathcal{M}; \partial \mathcal{M} = \emptyset) \neq 1$ suggests that the boundary theory on $\mathcal{N}$ is anomalous, and the class of anomaly is encoded in the bulk partition function, which should be a gauge invariant U(1) phase factor. Below we will use this bulk partition function to characterize the boundary anomaly.[4]

The case that concerns us is a $(2+1)D$ symmetry-enriched topological order described by a UMTC $\mathcal{C}$ and a global symmetry $G$. In the case where $G$ is trivial, the UMTC indeed defines a $(3+1)D$ invertible TQFT called the *Crane-Yetter model* [49]. However, the physical system that the Crane-Yetter model defines is trivial in the sense that the partition function on any close 4-manifold can be tuned to 1 without closing the gap or breaking any symmetry (in fact no symmetry is imposed at all in this model). Mathematically, the partition function corresponds to some element that belongs to $\mathrm{Hom}(\Omega_4^{SO}(\star), U(1)) \cong U(1)$, and all these elements are smoothly connected to the trivial element. This means that there is no intrinsic topological order in the bulk defined by the UMTC $\mathcal{C}$ in this way [4, 50, 52, 73]. Nevertheless, the $(3+1)D$ theory on a manifold with boundary hosts a $(2+1)D$ topological state at its boundary, whose anyon excitations are described by the UMTC $\mathcal{C}$ [53]. Moreover, the partition function of the Crane-Yetter model is related to the *framing anomaly* of the $(2 + 1)D$ topological state, as discussed in Refs. [34, 74].

In the presence of symmetries, the $(3+1)D$ bulk is generically an SPT state. The partition function of this SPT state corresponds to some element of the cobordism group $\Omega_{SO}^4((BG)^{q-1})$, with $q : G \to \mathbb{Z}_2$ as in Eq. (15) labeling anti-unitary symmetries (see Appendix D for the precise definition).[5] Therefore, in order to understand the SPT, we just need to calculate the partition function on a few *representative manifolds*, given by the generators of the dual bordism group $\Omega_4^{SO}((BG)^{q-1})$. A complete set of such partition functions, expressed in terms of the data characterizing $(2+1)D$ symmetry-enriched topological orders, are the *anomaly indicators*. The values of these anomaly indicators for a given symmetry-enriched topological order characterize its anomaly, corresponding to an element in the relevant cohomology or cobordism group.[6] Namely, there is an injection that maps the possible values of the anomaly indicators to elements of the relevant cohomology or cobordism group.[7]

---

[4]In the literature, we usually say that there exists a bulk $G$-SPT that can "cancel" the anomaly on the boundary, such that the total partition function of the combined bulk and boundary system is gauge invariant. According to our convention, the partition function of such bulk $G$-SPT should be the inverse of $\mathcal{Z}_{d+1+1}(\mathcal{M}; \partial \mathcal{M} = \emptyset)$ that we present here.

[5]To ease the notation, we will omit the superscript $q - 1$ when $G$ contains unitary symmetries only.

[6]More precisely, after choosing a basis of the cohomology or cobordism group, the anomaly indicators are the expansion coefficients of the element under this basis.

[7]For a finite group $G$, because any $(3+1)D$ SPT can have symmetric topologically ordered boundary, this injection should be a bijection [75, 76]. However, for a continuous group $G$, because sometimes the $(3+1)D$ SPT cannot have any symmetric topologically ordered boundary [26, 61, 62], this injection is generically not surjective.

## 3.2 General construction of TQFT

In this subsection we review the basic facts about TQFT that concern us in the context of topological order, which ultimately lead to our recipe for calculating the partition function in Sec. 3.4. The presentation here loosely follows Refs. [34, 69, 77]. See also Ref. [48]. This subsection is rather formal, and readers uninterested in the origin of various rules of the calculations can skip this subsection and take the recipe in Sec. 3.4 as the definition of our TQFT.

According to Ref. [68], an $n$-dimensional TQFT for oriented manifolds (with no $G$-bundle structure), taking values in $\mathbb{C}$, requires the specification of the following information:

a. For every closed oriented $n$-dimensional manifold $\mathcal{M}$, a $\mathbb{C}$-number $\mathcal{Z}(\mathcal{M}) \in \mathbb{C}$.

b. For every closed oriented $(n-1)$-dimensional manifold $\mathcal{N}$, a $\mathbb{C}$-linear vector space $\mathcal{V}(\mathcal{N})$. When $\mathcal{N}$ is empty, the vector space $\mathcal{V}(\mathcal{N})$ is canonically isomorphic to $\mathbb{C}$.

c. For every oriented $n$-dimensional manifold $\mathcal{M}$, a vector $|\mathcal{Z}(\mathcal{M})\rangle$ of the vector space $\mathcal{V}(\partial \mathcal{M})$. When $\partial \mathcal{M} = \emptyset$, this vector space is cannonically identified with $\mathbb{C}$, and gives the same $\mathbb{C}$-number as we get in [a].

They should satisfy a series of consistency conditions that we do not specify here. We usually choose a set of orthonormal basis vectors $\{|\beta_{\partial \mathcal{M}}\rangle\}$ for $\mathcal{V}(\partial \mathcal{M})$, and then $|\mathcal{Z}(\mathcal{M})\rangle$ can be written as sum of basis vectors, i.e., $|\mathcal{Z}(\mathcal{M})\rangle = \sum_{\beta} \langle \beta_{\partial \mathcal{M}} | \mathcal{Z}(\mathcal{M}) \rangle | \beta_{\partial \mathcal{M}} \rangle$. We call the inner product $\langle \beta_{\partial \mathcal{M}} | \mathcal{Z}(\mathcal{M}) \rangle$ the partition function of $\mathcal{M}$ with *label* $|\beta_{\partial \mathcal{M}}\rangle$ put on $\partial \mathcal{M}$, and denote it by $\mathcal{Z}(\mathcal{M}; \beta_{\partial \mathcal{M}})$.

One of the most important facts of TQFT is that the partition function $\mathcal{Z}(\mathcal{M})$ of some $n$-manifold $\mathcal{M}$ can be evaluated via the gluing formula. Let us cut a closed $n$-manifold $\mathcal{M}$ along some $(n-1)$-manifold $\mathcal{N}$, then we get a new $n$-manifold $\mathcal{M}_{\mathrm{cut}}$ with boundary $\partial \mathcal{M}_{\mathrm{cut}} = \mathcal{N} \cup \overline{\mathcal{N}}$, where $\overline{\mathcal{N}}$ is the same manifold $\mathcal{N}$ with opposite orientation. From the axioms of TQFT we have the following gluing formula:

$$\mathcal{Z}(\mathcal{M}) = \sum_{\beta} \frac{\mathcal{Z}(\mathcal{M}_{\mathrm{cut}}; \beta_{\mathcal{N}})}{\langle \beta_{\mathcal{N}} | \beta_{\mathcal{N}} \rangle_{\mathcal{V}(\mathcal{N})}}. \tag{33}$$

Here $\{\beta_{\mathcal{N}}\}$ is a set of orthonormal basis for $\mathcal{V}(\mathcal{N})$.

From the gluing formula, it is clear that in order to calculate the partition function on some complicated manifold $\mathcal{M}$, we can chop $\mathcal{M}$ up into simpler pieces and calculate the partition functions of the individual pieces, so that we can obtain the partition function of the original manifold $\mathcal{M}$ with the help of the gluing formula Eq. (33). Therefore, in order to understand the TQFT, which in principle is defined on any manifold that can be arbitrarily complex, the hope is that it suffices to specify a relatively small amount of information about $\mathcal{M}_{\mathrm{cut}}$ and $\mathcal{N}$.

Yet the manifold $\mathcal{N}$ as an $(n-1)$-manifold can be very complicated as well, and thus $\mathcal{V}(\mathcal{N})$ can be very complicated. The idea of *2-extended TQFT* is to extend the construction once down, i.e., we wish to extend the construction of TQFT properly to incorporate the case where $\mathcal{N}$ has boundaries as well, and $\mathcal{V}(\mathcal{N})$ can also be obtained by gluing relatively simple pieces together. This extension will further simplify the analysis and the calculation of the partition function. We will also immediately see that the data of a UMTC can be manifestly incorporated into the construction, since we will soon put anyons on an $(n-2)$-manifold $\mathcal{O}$.

Specifically, to specify the data of a *2-extended* TQFT, beyond the data of an ordinary TQFT, we need to put an object of some $\mathbb{C}$-linear category, reminiscent of anyons, on "the boundary of the boundary". More precisely, on top of the information defining an ordinary TQFT, this further information includes

    d. For every closed oriented $(n-2)$-manifold $\mathcal{O}$, a $\mathbb{C}$-linear category $\mathcal{C}(\mathcal{O})$. When $\mathcal{O}$ is empty, the category $\mathcal{C}(\mathcal{O})$ is canonically isomorphic to the category of $\mathbb{C}$-linear vector spaces.

    e. For every oriented $(n-1)$-manifold $\mathcal{N}$, an object $\mathcal{V}(\mathcal{N})$ of the category $\mathcal{C}(\partial \mathcal{N})$. When $\partial \mathcal{N} = \emptyset$, this object is canonically identified with a $\mathbb{C}$-linear vector space, and gives the same $\mathbb{C}$-linear vector space as we get in [b].

Similar to the fact that a vector can be written as sum of basis vectors, an object can be written as a (direct) sum of simple objects $\{\beta_{\mathcal{O}}\}$ for $\mathcal{C}(\mathcal{O})$ a semisimple category. Therefore, similar to the previous analysis of ordinary TQFT, we will also associate an object $\beta_{\partial \mathcal{N}}$ to $\partial \mathcal{N}$ and call $\mathrm{Hom}(\beta_{\partial \mathcal{N}}, \mathcal{V}(\mathcal{N}))$ the vector space of $\mathcal{N}$ with *label* $\beta_{\partial \mathcal{N}}$ put on $\partial \mathcal{N}$, and denote it by $\mathcal{V}(\mathcal{N}; \beta_{\partial \mathcal{N}})$.

    From this construction, we define the vector space $\mathcal{V}(\mathcal{N}; \beta_{\partial \mathcal{N}})$ associated to $\mathcal{N}$ with boundary $\partial \mathcal{N} \neq \emptyset$, after putting labels $\beta_{\partial \mathcal{N}}$ on the boundary. Moreover, $\mathcal{V}(\mathcal{N}; \beta_{\partial \mathcal{N}})$ can be obtained by chopping $\mathcal{N}$ along some $(n-2)$-manifold $\mathcal{O}$ and using "gluing formula" similar to Eq. (33).

    Now we specialize to the TQFT that concerns us the most, i.e., a TQFT defined on 4-dimensional manifolds from the data of a UMTC $\mathcal{C}$. We can start using the language of anyons and topological state spaces. We define $\mathcal{C}(\mathcal{O})$ as $\mathcal{C}^{\otimes n}$ where $n$ is the number of connected components of $\mathcal{O}$. In particular, when $\mathcal{O} = \emptyset$, we say $n = 0$ and $\mathcal{C}^{\otimes 0}$ is defined as the UMTC with only object 1, i.e., a trivial anyon. Therefore, for closed $(n-1)$-manifold $\mathcal{N}$ with $\partial \mathcal{N} = \emptyset$, e.g., $S^3$, $\mathcal{V}(\mathcal{N})$ is a 1-dimensional $\mathbb{C}$-vector space, i.e., we have

$$\mathcal{V}(S^3) \simeq \mathbb{C}. \tag{34}$$

To finish the definition of the TQFT, we associate the object 1 to $\mathcal{N} = D^3$. When writting down the vector space of $D^3$ given some label on $\partial D^3 = S^2$, sometimes we need to associate a direction of the flow of anyons, i.e., whether an anyon comes into or out of the $S^2$ ball. This choice is similar to the choice of an orientation of $\mathcal{N}$, and when $\mathcal{N} = \partial \mathcal{M}$ it can be the same as or opposite to the orientation induced from $\mathcal{M}$. Now we assign $a_1, \dots$ anyons coming out of $S^2$ and $b_1, \dots$ anyons coming into $S^2$, and we have the canonical identification of the vector space given such labels as the topological state space of fusing $b_1, \dots$ anyons into $a_1, \dots$ anyons, i.e.,

$$\mathcal{V}\left(D^3; (a_1, \dots; b_1, \dots)\right) \simeq V_{b_1, \dots}^{a_1, \dots}. \tag{35}$$

After this assignment, we can in principle identify all vector spaces associated to $\mathcal{N}$ with some label on $\partial \mathcal{N}$. For example, for $S^2 \times D^1$ with trivial anyon on the boundary, we have

$$\mathcal{V}\left(S^2 \times D^1; \emptyset\right) \simeq \mathbb{C}. \tag{36}$$

For $S^1 \times D^2$ with trivial anyon on the boundary, we have

$$\mathcal{V}\left(S^1 \times D^2; \emptyset\right) \simeq \mathbb{C}^{|\mathcal{C}|}, \tag{37}$$

where $|\mathcal{C}|$ denotes the number of simple anyons in $\mathcal{C}$, and $\emptyset$ denotes the trivial anyon on the boundary. A basis vector in $\mathcal{V}(S^1 \times D^2; \emptyset)$ corresponds to putting an anyon loop labeled by $a \in \mathcal{C}$ along $S^1 \times \{pt\} \subset S^1 \times D^2$, where $\{pt\}$ denotes a point in $D^2$.

    We mention that in Ref. [34], $\mathcal{V}(\mathcal{N}; \beta_{\partial \mathcal{N}})$ is defined as the space of formal linear superpositions (with complex coefficients) of all anyon diagrams, which can end on the anyons labeled by $\beta_{\partial \mathcal{N}}$ on the boundary $\partial \mathcal{N}$, modulo the equivalence from local relations given by fusion of anyon lines, $F$-moves, and $R$-moves, i.e.,

$$\mathcal{V}(\mathcal{N}; \beta_{\partial \mathcal{N}}) = \mathbb{C}[\mathcal{C}(\mathcal{N}; \beta_{\partial \mathcal{N}})]/\sim, \tag{38}$$

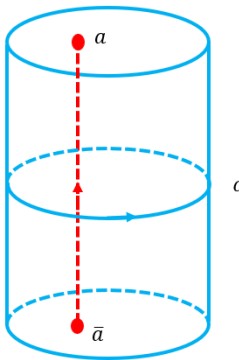

Figure 1: The illustration of some anyon diagram on $D^2 \times D^1$, with some anyon lines ending on anyon $a$ and $\overline{a}$ put on the boundary.

where $\mathcal{C}(\mathcal{N}; \beta_{\partial\mathcal{N}})$ denotes the set of all such anyon diagrams and $\sim$ is the equivalence given by these local relations. This serves as a nice diagrammatic illustration of the vector spaces defined above, as simply illustrated in Fig. 1. (See also Ref. [53] for the connection to Hamiltonian formalism.) In Appendix A.1 we rederive various vector spaces mentioned using the above definition, which serves as a nice consistency check.

Another piece of information that we should attribute to the vector space is the inner product in $\mathcal{V}(\mathcal{N}; \beta_{\partial\mathcal{N}})$. Following the expectation from gluing formula as in Eq. (40), the inner product in $\mathcal{V}(\mathcal{N}; \beta_{\partial\mathcal{N}})$ is supposed to be the partition function of $\mathcal{N} \times D^1$ with the labels on the boundary of $\mathcal{N}$ and $\overline{\mathcal{N}}$ attached to each other:

$$\langle x | y \rangle_{\mathcal{V}(\mathcal{N};\beta_{\partial\mathcal{N}})} = \mathcal{Z}(\mathcal{N} \times D^1; \overline{x} \cup y), \tag{39}$$

where $x, y$ are two vectors in $\mathcal{V}(\mathcal{N}; \beta_{\partial\mathcal{N}})$ and $\overline{x}$ is the dual vector of $x$ in the dual vector space $\mathcal{V}(\overline{\mathcal{N}}; \overline{\beta}_{\partial\overline{\mathcal{N}}})$.

For our purpose, we have to deal with manifold $\mathcal{M}$ with an additional $G$-bundle structure $\mathcal{G}$. Now we specialize to a finite symmetry group $G$, and thus a $G$-bundle $\mathcal{G}$ is fully characterized by the holonomy around all noncontractible cycles of $\mathcal{M}$. Such noncontractible cycles are generators of $\pi_1(\mathcal{M})$ that we call 1-cycles, and the holonomy assigns a group element $\mathbf{g} \in G$ to every generator of $\pi_1(\mathcal{M})$. To facilitate the usage of gluing formula, we can use a defect network to represent the holonomy, and the $G$-bundle structure is completely determined by which group elements (i.e., defects) we put on noncontractible cycles of $\mathcal{M}$, up to conjugation by elements in $G$.

According to the general recipe in Ref. [69], the category $\mathcal{C}(\mathcal{O})$ and the vector space $\mathcal{V}(\mathcal{N})$ should be equipped with a categorical $G$-action. This is precisely the data $\{\rho_{\mathbf{g}}; U_{\mathbf{g}}(a, b; c), \eta_a(\mathbf{g}, \mathbf{h})\}$ in Sec. 2.2 that defines a categorical $G$ action on $\mathcal{C}$. Labels should be acted by $\rho_{\mathbf{g}}$ or $\rho_{\mathbf{g}}^{-1}$ when crossing a defect corresponding to the group element $\mathbf{g}$ (whether it is $\rho_{\mathbf{g}}$ or $\rho_{\mathbf{g}}^{-1}$ will be explained later). Moreover, a 1-cycle of $\mathcal{M}$, thought of as a 1-morphism in the language of higher category, should be assigned a functor acting on vector spaces, while a 2-cycle of $\mathcal{M}$, thought of as a 2-morphism in the language of higher category, should be assigned a natural isomorphism acting on objects. The former precisely gives an extra piece $U_{\mathbf{g}}(a, b; c)$ in the partition function, which will be refered to as a $U$-factor; the latter gives an extra piece $\eta_a(\mathbf{g}, \mathbf{h})$ in the partition function, which will be refered to as an $\eta$-factor. Because of Eq. (24), we do not need 3- or higher morphisms to connect different compositions of 2-morphisms, hence introducing appropriate $U$-factors and $\eta$-factors is enough to determine such TQFT and calculate the partition function of it.

Finally, we collect the above results to write down the gluing formula for the TQFT, which

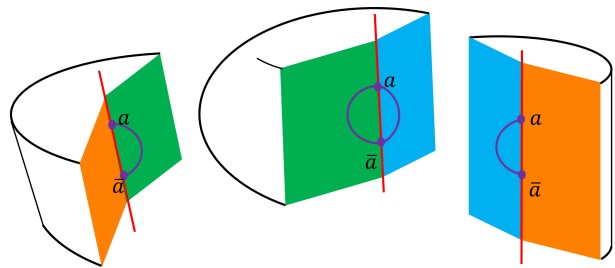

Figure 2: Illustration of the usage of gluing formula, where orange, green and blue faces are attached to each other while the red line denotes the (common) boundary of the faces.

is the main tool for the calculation of the partition function of the TQFT

$$\mathcal{Z}(\mathcal{M}, \mathcal{G}) = \sum_{\beta} \frac{\mathcal{Z}\left(\mathcal{M}_{\text{cut}}, \mathcal{G}_{\text{cut}}; \beta_{\mathcal{N}}, \beta_{\partial \mathcal{N}}\right)}{\langle \beta_{\mathcal{N}} | \beta_{\mathcal{N}} \rangle_{\mathcal{V}(\mathcal{N}; \beta_{\partial \mathcal{N}})}} \, . \tag{40}$$

Here, $\mathcal{M}$ is an $n$-dimensional closed manifold with a $G$-bundle structure $\mathcal{G}$, and we cut $\mathcal{M}$ along $\mathcal{N}$ to get a new manifold $\mathcal{M}_{\text{cut}}$ with boundary and corner. $\{\beta_{\partial \mathcal{N}}\}$ is a set of simple anyons we put on $\partial \mathcal{N}$ after the cut, while $\{\beta_{\mathcal{N}}\}$ is a set of orthonormal basis states for $\mathcal{V}(\mathcal{N}; \beta_{\partial \mathcal{N}})$. Notice that we should sum up both kinds of labels, collectively denoted by $\beta$.

With the help of the language of higher category [69], this definition of TQFT can be extended all the way to 0-dimensional points, giving rise to a *fully-extended TQFT*. For example, Crane-Yetter model has already been established as a fully-extended TQFT [48, 78, 79]. Although it is cumbersome to directly check that our construction satisfies all the consistency conditions of a fully-extended TQFT, we believe the TQFT that we are working with is indeed a fully-extended TQFT, given the infinity category presented in Ref. [48], equipped with $G$ action. For most of our exposition, it is enough to consider 2-extended TQFT. But being a fully-extended TQFT does allow us to chop the target 4-manifold $\mathcal{M}$ up in any way we like, without worrying about some small-dimensional submanifold on the boundary of which no data is defined. In particular, we can chop $\mathcal{M}$ up into $D^4$ pieces, which is essentially the handle decomposition that we will review in the next subsection.

## 3.3 Handle decomposition

In this section, we review basic facts about handle decomposition that will be used in this paper. Some standard textbooks of handle decomposition and 4-manifold topology are Refs. [80–82]. Handle decompositions of specific manifolds used in this paper are summarized in Appendix E.

Handle decomposition is nothing but a canonical way of chopping an $n$-dimensional manifold up into simple pieces of $D^n$, where every $D^n$ piece is called a *handle*. Every smooth manifold admits a handle decomposition [80]. A handle decomposition of an $n$-manifold $\mathcal{M}$ is a decomposition of $\mathcal{M}$ into 0-handles, 1-handles, $\cdots$, $n$-handles. The union of all 0-handles, 1-handles, $\cdots$, $m$-handles is called the $m$-handlebody of this handle decomposition for $m \leqslant n$, temporarily denoted by $\mathcal{M}^{(m)}$ here. A handle decomposition can always be done such that lower-handles are first specified, and higher handles are attached along their *attaching regions* to the boundary of the already-specified lower handlebodies by embedding maps. Specifically, for an $n$-dimensional $k$-handle, it is topologically equivalent to $D^k \times D^{n-k}$ and its attaching region is the part of its boundary that is topologically equivalent to $\partial(D^k) \times D^{n-k} \cong S^{k-1} \times D^{n-k}$.

The attaching region is attached to $\mathcal{M}^{(k-1)}$ via an *embedding map*:[8]

$$\varphi : S^{k-1} \times D^{n-k} \to \partial \mathcal{M}^{(k-1)}. \tag{41}$$

A handle decomposition is specified by specifying all handles and the embedding maps that attach all handles together. See Fig. 3 for an illustration of 1-handles and 2-handles together with their attaching regions.

There is some formal analogy between handle decompositions and cell decompositions. In fact, it is often useful to think of a handle decomposition as a "thickened" version of a cell decomposition. For example, one can take a triangulation or cellulation of an $n$-dimensional manifold $\mathcal{M}$, and thicken the 0-cells into $n$-balls $D^n$. Next, one can thicken the 1-cells to $n$-balls as well, and glue them to the boundary of 0-cells along two $D^{n-1}$ pieces of $S^0 \times D^{n-1} \subset \partial(D^n)$. The 2-cells can be thickened to $n$-balls, and glued to the boundary of 0- and 1- cells along $S^1 \times D^{n-2}$, and so on.

For a connected $n$-manifold $\mathcal{M}$, we can choose to have only one 0-cell. A handle decomposition of $\mathcal{M}$ with a unique 0-handle then determines a presentation of $\pi_1(\mathcal{M})$. Namely, each 1-handle together with the 0-handle forms an $S^1 \times D^{n-1}$ and determines a generator of $\pi_1(\mathcal{M})$, and the attaching region $S^1 \times D^{n-2}$ of each 2-handle gives a relation among the generators (as this $S^1$ is always contractible). This is also what we expect from cell decompositions. We will sometimes call the cycle formed this way from joining a 1-handle with the 0-handle the *induced (1-)cycle* of the 1-handle, as shown in Fig. 4.

Given a $k$-handle, in order to specify how it is attached to lower handles $\mathcal{M}^{(k-1)}$, we just need to specify the attaching region, which requires the following two pieces of information:

1. How $S^{k-1} \times \{pt\}$ is embedded in $\partial \mathcal{M}^{(k-1)}$, where $\{pt\} \in D^{n-k}$ is any point in the interior of $D^{n-k}$.

2. How to choose a trivialization in the tubular neighborhood of $S^{k-1} \times \{pt\}$ in $\partial \mathcal{M}'$ that is supposed to be identified with $\partial(D^k) \times D^{n-k}$.

The second piece of information is called the *framing* of the $k$-handle. This information is not directly present in cell decomposition. In particular, the framing of a 1-handle is classified by $\pi_0(O(1)) \cong \mathbb{Z}_2$, and is given by whether the induced cycle of the 1-handle is orientable or not. With slight abuse, if this induced cycle is orientable (non-orientable), we will say that the 1-handle is orientable (non-orientable). The framing of 2-handle is classified by $\pi_1(O(2)) \cong \mathbb{Z}$, which is the self-intersection number of $S^1 \times \{pt\}$ on the boundary of the 0-handle (see Ref. [81] for more information regarding this).

Now let us specialize to 4-dimensional manifolds. In order to illustrate the handle decomposition, we introduce *Kirby diagrams*. Suppose we have some 4-dimensional closed connected manifold $\mathcal{M}$. We assume that there is a unique 0-handle $D^4$, whose boundary $S^3$ can be thought of as $\mathbb{R}^3 \cup \{\infty\}$. We then try to draw the attaching regions of the remaining handles in $\mathbb{R}^3$. The attaching region of each 1-handle is two copies of $D^3$, which we draw as a pair of round balls. For 2-handles whose attaching regions are $S^1 \times D^2$, we draw the image of $S^1 \times \{pt\} \subset S^1 \times D^2$ on $\mathbb{R}^3$, and pay attention that in $\mathbb{R}^3$ circles can be knotted and linked. It is known that 3-handles and 4-handles are uniquely defined once we have determined how 1-handles and 2-handles are attached.

We must then deal with framings. Specifically, given whether the induced cycle of some 1-handle is orientable or non-orientable, we need to connect points on the two balls in different

---

[8]When there are multiple $k$-handles, the first of them is attached to $\mathcal{M}^{(k-1)}$ in this way, which results a manifold $\mathcal{M}^{(k-1),1}$. Then one needs to attach the second $k$-handle to $\mathcal{M}^{(k-1),1}$ in a similar way. This procedure continues until all $k$-handles are attached to result in $\mathcal{M}^{(k)}$. The manifold obtained this way is independent of the sequence of attachment.

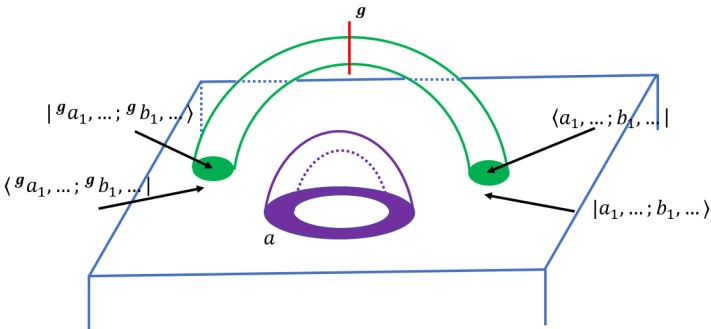

Figure 3: Illustration of a blue 0-handle, a green 1-handle and a purple 2-handle together with labels assigned to their attaching regions. The green shaded regions are the attaching regions $S^0 \times D^3$ of the 1-handle, and the purple shaded regions are the attaching regions $S^1 \times D^2$ of the 2-handle. The red line displays a defect, which crosses the 1-handle with the section being $D^3$. We associate an anyon $a$ to the 2-handle. We also associate a vector $|a_1, \ldots; b_1, \ldots\rangle$ and a dual vector $\langle a_1, \ldots; b_1, \ldots|$ to the attaching regions living on the 0-handle side and 1-handle side, respectively (these two sides are identified by the embedding map that attaches the 1-handle to the 0-handle).

ways. Specifically, the two balls are glued together by the 1-handle with the opposite (same) orientation if the cycle is orientable (non-orientable). In this paper, for an orientable 1-handle points related to each other by mirror reflection through the plane perpendicularly bisecting the lines joining their centers are connected to each other, as in Ref. [80, 81]. For a non-orientable 1-handle, we use the convention that parallel points, e.g., the bottom points or the top points of two balls, are connected to each other by the 1-handle, in contrast to the convention in Ref. [81]. These are illustrated in Fig. 4. For 2-handles, we need to add the correct amount of topological twists to account for the correct framing. One important way to determine the linking and framing of 2-handles is through the intersection form and mod-2 intersection form of $\mathcal{M}$ [80], which can be calculated relatively easily in algebraic topology.

## 3.4 Recipe for calculating the partition function

Having laid down the foundation, in this subsection we spell out the recipe for calculating the partition function on any (3+1)$D$ manifold $\mathcal{M}$ equipped with a $G$-bundle $\mathcal{G}$, given the data of a UMTC $\mathcal{C}$ and the data of symmetry action of some finite group $G$ on $\mathcal{C}$. This recipe is summarized by Eq. (44). Note that $\mathcal{G}$ is fully characterized by the holonomy around all noncontractible cycles of $\mathcal{M}$, and we will use a defect network to represent the holonomy. In Appendix C, without resorting to its origin or its relation to gluing formula, we directly check that the partition function constructed here indeed satisfies various desired properties, including the independence on the handle decomposition, gauge invariance, cobordism invariance, etc., by directly manipulating the formula in Eq. (44).

The basic formula for the calculation is the gluing formula Eq. (40). For a specific handle decomposition of the manifold $\mathcal{M}$, we have [48]

$$\mathcal{Z}(\mathcal{M}, \mathcal{G}) = \sum_{\beta \in \mathcal{L}} \prod_{j=0}^{4} \prod_{h \in j\text{-handle}} \frac{\mathcal{Z}(h; \beta_{\partial h})}{\langle \beta_{\tilde{\partial} h} | \beta_{\tilde{\partial} h} \rangle_{\mathcal{V}(\tilde{\partial} h; \beta_{\partial(\tilde{\partial} h)})}} . \tag{42}$$

Here $\beta \in \mathcal{L}$ denotes all labels on the attaching regions of all $j$-handles, $\mathcal{Z}(h; \beta_{\partial h})$ is the partition function of some $j$-handle $h$ with label $\beta_{\partial h}$ on the boundary $\partial h$, and $\langle \beta_{\tilde{\partial} h} | \beta_{\tilde{\partial} h} \rangle_{\mathcal{V}(\tilde{\partial} h; \beta_{\partial(\tilde{\partial} h)})}$ is

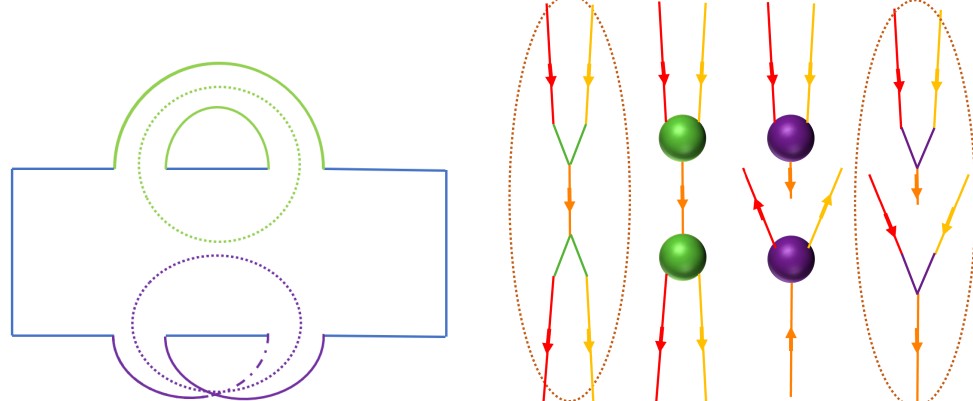

Figure 4: Left: Illustration of a greeen orientable 1-handle and a purple non-orientable 1-handle, attached to the blue 0-handle. The manifold is supposed to be 4-dimensional but we draw a 2-dimensional plane for illustration. The dashed green circle and the the dashed purple circle are the induced cycles of the two 1-handles. Right: the Kirby diagrams for the green and purple 1-handles (the two figures in the middle), together with the anyon diagrams associated with these Kirby diagrams (the two figures in dashed ellipses). Pay attention how points on the two $D^3$ components of attaching regions $S^0 \times D^3$ are connected to each other via the 1-handle.

the squared norm of the state $|\beta_{\tilde{\partial} h}\rangle$ in the vector space $\mathcal{V}\left(\tilde{\partial} h; \beta_{\partial(\tilde{\partial} h)}\right)$ associated with the 3D manifold of the attaching region $\tilde{\partial} h$ of $h$. From the formula we need to calculate various norms and the partition function on various handles given a prescribed label. We repeat the calculation of Refs. [34, 47, 48] in Appendix A, which concerns manifolds without a general $G$-bundle structure. A major innovation we introduce in this paper is how to deal with a $G$-bundle structure, and we discuss it in detail for finite group $G$ below.

The recipe for calculating the partition function $\mathcal{Z}(\mathcal{M}, \mathcal{G})$ of the manifold $\mathcal{M}$ with a $G$-bundle structure $\mathcal{G}$ on $\mathcal{M}$, with $G$ a finite group, is summarized here.

1. Identify a handle decomposition of the manifold $\mathcal{M}$. On each 1-handle put appropriate defects according to the $G$-bundle structure $\mathcal{G}$, as in Fig. 3.

2. The $S^1$ boundary of each 2-handle is separated by the defects into segments. Associate an anyon $a$ to an arbitrary segment on the $S^1$ boundary of each 2-handle, and the anyons on the other segments are related to $a$ by the $G$-actions given by the defects. Write down the $\eta$-factor coming from the natural isomorphism for $a$ that connects the functor of successive $G$-actions and the identity functor. (See Remark g below for more details.)

3. Associate a dual vector $\langle a_1, \ldots; b_1, \ldots |_{\mu \ldots} K^{q(\mathbf{g})}$[9] and a vector $|^{\mathbf{g}} a_1, \ldots; {}^{\mathbf{g}} b_1, \ldots \rangle_{\tilde{\mu} \ldots}$ to the two $D^3$ planes of the attaching region $S^0 \times D^3$ of every 1-handle as in Fig. 5, where $a_1, \ldots$ and $b_1, \ldots$ are labels of anyons running out of and into the lower $D^3$ plane of the attaching region of the 1-handle, respectively. Write down the $U$-factor from[10]

$$\langle a_1, \ldots; b_1, \ldots |_{\mu \ldots} K^{q(\mathbf{g})} \rho_{\mathbf{g}}^{-1} |^{\mathbf{g}} a_1, \ldots; {}^{\mathbf{g}} b_1, \ldots \rangle_{\tilde{\mu} \ldots} = U_{\mathbf{g}}^{-1} ({}^{\mathbf{g}} a_1, \ldots; {}^{\mathbf{g}} b_1, \ldots)_{\tilde{\mu} \ldots, \mu \ldots}. \quad (43)$$

4. Evaluate the anyon diagram from the Kirby diagram $\langle K \rangle$ of $\mathcal{M}$, given the prescribed anyon labels associated to the $S^1$ lines corresponding to 2-handles and vectors associated to the $D^3$ balls corresponding to 1-handles as in Fig. 3.

---

[9]See Remark e in the following paragraphs for some further explanation of the factor $K^{q(\mathbf{g})}$.
[10]The assignment of $\rho_{\mathbf{g}}^{-1}$ instead of e.g., $\rho_{\bar{\mathbf{g}}}$ is just to match the convention of Ref. [38].

5. Assemble the result as follows:

$$
\mathcal{Z}(\mathcal{M}, \mathcal{G}) = D^{-\chi + 2(N_4 - N_3)} \times \sum_{\text{labels}} \left( \frac{\prod\limits_{2 \text{ handle } i} d_{a_i}}{\prod\limits_{1 \text{ handle } x} \left( \prod\limits_{2 \text{ handle } j \text{ across } x} d_{a_j} \right)^{1/2}} \right. \tag{44}
$$

$$
\left. \times \left( \prod_i (\eta\text{-factors})_i \right) \times \left( \prod_x (U\text{-factors})_x \times \langle K \rangle \right) \right).
$$

Here $N_k$ is the number of $k$-handles in this handle decomposition, and $\chi \equiv N_0 - N_1 + N_2 - N_3 + N_4$ is the Euler number of $\mathcal{M}$.

There are a few extra points that may clarify the meanings or ease the computation. We summarize them below:

a. Without loss of generality, we assume that $\mathcal{M}$ is connected. Then the numbers of 0- and 4-handles in the handle decomposition of $\mathcal{M}$ can be chosen to be 1. If $\mathcal{M}$ is disconnected, then the partition function is the product of the partition functions on each of its disconnected components.

b. Since $G$ is finite, the $G$-bundle is fully characterized by the holonomy around noncontractible cycles. Recall that noncontractible cycles are the induced cycles of some 1-handles. Therefore, we interpret a holonomy labeled by group element $\mathbf{g}$ around such a cycle as a defect we put across the associated 1-handle along its $D^3$ plane, such that each anyon gets acted upon by $\mathbf{g}$ when crossing this defect. Without loss of generality, we assume that no defect intersects the 0-handle, which can always be achieved.

c. If $G$ contains unitary symmetries only, $\mathcal{M}$ is always oriented. On the other hand, in the presence of anti-unitary symmetries, $\mathcal{M}$ can be an unorientable manifold with a nontrivial first Stiefel-Whitney class $w_1^{TM}$. Moreover, there must be a $\mathbf{g}$-defect on each nonorientable cycle, where $\mathbf{g}$ is an anti-unitary symmetry. On the anyon diagram, anyons should flip the direction of the flow after crossing such $\mathbf{g}$-defect, as illustrated in Figs. 4 (pay special attention to the right two graphs of the lower figure).

d. It is of paramount importance to keep track of the framing of 1-handles and 2-handles when drawing and evaluating the Kirby diagram. Let us emphasize that we use the convention according to which, for an orientable 1-handle, points on each pair of $D^3$ balls related to each other by a reflection with respect to the plane perpendicularly bisecting the centers of these $D^3$ balls are connected to each other by the 1-handle, while, for a non-orientable 1-handle, points on the pair of $D^3$ balls are connected to each other by the 1-handle, if these points are related to each other by a translation that relates the two $D^3$ balls. This convention is illustrated in Fig. 4. For 2-handles, we should pay special attention to whether we should add extra topological twists/kinks to the Kirby diagram as in Eq. (11), accounting for the correct self-intersection number of the $S^1$ loop associated to the 2-handle.

e. We further comment on assigning vectors and dual vectors to 1-handles and 0-handles. Note that when we attach a 1-handle and a 0-handle, we should assign a vector and a dual vector to the 1-handle and the 0-handle respectively as in Fig. 3. In a Kirby diagram, we can put the two $D^3$ balls corresponding to a single 1-handle on the upper and lower

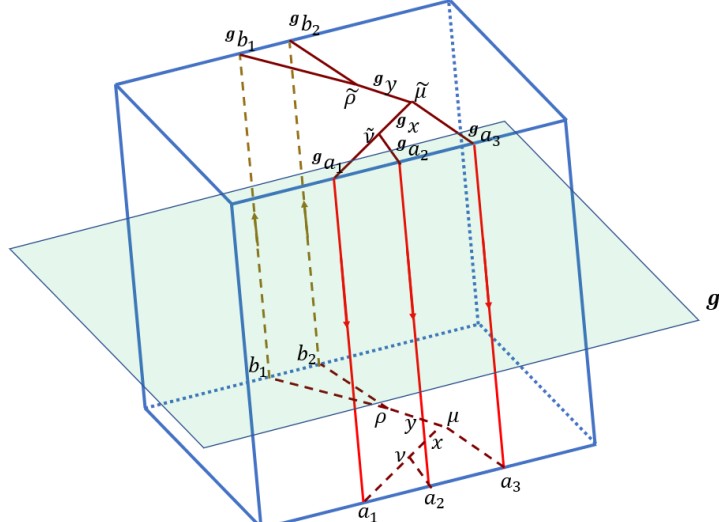

Figure 5: Illustration of the 1-handle. The 1-handle has the topology of a $D^4$ but we draw it as a $D^3$ for illustration. The shaded region represents a **g**-defect for unitary **g**, which cuts through the 1-handle along its $D^3$ plane (drawn as a $D^2$ plane here). The lower plane displays a dual vector $\langle a_1, a_2, a_3; b_1, b_2|_{(x,y,\mu,\nu,\rho)}$ that lives in the vector space associated to $D^3$, i.e., $\mathcal{V}\left(D^3;(a_1, a_2, a_3; b_1, b_2)\right) \simeq V^{a_1,a_2,a_3}_{b_1,b_2}$, while the upper plane displays a vector $|{}^g a_1, {}^g a_2, {}^g a_3; {}^g b_1, {}^g b_2\rangle_{({}^g x, {}^g y, \tilde{\mu}, \tilde{\nu}, \tilde{\rho})}$. The evaluation of the diagram is given by Eq. (A.8) if no defect is present. In the presence of the **g**-defect we just need to add the $U$-factor as in Eq. (43). See Remarks d,e for further treatment when **g** is anti-unitary.

parts of the diagram, and associate the dual vector $\langle {}^g a_1, \ldots; {}^g b_1, \ldots|$ and the vector $K^{q(\mathbf{g})}|a_1, \ldots; b_1, \ldots\rangle$ to the upper and lower ball, respectively. As illustrated in the lower figure of Fig. 4, according to the convention in Remark d, if **g** is anti-unitary we draw $K^{q(\mathbf{g})}|a_1, \ldots; b_1, \ldots\rangle$ in the same way as a dual vector on the anyon diagram. According to this convention, on the 1-handle we assign a dual vector $\langle a_1, \ldots; b_1, \ldots|K^{q(\mathbf{g})}$ and a vector $|{}^g a_1, \ldots; {}^g b_1, \ldots\rangle$, and therefore the $U$-factor is given by Eq. (43), as illustrated in Fig. 5.

f. In this convention, anyons running "upward" in the 1-handles are acted upon by $\rho_{\mathbf{g}}$ while anyons running "downward" in the 1-handles are acted upon by $\rho_{\mathbf{g}}^{-1}$, when we put a **g**-defect across the 1-handle, as in Fig. 6.

g. Here we explain how to get $\eta$-factors in detail. In general, the $S^1$ line of a 2-handle is separated into multiple segments by the defects. Starting from an arbitrary segment on this $S^1$ line with anyon label $a$, we move along the $S^1$ line on the Kirby diagram and use the above prescription to get a functor describing the successive symmetry actions, which takes the form $\rho_{\mathbf{g}_1}^{s_1} \circ \rho_{\mathbf{g}_2}^{s_2} \circ \cdots$, where $\mathbf{g}_{1,2,\ldots}$ denotes the defect and $s_{1,2,\ldots} = 1$ ($s_{1,2,\ldots} = -1$) if the anyon crosses this defect in the "upward" ("downward") direction. Note that this $S^1$ is contractible, so consistency requires that the combination of all these defects is a trivial defect, i.e., $\mathbf{g}_1^{s_1} \mathbf{g}_2^{s_2} \cdots = \mathbf{1}$. The $\eta$-factor associated with this 2-handle comes from the natural isomorphism that connects $\rho_{\mathbf{g}_1}^{s_1} \circ \rho_{\mathbf{g}_2}^{s_2} \circ \cdots$ and the identity functor. The explicit expression of the $\eta$-factor is not unique, and different expressions can be converted into each other using Eq. (24). In Appendix B, we present such an expression explicitly. In the following, we demonstrate this analysis via concrete examples.

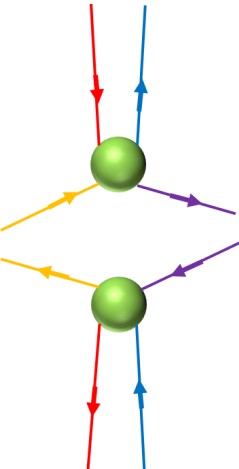

Figure 6: Suppose a **g**-defect is on the green 1-handle. Following their arrows, anyons in the red and yellow (blue and purple) lines enter the upper (lower) $D^3$ ball and exit from the lower (upper) $D^3$ ball, and they are said to move "downward" ("upward") and are acted by $\rho_{\mathbf{g}}^{-1}$ ($\rho_{\mathbf{g}}$).

First consider the situation where $C$ is a $\mathbb{Z}_2$ generator and some anyon $a$ associated to a 2-handle crosses a $C$-defect twice. Then there is a natural isomorphism $\eta(C,C)$ connecting $\rho_C \circ \rho_C$ to the identity functor, which gives the desired $\eta$-factor to be $\eta_a(C,C)$. With slight abuse of notation, we will say that $\rho_C \circ \rho_C$ acting on $a$ gives a phase $\eta_a(C,C)$. As another example, consider the situation where $C_1, C_2$ are any two generators of a unitary symmetry such that $C_1 C_2 = C_2 C_1$, and $a$ is acted upon by $\rho_{C_2} \circ \rho_{C_1} \circ \rho_{C_2}^{-1} \circ \rho_{C_1}^{-1}$. Then connecting $\rho_{C_2} \circ \rho_{C_1}$ to $\rho_{C_2 C_1}$ gives a phase $\eta_a(C_2, C_1)$, while connecting $\rho_{C_2 C_1}$ to $\rho_{C_1} \circ \rho_{C_2}$ gives another phase $1/\eta_a(C_1, C_2)$. By definition, the composition of $\rho_{C_1} \circ \rho_{C_2}$ with $\rho_{C_2}^{-1} \circ \rho_{C_1}^{-1}$ is the identity functor. Therefore, the desired $\eta$-factor is $\frac{\eta_a(C_2, C_1)}{\eta_a(C_1, C_2)}$.

## 4 Examples: Finite group symmetry

After spelling out the recipe for calculation, in this section we go to specific examples of finite group symmetries that concern us the most, including the case of no symmetry (i.e., Crane-Yetter model), $\mathbb{Z}_2^T$, $\mathbb{Z}_2 \times \mathbb{Z}_2$ and $\mathbb{Z}_2^T \times \mathbb{Z}_2^T$. We will calculate the anomaly indicators of these symmetries, which are the partition functions defined in Sec. 3 evaluated on appropriate manifolds with certain bundle structures (see Appendix D for how to identify the manifolds and bundle structures that are relevant to the anomaly indicators). Especially, the calculation of the anomaly indicators of the mutual anomaly of $\mathbb{Z}_2 \times \mathbb{Z}_2$ and $\mathbb{Z}_2^T \times \mathbb{Z}_2^T$ is new, and their results are given by Eq. (53) and Eq. (55), respectively.

### 4.1 No symmetry

Even in the absence of any symmetry, the partition function is not completely trivial and it reduces to the original Crane-Yetter model [49,50]. Since the partition function is a cobordism invariant, to evaluate the partition function on any oriented 4$D$ manifold, we just need to evaluate it on the generating manifold of $\Omega_4^{SO}(\star) \cong \mathbb{Z}$, which is $\mathbb{CP}^2$.

The minimum handle decomposition of $\mathbb{CP}^2$ contains 1 0-handle, 1 2-handle and 1 4-handle, as listed in Table 1. No symmetry defect is present, so there is no appearance of $\eta$-factor or $U$-factor. Given label $a$ to the anyon associated with the 2-handle, the Kirby diagram

Table 1: Basic Information about handle decomposition of various manifolds used in Section 4. See Appendix E for more information about their handle decomposition.

| Manifold $\mathcal{M}$ | Orientability | 0-handles | 1-handles | 2-handles | 3-handles | 4-handles |
|:---:|:---:|:---:|:---:|:---:|:---:|:---:|
| $\mathbb{CP}^2$ | Yes | 1 | 0 | 1 | 0 | 1 |
| $\mathbb{RP}^4$ | No | 1 | 1 | 1 | 1 | 1 |
| $\mathbb{RP}^3 \times S^1$ | Yes | 1 | 2 | 2 | 2 | 1 |
| $\mathbb{RP}^2 \times \mathbb{RP}^2$ | No | 1 | 2 | 3 | 2 | 1 |

is evaluated as

$$\left\langle \rule{0pt}{0pt} \right\rangle = d_a \theta_a . \tag{45}$$

The topological twist reflects the $+1$ intersection number of $\mathbb{CP}^2$. Assembling all factors as in Eq. (44), we have

$$\mathcal{Z}\left(\mathbb{CP}^2\right) = \frac{1}{D}\sum_a d_a^2 \theta_a . \tag{46}$$

It is well-known that the right hand side of this expression is related to the *chiral central charge* $c$ mod 8, i.e.,

$$e^{2\pi i c/8} = \frac{1}{D}\sum_a d_a^2 \theta_a . \tag{47}$$

Physically, the partition function $\mathcal{Z}(\mathbb{CP}^2)$ and the chiral central charge gives the thermal Hall conductance of the $(2+1)D$ topological order, which is very well-known in the literature [63, 83].

An important fact in 4-dimensional topology is that any oriented manifold $\mathcal{M}$ is cobordant with $\#\left(\mathbb{CP}^2\right)^{\sigma(\mathcal{M})}$, i.e., the connected sum of $\sigma(\mathcal{M})$ copies of $\mathbb{CP}^2$, where $\sigma(\mathcal{M})$ is the *intersection number* of $\mathcal{M}$ [82]. Then the partition function on any oriented manifold $\mathcal{M}$ is given by

$$\mathcal{Z}_{\text{CY}}(\mathcal{M}) = e^{(2\pi i c/8)\cdot\sigma(\mathcal{M})} , \tag{48}$$

which is indeed the correct form of the Crane-Yetter model [50–52].

## 4.2   $\mathbb{Z}_2^T$

For the group $\mathbb{Z}_2^T$, the bordism group that we should consider is $\Omega_4^O(\star) \cong \mathbb{Z}_2 \oplus \mathbb{Z}_2$, and the two $\mathbb{Z}_2$ factors are generated by $\mathbb{CP}^2$ and $\mathbb{RP}^4$, respectively. $\mathcal{I}_0 \equiv \mathcal{Z}\left(\mathbb{CP}^2\right)$ has been calculated in Section 4.1 and given by Eq. (46), which is referred to as the "beyond-cohomology" anomaly indicator for $\mathbb{Z}_2^T$. In fact, in the presence of anti-unitary symmetry, there is always this "beyond-cohomology" anomaly indicator $\mathcal{I}_0 = \mathcal{Z}\left(\mathbb{CP}^2\right)$. Below we present the calculation for the partition function on $\mathbb{RP}^4$, which is referred to as the "in-cohomology" anomaly indicator for $\mathbb{Z}_2^T$. These anomaly indicators are first conjectured in Ref. [32] and derived in Ref. [34]. We will see that this is the simplest example involving 1-handle in the handle decomposition of the manifold.

The minimal handle decomposition of $\mathbb{RP}^4$ contains 1 0-handle, 1 1-handle, 1 2-handle, 1 3-handle and 1 4-handle, as listed in Table 1. Since $\mathbb{RP}^4$ is non-orientable, we should consider

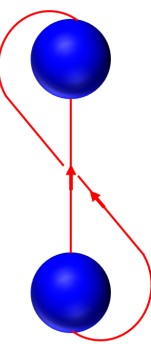

Figure 7: The Kirby diagram of $\mathbb{RP}^4$. The two blue balls illustrate the attaching region of the 1-handle and the red lines illustrate the attaching region of the 2-handle. The 1-handle is nonorientable.

the effect of the "$\mathbb{Z}_2^T$-defect", or more commonly referred to as a *crosscap*, across the 1-handle. Namely, in the Kirby diagram shown in Fig. 7, the 1-handle (represented by the pair of blue balls) is crossed by such a $\mathcal{T}$-defect, with $\mathcal{T}$ the generator of $\mathbb{Z}_2^T$.

Now we put anyon $a$ and $^\mathcal{T}a$ on the $S^1$ line of the 2-handle. Following remark g in Sec. 3.4, the $\eta$-factor from the 2-handle is given by action $\rho_\mathcal{T} \circ \rho_\mathcal{T}$ on $a$, which is $\eta_a(\mathcal{T},\mathcal{T})$. On the 1-handle we associate a dual vector $\langle^\mathcal{T}a;a|$ and a vector $|a;^\mathcal{T}a\rangle$, and they are nonzero only when $^\mathcal{T}a = a$. Pay attention that after touching the crosscap, the direction of the flow of one of the anyons should change. Specifically, comparing Fig. 7 and the diagram in Eq. (49), the curvy red line changes the direction of the flow. Also note that when $^\mathcal{T}a = a$, $\eta_a(\mathcal{T},\mathcal{T})$ is invariant under the gauge transformation Eq. (26). According to Eq. (27), the $U$-factor from the 1-handle is simply 1. Finally, the Kirby diagram in Fig. 7 can be translated to the following anyon diagram and evaluated as

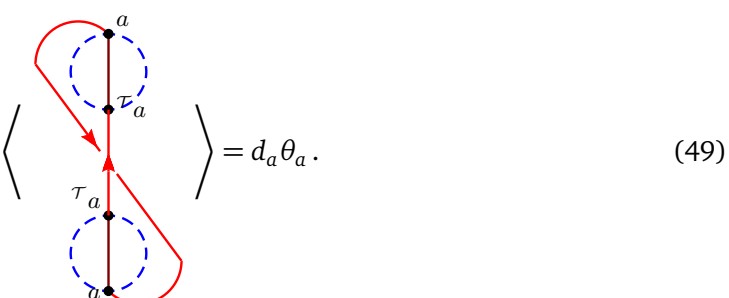

$$\left\langle \quad \right\rangle = d_a \theta_a. \tag{49}$$

Again, there is a factor of $\theta_a$ coming from the +1 framing of the 2-handle.

Assembling all factors, we have

$$\mathcal{Z}\left(\mathbb{RP}^4;\mathcal{T}\right) = \frac{1}{D} \sum_{\substack{a \\ ^\mathcal{T}a=a}} d_a \theta_a \times \eta_a(\mathcal{T},\mathcal{T}). \tag{50}$$

This is preciesly the in-cohomology anomaly indicator for $\mathbb{Z}_2^T$ symmetry [32, 34].

In summary, the beyond-cohomology anomaly indicator for $\mathbb{Z}_2^T$ symmetry is $\mathcal{I}_0 = \mathcal{Z}(\mathbb{CP}^2)$, given by Eq. (46), while the in-cohomology anomaly indicator for $\mathbb{Z}_2^T$ symmetry is $\mathcal{I}_1 = \mathcal{Z}\left(\mathbb{RP}^4;\mathcal{T}\right)$, given by Eq. (50).[11] As such, the anomaly/partition function $\mathcal{O}$ can be written as

$$\mathcal{O} = (\mathcal{I}_0)^{\left(w_2^{TM}\right)^2} \cdot (\mathcal{I}_1)^{t^4}, \tag{51}$$

---

[11]Notice that $\mathcal{I}_0$ and $\mathcal{I}_1$ are numbers that will serve as coefficients in front of a certain basis in the expression of the anomaly.

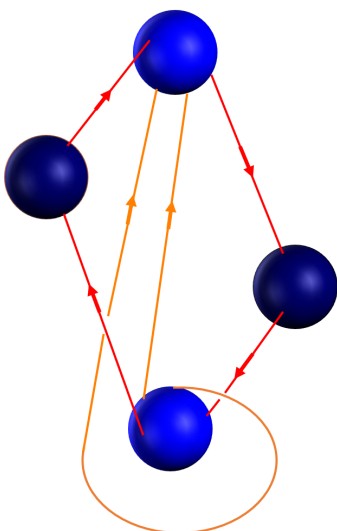

Figure 8: The Kirby diagram of $\mathbb{RP}^3 \times S^1$. The blue balls and dark blue balls illustrate the two 1-handles, and the red lines and orange lines illustrate the two 2-handles. Both 1-handles are orientable.

where $t$ is the generator of $\mathcal{H}^1(\mathbb{Z}_2, \mathbb{Z}_2)$, and $\left(w_2^{TM}\right)^2$ is the generator of the beyond-cohomology piece of anomaly.

### 4.3 $\mathbb{Z}_2 \times \mathbb{Z}_2$

Let us go to the simplest non-trivial group involving unitary symmetry only: $\mathbb{Z}_2 \times \mathbb{Z}_2$. The anomalies of $\mathbb{Z}_2 \times \mathbb{Z}_2$ in $(2+1)$-dimension are classified by $\mathbb{Z}_2 \oplus \mathbb{Z}_2$, and the representative manifold is $\mathbb{RP}^3 \times S^1$ with two different $\mathbb{Z}_2 \times \mathbb{Z}_2$-bundles, one with a $C_1$ defect across the noncontractible cycle of $\mathbb{RP}^3$ and a $C_2$ defect across $S^1$, and the other with a $C_2$ defect across the noncontractible cycle of $\mathbb{RP}^3$ and a $C_1$ defect across $S^1$, where $C_1$ and $C_2$ are two $\mathbb{Z}_2$ generators of $\mathbb{Z}_2 \times \mathbb{Z}_2$.

Without loss of generality, let us first put a $C_1$ defect across the noncontractible cycle of $\mathbb{RP}^3$ and a $C_2$ defect across $S^1$. The minimum handle decomposition of $\mathbb{RP}^3 \times S^1$ contains 1 0-handle, 2 1-handle, 2 2-handle, 2 3-handle and 1 4-handle, as listed in Table 1. The Kirby diagram and the associated anyon diagram are drawn in Figs. 8 and 9, respectively.

Now we put anyon $a$ and $b$ on a red and orange segment of the 2-handles, respectively, and anyons on other segments can be obtained by symmetry actions on $a$ and $b$, as shown in Fig. 9. From the two 1-handles we have two constraints $^{C_1}a = a$ and $a \times b \times {}^{C_1}b \to {}^{C_2}a$. The second constraint means that $^{C_2}a$ should be in the fusion channel of $a$, $b$ and $^{C_1}b$.

The $\eta$-factor from anyon $a$ is given by action $\rho_{C_1}^{-1} \circ \rho_{C_2} \circ \rho_{C_1} \circ \rho_{C_2}^{-1}$ on $a$, which is $\frac{\eta_a(C_2, C_1)}{\eta_a(C_1, C_2)}$. The $\eta$-factor from anyon $b$ is given by action $\rho_{C_1}^{-1} \circ \rho_{C_1}^{-1}$ on $b$, which is $\frac{1}{\eta_b(C_1, C_1)}$. The $U$-factor from the blue 1-handle is $U_{C_1}^{-1}(a, b; x)_{\mu\tilde{\mu}} U_{C_1}^{-1}(x, {}^{C_1}b; {}^{C_2}a)_{\nu\tilde{\nu}}$, while the $U$-factor from the darkblue 1-handle is simply 1 according to Eq. (27). Finally, we need to evaluate the anyon diagram Fig. 8, which is

$$d_a d_b \frac{\theta_x}{\theta_a} \left(R_u^{b, {}^{C_1}b}\right)_{\rho\sigma} \left(F_{C_2 a}^{a, b, {}^{C_1}b}\right)^*_{(x, \tilde{\mu}, \tilde{\nu})(u, \sigma, \alpha)} \left(F_{C_2 a}^{a, {}^{C_1}b, b}\right)_{({}^{C_1}x, \mu, \nu)(u, \rho, \alpha)}. \tag{52}$$

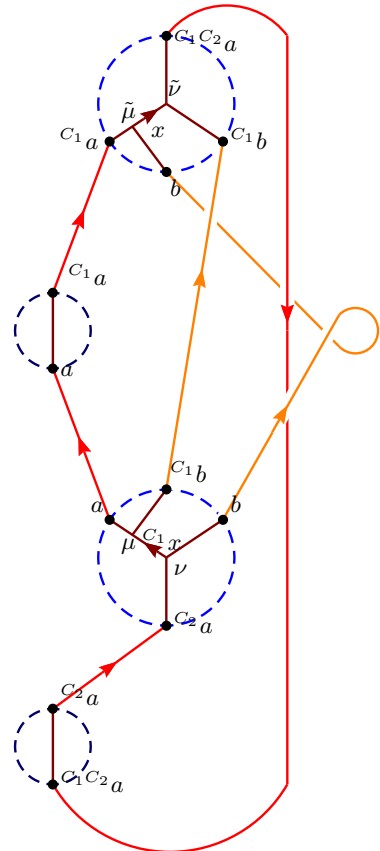

Figure 9: Anyon diagram from the Kirby diagram of $\mathbb{RP}^3 \times S^1$ in Fig. 8. Pay attention to the extra topological twist of the orange line from the correct framing of the corresponding 2-handle.

Assembling all factors as in Eq. (44), we have

$$
\begin{aligned}
\mathcal{Z}\left(\mathbb{RP}^3 \times S^1; C_1, C_2\right) = \frac{1}{D^2} \sum_{\substack{a,b,x,u \\ \mu\nu\tilde{\mu}\tilde{\nu}\rho\sigma\alpha \\ C_1 a = a \\ a \times b \times {}^{C_1}b \to {}^{C_2}a}} d_b \frac{\theta_x}{\theta_a} \left(R_u^{b, C_1 b}\right)_{\rho\sigma} \left(F_{C_2 a}^{a, b, C_1 b}\right)^*_{(x,\tilde{\mu},\tilde{\nu})(u,\sigma,\alpha)} \\
\times \left(F_{C_2 a}^{a, C_1 b, b}\right)_{(C_1 x, \mu, \nu)(u, \rho, \alpha)} \times U_{C_1}^{-1}(a, b; x)_{\tilde{\mu}\mu} U_{C_1}^{-1}(x, {}^{C_1} b; {}^{C_2} a)_{\tilde{\nu}\nu} \\
\times \frac{1}{\eta_b(C_1, C_1)} \frac{\eta_a(C_2, C_1)}{\eta_a(C_1, C_2)} .
\end{aligned}
\tag{53}
$$

It is straightforward to check that this expression is invariant under the vertex basis transformation Eqs. (10),(18) and the symmetry action gauge transformation Eq. (26). The general proof of the cobordism invariance and invertibility of this partition function (see Appendix C) indicates that this expression is $\pm 1$.

Therefore, the two anomaly indicators of $\mathbb{Z}_2 \times \mathbb{Z}_2$ symmetry are $\mathcal{I}_1 = \mathcal{Z}\left(\mathbb{RP}^3 \times S^1; C_1, C_2\right)$ and $\mathcal{I}_2 = \mathcal{Z}\left(\mathbb{RP}^3 \times S^1; C_2, C_1\right)$, given by Eq. (53), and the anomaly $\mathcal{O} \in \mathcal{H}^4(\mathbb{Z}_2 \times \mathbb{Z}_2, U(1))$ can be written as

$$
\mathcal{O} = (\mathcal{I}_1)^{c_1^3 c_2} \cdot (\mathcal{I}_2)^{c_2^3 c_1} ,
\tag{54}
$$

where $c_1$ and $c_2$ are two generators of $\mathcal{H}^1(\mathbb{Z}_2 \times \mathbb{Z}_2, \mathbb{Z}_2)$ corresponding to $C_1$ and $C_2$, respectively.

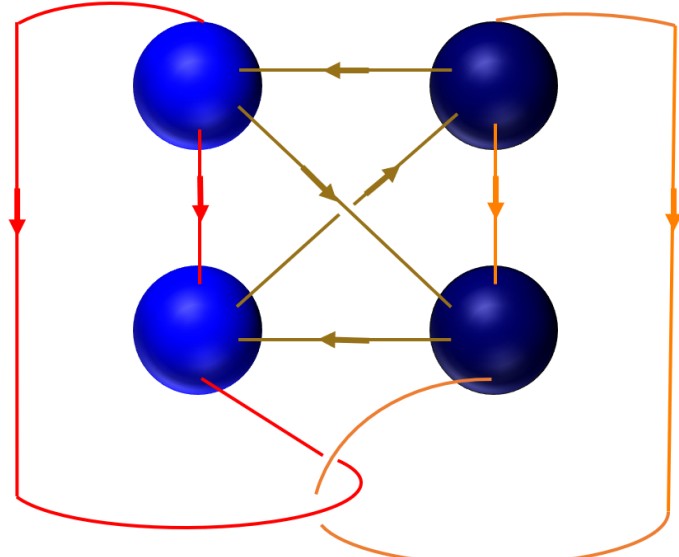

Figure 10: The Kirby diagram of $\mathbb{RP}^2 \times \mathbb{RP}^2$. The blue balls and dark blue balls illustrate two 1-handles and the red, orange and sand-dune lines illustrate three 2-handles. Both 1-handles are nonorientable.

## 4.4 $\mathbb{Z}_2^T \times \mathbb{Z}_2^T$

Finally, let us consider the group $\mathbb{Z}_2^T \times \mathbb{Z}_2^T$. The anomalies of $\mathbb{Z}_2^T \times \mathbb{Z}_2^T$ in $(2+1)$-dimension are classified by $(\mathbb{Z}_2)^4$. Suppose the two anti-unitary generators of $\mathbb{Z}_2^T \times \mathbb{Z}_2^T$ are $\mathcal{T}_1$ and $\mathcal{T}_2$. The representative manifold for the four $\mathbb{Z}_2$ pieces are $\mathbb{CP}^2$, $\mathbb{RP}^4$ with a $\mathcal{T}_1$ defect across the crosscap, $\mathbb{RP}^4$ with a $\mathcal{T}_2$ defect across the crosscap, and $\mathbb{RP}^2 \times \mathbb{RP}^2$ with a $\mathcal{T}_1$ defect across the crosscap of the first $\mathbb{RP}^2$ piece and a $\mathcal{T}_2$ defect across the crosscap of the second $\mathbb{RP}^2$ piece. Given the result Eq. (50), we just need to focus on the last manifold.

The minimum handle decomposition of $\mathbb{RP}^2 \times \mathbb{RP}^2$ contains 1 0-handle, 2 1-handle, 3 2-handle, 2 3-handle and 1 4-handle, as listed in Table 1. The Kirby diagram and the associated anyon diagram are drawn in Figs. 10 and 11, respectively.

Now we put anyon $a$, $b$ and $c$ on a red, orange and sand-dune segment of the 2-handles, respectively, and anyons on other segments can be obtained by symmetry actions on $a$, $b$ and $c$, as shown in Fig. 11. From the two 1-handles we have two constraints $^{\mathcal{T}_1}a \times {}^{\mathcal{T}_2}c \times c \to a$ and $^{\mathcal{T}_1}c \times c \times b \to {}^{\mathcal{T}_2}b$.

The $\eta$-factor from anyon $a$ is given by action $\rho_{\mathcal{T}_1} \circ \rho_{\mathcal{T}_1}$ on $a$, which is $\eta_a(\mathcal{T}_1, \mathcal{T}_1)$. The $\eta$-factor from anyon $b$ is given by action $\rho_{\mathcal{T}_2} \circ \rho_{\mathcal{T}_2}$ on $b$, which is $\eta_b(\mathcal{T}_1, \mathcal{T}_1)$. The $\eta$-factor from anyon $c$ is given by action $\rho_{\mathcal{T}_2} \circ \rho_{\mathcal{T}_1} \circ \rho_{\mathcal{T}_2}^{-1} \circ \rho_{\mathcal{T}_1}^{-1}$ on $c$, which is $\frac{\eta_c(\mathcal{T}_2, \mathcal{T}_1)}{\eta_c(\mathcal{T}_1, \mathcal{T}_2)}$. The $U$-factor from the blue 1-handle is $U_{\mathcal{T}_1}^{-1}({}^{\mathcal{T}_1}a, {}^{\mathcal{T}_2}c; x)_{\mu_x \tilde{\mu}_x} U_{\mathcal{T}_1}^{-1}(x, c; a)_{\nu_x \tilde{\nu}_x}$, and the $U$-factor from the darkblue 1-handle is $U_{\mathcal{T}_2}^{-1}({}^{\mathcal{T}_1}c, y; {}^{\mathcal{T}_2}b)_{\mu_y \tilde{\mu}_y}^* U_{\mathcal{T}_2}^{-1}(c, b; y)_{\nu_y \tilde{\nu}_y}^*$. Finally, we need to evaluate the anyon diagram Fig. 10.

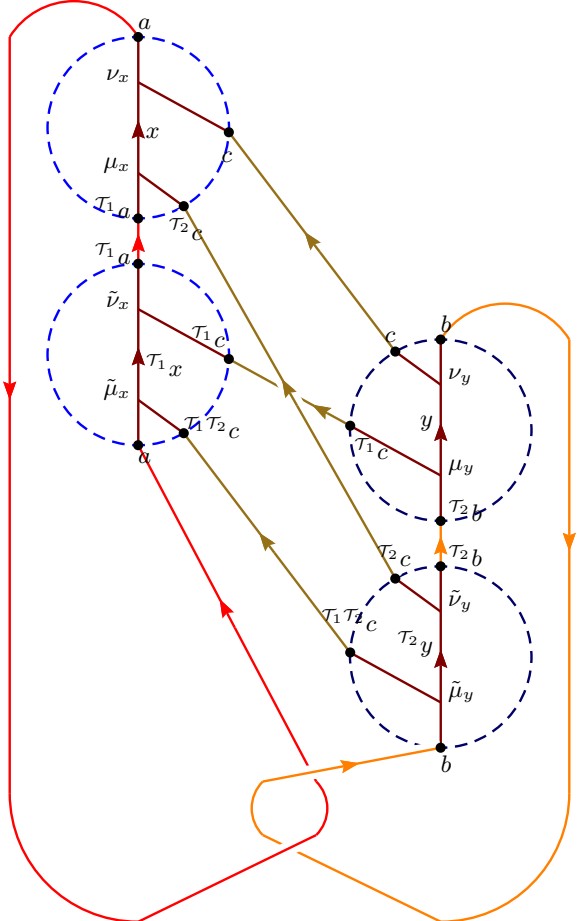

Figure 11: Anyon diagram from the Kirby diagram of $\mathbb{RP}^2 \times \mathbb{RP}^2$ in Fig. 10.

Assembling all factors, we have

$$
\begin{aligned}
\mathcal{Z}\left(\mathbb{RP}^2 \times \mathbb{RP}^2; \mathcal{T}_1, \mathcal{T}_2\right) = \frac{1}{D^3} \sum_{\substack{a,b,c,x,y,u,v \\ \mu_x \nu_x \mu_y \nu_y \tilde{\mu}_x \tilde{\nu}_x \tilde{\mu}_y \tilde{\nu}_y \rho\sigma\tau\alpha\beta\gamma\delta \\ \mathcal{T}_1 a \times \mathcal{T}_2 c \times c \to a \\ \mathcal{T}_1 c \times c \times b \to \mathcal{T}_2 b}} & d_c d_v \frac{\theta_v}{\theta_a \theta_b} \left(R_u^{\mathcal{T}_1 c, \mathcal{T}_2 c}\right)_{\rho\sigma} \\
\times \left(F_v^{a, \mathcal{T}_1 \mathcal{T}_2 c, \mathcal{T}_2 y}\right)_{(\mathcal{T}_1 x, \tilde{\mu}_x, \alpha)(b, \tilde{\mu}_y, \tau)}^* & \left(F_{\mathcal{T}_2 y}^{\mathcal{T}_2 c, \mathcal{T}_1 c, y}\right)_{(u, \rho, \beta)(\mathcal{T}_2 b, \mu_y, \tilde{\nu}_y)}^* \\
\times \left(F_x^{\mathcal{T}_1 x, \mathcal{T}_1 c, \mathcal{T}_2 c}\right)_{(\mathcal{T}_1 a, \tilde{\nu}_x, \mu_x)(u, \sigma, \gamma)}^* & \left(F_v^{\mathcal{T}_1 x, u, y}\right)_{(x, \gamma, \delta)(\mathcal{T}_2 y, \beta, \alpha)}^* \left(F_v^{x, c, b}\right)_{(a, \nu_x, \tau)(y, \nu_y, \delta)}^* \\
\times U_{\mathcal{T}_1}^{-1}(\mathcal{T}_1 a, \mathcal{T}_2 c; x)_{\mu_x \tilde{\mu}_x} & U_{\mathcal{T}_1}^{-1}(x, c; a)_{\nu_x \tilde{\nu}_x} U_{\mathcal{T}_2}^{-1}(\mathcal{T}_1 c, y; \mathcal{T}_2 b)_{\mu_y \tilde{\mu}_y}^* U_{\mathcal{T}_2}^{-1}(c, b; y)_{\nu_y \tilde{\nu}_y}^* \\
\times \eta_a(\mathcal{T}_1, \mathcal{T}_1) \eta_b(\mathcal{T}_2, \mathcal{T}_2) & \frac{\eta_c(\mathcal{T}_2, \mathcal{T}_1)}{\eta_c(\mathcal{T}_1, \mathcal{T}_2)}.
\end{aligned}
\tag{55}
$$

It is straightforward to check that this expression is invariant under the vertex basis transformation Eqs. (10),(18) and the symmetry action gauge transformation Eq. (26). Again, the general proof of the cobordism invariance and invertibility of this partition function (see Appendix C) indicates this expression is $\pm 1$.

Therefore, the four anomaly indicators of $\mathbb{Z}_2^T \times \mathbb{Z}_2^T$ symmetry are $\mathcal{I}_0 = \mathcal{Z}\left(\mathbb{CP}^2\right)$, given by Eq. (46), $\mathcal{I}_1 = \mathcal{Z}\left(\mathbb{RP}^4; \mathcal{T}_1\right)$, $\mathcal{I}_2 = \mathcal{Z}\left(\mathbb{RP}^4; \mathcal{T}_2\right)$, given by Eq. (50), and $\mathcal{I}_3 = \mathcal{Z}\left(\mathbb{RP}^2 \times \mathbb{RP}^2; \mathcal{T}_1, \mathcal{T}_2\right)$, given by Eq. (50). When extracting the cohomology element

from the anomaly indicators, we should be careful that the manifold $\mathbb{RP}^2 \times \mathbb{RP}^2$ has nontrivial $\left(w_2^{TM}\right)^2$ as well. As a result, the anomaly/partition function $\mathcal{O}$ can be written as

$$\mathcal{O} = (\mathcal{I}_0)^{\left(w_2^{TM}\right)^2} \cdot (\mathcal{I}_1)^{t_1{}^4} \cdot (\mathcal{I}_2)^{t_2{}^4} \cdot (\mathcal{I}_0\mathcal{I}_3)^{t_1^2 t_2^2}, \tag{56}$$

where $t_1$ and $t_2$ are two generators of $\mathcal{H}^1(\mathbb{Z}_2^T \times \mathbb{Z}_2^T, \mathbb{Z}_2)$ corresponding to $\mathcal{T}_1$ and $\mathcal{T}_2$, respectively, and $\left(w_2^{TM}\right)^2$ is the generator of the beyond-cohomology piece of anomaly.

### 4.4.1 All-fermion $\mathbb{Z}_2$ topological order

In order to demonstrate the power of the new anomaly indicators, in this subsection we systematically study a concrete example, the all-fermion $\mathbb{Z}_2$ topological order, which is a cousin of the standard $\mathbb{Z}_2$ topological order but all its nontrivial anyons are fermions [63, 84–86]. We will classify all $\mathbb{Z}_2^T \times \mathbb{Z}_2^T$ symmetry fractionalization classes for this topological order, and calculate the anomaly for each class. We will see that the anomalies of some symmetry fractionalization classes can be obtained using (generalizations of) methods developed in the previous literature, but we also point out examples of symmetry fractionalization classes whose anomalies can only be calculated using the anomaly indicators derived here, as far as we can tell.

The data of the underlying UMTC of the all-fermion $\mathbb{Z}_2$ topological order is collected in Ref. [63]. In particular, it has four simple anyons, $1$, $e$, $m$, $\psi = e \times m$. We can label an anyon $a$ by two $\mathbb{Z}_2$ numbers $a = (a_e, a_m)$ as $e^{a_e} \times m^{a_m}$. In a choice of gauge, the $F$-symbols are all trivial and the nontrivial $R$-symbols are given by

$$R^{ee} = R^{mm} = R^{\psi\psi} = R^{\psi e} = R^{m\psi} = R^{em} = (-1). \tag{57}$$

Here we omit the subscript of the $R$-symbol since the outcome of the fusion rules is unique. A $\mathbb{Z}_2^T \times \mathbb{Z}_2^T$ symmetry fractionalization class is specified by the data $\{\rho; U, \eta\}$, which will be classified below.

First we consider the situation where the $\mathbb{Z}_2^T \times \mathbb{Z}_2^T$ symmetry does not permute anyons. In this case, to satisfy Eq. (19) all $U$-symbols can be set to 1. Different symmetry fractionalization classes are then classified by

$$\mathcal{H}^2(\mathbb{Z}_2 \times \mathbb{Z}_2, \mathbb{Z}_2 \times \mathbb{Z}_2) = \mathbb{Z}_2^6. \tag{58}$$

Denoting a representative cocycle of an element in $\mathcal{H}^2(\mathbb{Z}_2 \times \mathbb{Z}_2, \mathbb{Z}_2 \times \mathbb{Z}_2)$ by $\mathbf{t}(\mathbf{g}, \mathbf{h})$ with $\mathbf{g}, \mathbf{h} \in \mathbb{Z}_2^T \times \mathbb{Z}_2^T$, different cohomology elements are distinguished by $\mathbf{t}(\mathcal{T}_1, \mathcal{T}_1)$, $\mathbf{t}(\mathcal{T}_2, \mathcal{T}_2)$, $\mathbf{t}(\mathcal{T}_1\mathcal{T}_2, \mathcal{T}_1\mathcal{T}_2)$. Here we use the gauge convention that $\mathbf{t}(\mathbf{g}, \mathbf{1}) = \mathbf{t}(\mathbf{1}, \mathbf{h}) = 1$, in order to be compatible with the gauge choice Eq. (27). Relatedly, we have

$$\eta_a(\mathcal{T}_1, \mathcal{T}_1) = M_{a, \mathbf{t}(\mathcal{T}_1, \mathcal{T}_1)}, \quad \eta_a(\mathcal{T}_2, \mathcal{T}_2) = M_{a, \mathbf{t}(\mathcal{T}_2, \mathcal{T}_2)}, \quad \eta_a(\mathcal{T}_1\mathcal{T}_2, \mathcal{T}_1\mathcal{T}_2) = M_{a, \mathbf{t}(\mathcal{T}_1\mathcal{T}_2, \mathcal{T}_1\mathcal{T}_2)}. \tag{59}$$

These three $\eta$-phases characterize whether anyon $a$ is a Kramers doublet under $\mathcal{T}_1$, a Kramers doublet under $\mathcal{T}_2$ and charge $1/2$ under $\mathcal{T}_1\mathcal{T}_2$, respectively. In total, there are 36 inequivalent symmetry fractionalization classes in this situation (Of the 64 possible classes associated with $\mathcal{H}^2(\mathbb{Z}_2 \times \mathbb{Z}_2, \mathbb{Z}_2 \times \mathbb{Z}_2) = (\mathbb{Z}_2)^6$, relabeling $e$ and $m$ gives 36 inequivalent classes).

Substituting the UMTC data to the previously derived expressions of $\mathcal{I}_{0,1,2,3}$, the anomaly indicators become

$$\mathcal{I}_0 = \frac{1}{2} \sum_a \theta_a \, ,$$

$$\mathcal{I}_1 = \frac{1}{2} \sum_a \theta_a \eta_a(\mathcal{T}_1, \mathcal{T}_1) \, ,$$

$$\mathcal{I}_2 = \frac{1}{2} \sum_a \theta_a \eta_a(\mathcal{T}_2, \mathcal{T}_2) \, ,$$

$$\mathcal{I}_3 = \frac{1}{8} \sum_{abc} \frac{\theta_{a\times b} \theta_c}{\theta_a \theta_b} \eta_a(\mathcal{T}_1, \mathcal{T}_1) \eta_b(\mathcal{T}_2, \mathcal{T}_2) \frac{\eta_c(\mathcal{T}_2, \mathcal{T}_1)}{\eta_c(\mathcal{T}_1, \mathcal{T}_2)} \, . \tag{60}$$

In particular, $\mathcal{I}_3$ simplifies dramatically in this context.

Following Ref. [37], we make Table. 2 to summarize the anomalies for all of the 36 inequivalent symmetry fractionalization classes. In Table. 2 we use the labeling convention of Ref. [85]: If an excitation carries half charge under the unitary $\mathbb{Z}_2$ symmetry generated by $\mathcal{T}_1 \mathcal{T}_2$, it is followed by a $C$ in the labeling. If it carries Kramers degeneracy under $\mathcal{T}_1$ or $\mathcal{T}_2$, then it is followed by a $\mathcal{T}_1$ or $\mathcal{T}_2$ in the labeling.[12]

From Table 2, we see that, when the symmetry fractionalization class is trivial, i.e., $\eta_a(\mathbf{g}, \mathbf{h}) = 1$ for all anyon $a$ and all group elements $\mathbf{g}, \mathbf{h}$, $\mathcal{I}_0 = \mathcal{I}_1 = \mathcal{I}_2 = \mathcal{I}_3 = -1$, signaling nontrivial anomaly. This is to be contrast to the case of the $\mathbb{Z}_2$ toric code with the trivial symmetry fractionalization class, where $\mathcal{I}_0 = \mathcal{I}_1 = \mathcal{I}_2 = \mathcal{I}_3 = 1$ and no anomaly is present [37].

We mention that this result can also be achieved by considering the projection $p : \mathbb{Z}_2^T \times \mathbb{Z}_2^T \to \mathbb{Z}_2^{T0}$, where $\mathbb{Z}_2^{T0}$ is thought of as an anti-unitary symmetry on $\mathcal{C}$ that does not permute anyons as well. The anomaly indicators of $\mathbb{Z}_2^{T0}$ are already known in previous literature [32, 34] and reproduced in Eqs. (46) and (50). Notice that the trivial symmetry fractionalization class of $\mathbb{Z}_2^T \times \mathbb{Z}_2^T$ denoted by $ef mf$ here is the "pullback" of the trivial symmetry fractionalization class of $\mathbb{Z}_2^{T0}$, denoted by $ef mf$ as well in the literature. The anomaly of $ef mf$ for $\mathbb{Z}_2^T \times \mathbb{Z}_2^T$ is the pullback of the anomaly of $ef mf$ for $\mathbb{Z}_2^{T0}$. From Eqs (46) and (50), the latter anomaly is $(w_2^{TM})^2 + t^4$ where $t$ is the generator of $\mathcal{H}^1(\mathbb{Z}_2^{T0}, \mathbb{Z}_2)$, whose pullback to $\mathbb{Z}_2^T \times \mathbb{Z}_2^T$ is $(w_2^{TM})^2 + t_1^4 + t_2^4$. Comparing this result with Eq. (56), we get the first line of Table 2. Based on the anomaly of this symmetry fractionalization class, the rest of the Table 2 can be achieved from relative anomaly as in Ref. [37].

Next consider the situation where anyons are permuted under some elements of $\mathbb{Z}_2^T \times \mathbb{Z}_2^T$ symmetry. There are two possibilities:

(a) $\mathcal{T}_1$ and $\mathcal{T}_2$ both exchange two of three nontrivial anyons.

(b) $\mathcal{T}_1$ and $\mathcal{T}_1 \mathcal{T}_2$ both exchange two of three nontrivial anyons.

Without loss of generality, we will take the anyons being exchanged as $e$ and $m$.

In either case, if some (unitary or anti-unitary) element $\mathbf{g} \in \mathbb{Z}_2^T \times \mathbb{Z}_2^T$ permutes $e$ and $m$, to satisfy Eq. (19), we can demand that $\rho_{\mathbf{g}}$ action on $|a, b; c\rangle$ be such that

$$U_{\mathbf{g}}(a, b; c) = (-1)^{a_e b_m} \, , \tag{61}$$

with $(a_e, a_m), (b_e, b_m)$ the $\mathbb{Z}_2$ labels of $a, b$. For any element $\mathbf{g}$ that does not permute anyons, we can take $U_{\mathbf{g}}(a, b; c) = 1$. To satisfy Eqs. (23) and (24), a specific valid choice of the $\eta$-symbols is

$$\eta_\psi^{(1)}(\mathbf{g}, \mathbf{g}) = -1 \, , \tag{62}$$

---

[12]$\mathcal{I}_3$ in Table II of Ref. [37] is in fact our $\mathcal{I}_1 \mathcal{I}_2 \mathcal{I}_3$.

Table 2: Anomalies for all-fermion $\mathbb{Z}_2$ topological order with $\mathbb{Z}_2^T \times \mathbb{Z}_2^T$ symmetry, where symmetries do not permute anyons. $ef\,mf$ refers to the trivial symmetry fractionalization class. All classes have $\mathcal{I}_0 = -1$ and hence the beyond-cohomology anomaly.

| Label | $\mathsf{t}(\mathcal{T}_1\mathcal{T}_2, \mathcal{T}_1\mathcal{T}_2), \mathsf{t}(\mathcal{T}_1, \mathcal{T}_1), \mathsf{t}(\mathcal{T}_2, \mathcal{T}_2)$ | $(\mathcal{I}_1, \mathcal{I}_2, \mathcal{I}_0\mathcal{I}_3)$ |
|---|---|---|
| $ef\,mf$ | $(1,1,1)$ | $(-1,-1,1)$ |
| $ef\,mf\,\mathcal{T}_2$ | $(1,1,m)$ | $(-1,1,-1)$ |
| $ef\,\mathcal{T}_2 mf\,\mathcal{T}_2$ | $(1,1,\psi)$ | $(-1,1,-1)$ |
| $ef\,\mathcal{T}_1 mf$ | $(1,m,1)$ | $(1,-1,-1)$ |
| $ef\,\mathcal{T}_1 mf\,\mathcal{T}_2$ | $(1,m,e)$ | $(1,1,1)$ |
| $ef\,\mathcal{T}_1\mathcal{T}_2 mf$ | $(1,m,m)$ | $(1,1,1)$ |
| $ef\,\mathcal{T}_1\mathcal{T}_2 mf\,\mathcal{T}_2$ | $(1,m,\psi)$ | $(1,1,1)$ |
| $ef\,\mathcal{T}_1 mf\,\mathcal{T}_1$ | $(1,\psi,1)$ | $(1,-1,-1)$ |
| $ef\,\mathcal{T}_1\mathcal{T}_2 mf\,\mathcal{T}_1$ | $(1,\psi,m)$ | $(1,1,1)$ |
| $ef\,\mathcal{T}_1\mathcal{T}_2 mf\,\mathcal{T}_1\mathcal{T}_2$ | $(1,\psi,\psi)$ | $(1,1,1)$ |
| $ef\,mf\,C$ | $(e,1,1)$ | $(-1,-1,-1)$ |
| $ef\,mf\,C\mathcal{T}_2$ | $(e,1,e)$ | $(-1,1,1)$ |
| $ef\,\mathcal{T}_2 mf\,C$ | $(e,1,m)$ | $(-1,1,-1)$ |
| $ef\,\mathcal{T}_2 mf\,C\mathcal{T}_2$ | $(e,1,\psi)$ | $(-1,1,-1)$ |
| $ef\,mf\,C\mathcal{T}_1$ | $(e,e,1)$ | $(1,-1,1)$ |
| $ef\,mf\,C\mathcal{T}_1\mathcal{T}_2$ | $(e,e,e)$ | $(1,1,-1)$ |
| $ef\,\mathcal{T}_2 mf\,C\mathcal{T}_1$ | $(e,e,m)$ | $(1,1,1)$ |
| $ef\,\mathcal{T}_2 mf\,C\mathcal{T}_1\mathcal{T}_2$ | $(e,e,\psi)$ | $(1,1,1)$ |
| $ef\,\mathcal{T}_1 mf\,C$ | $(e,m,1)$ | $(1,-1,-1)$ |
| $ef\,\mathcal{T}_1 mf\,C\mathcal{T}_2$ | $(e,m,e)$ | $(1,1,1)$ |
| $ef\,\mathcal{T}_1\mathcal{T}_2 mf\,C$ | $(e,m,m)$ | $(1,1,-1)$ |
| $ef\,\mathcal{T}_1\mathcal{T}_2 mf\,C\mathcal{T}_2$ | $(e,m,\psi)$ | $(1,1,-1)$ |
| $ef\,\mathcal{T}_1 mf\,C\mathcal{T}_1$ | $(e,\psi,1)$ | $(1,-1,-1)$ |
| $ef\,\mathcal{T}_1 mf\,C\mathcal{T}_1\mathcal{T}_2$ | $(e,\psi,e)$ | $(1,1,1)$ |
| $ef\,\mathcal{T}_1\mathcal{T}_2 mf\,C\mathcal{T}_1$ | $(e,\psi,m)$ | $(1,1,-1)$ |
| $ef\,\mathcal{T}_1\mathcal{T}_2 mf\,C\mathcal{T}_1\mathcal{T}_2$ | $(e,\psi,\psi)$ | $(1,1,-1)$ |
| $ef\,Cmf\,C$ | $(\psi,1,1)$ | $(-1,-1,-1)$ |
| $ef\,C\mathcal{T}_2 mf\,C$ | $(\psi,1,m)$ | $(-1,1,-1)$ |
| $ef\,C\mathcal{T}_2 mf\,C\mathcal{T}_2$ | $(\psi,1,\psi)$ | $(-1,1,1)$ |
| $ef\,Cmf\,C\mathcal{T}_1$ | $(\psi,e,1)$ | $(1,-1,-1)$ |
| $ef\,Cmf\,C\mathcal{T}_1\mathcal{T}_2$ | $(\psi,e,e)$ | $(1,1,-1)$ |
| $ef\,C\mathcal{T}_2 mf\,C\mathcal{T}_1$ | $(\psi,e,m)$ | $(1,1,-1)$ |
| $ef\,C\mathcal{T}_2 mf\,C\mathcal{T}_1\mathcal{T}_2$ | $(\psi,e,\psi)$ | $(1,1,1)$ |
| $ef\,C\mathcal{T}_1 mf\,CT_1$ | $(\psi,\psi,1)$ | $(1,-1,1)$ |
| $ef\,C\mathcal{T}_1\mathcal{T}_2 mf\,C\mathcal{T}_1$ | $(\psi,\psi,m)$ | $(1,1,1)$ |
| $ef\,C\mathcal{T}_1\mathcal{T}_2 mf\,C\mathcal{T}_1\mathcal{T}_2$ | $(\psi,\psi,\psi)$ | $(1,1,-1)$ |

where $\mathbf{g}$ is an element that permutes anyons, while all other $\eta$-symbols (such as $\eta_e^{(1)}(\mathbf{g},\mathbf{g})$ and $\eta_\psi^{(1)}(\mathbf{g},\mathbf{g}')$ with $\mathbf{g}' \neq \mathbf{g}$) are 1. To get all possible valid choices of the $\eta$-symbols, note that $\mathcal{H}_\rho^2(\mathbb{Z}_2 \times \mathbb{Z}_2, \mathbb{Z}_2 \times \mathbb{Z}_2) \cong \mathbb{Z}_2$ for both case (a) and case (b), which means that in either case there is one more symmetry fractionalization class. Denoting the nontrivial element in $\mathcal{H}_\rho^2(\mathbb{Z}_2 \times \mathbb{Z}_2, \mathbb{Z}_2 \times \mathbb{Z}_2) \cong \mathbb{Z}_2$ by $\mathsf{t}(\mathbf{g}, \mathbf{h})$ with $\mathbf{g}, \mathbf{h} \in \mathbb{Z}_2^T \times \mathbb{Z}_2^T$, the other valid choice of the $\eta$-symbols is related to the one above via Eq. (28), i.e., $\eta_a^{(2)}(\mathbf{g}, \mathbf{h}) = \eta_a^{(1)}(\mathbf{g}, \mathbf{h}) M_{a, \mathsf{t}(\mathbf{g}, \mathbf{h})}$. Under the

Table 3: Anomalies for all-fermion $\mathbb{Z}_2$ topological order with $\mathbb{Z}_2^T \times \mathbb{Z}_2^T$ symmetry, where $\mathcal{T}_1$ and $\mathcal{T}_2$ permute anyons, which is the reason for the subscripts for $e$ and $m$. The meanings of the other symbols are the same as in Table 2. All classes have $\mathcal{I}_0 = -1$ and hence the beyond-cohomology anomaly.

| Label | $\mathbf{t}(\mathcal{T}_1\mathcal{T}_2, \mathcal{T}_1\mathcal{T}_2), \mathbf{t}(\mathcal{T}_1, \mathcal{T}_1), \mathbf{t}(\mathcal{T}_2, \mathcal{T}_2)$ | $(\mathcal{I}_1, \mathcal{I}_2, \mathcal{I}_0\mathcal{I}_3)$ |
|---|---|---|
| $(ef\,mf)_{\mathcal{T}_1,\mathcal{T}_2}\psi f\,\mathcal{T}_1\mathcal{T}_2$ | $(1,1,1)$ | $(1,1,1)$ |
| $(ef\,Cmf\,C)_{\mathcal{T}_1,\mathcal{T}_2}\psi f\,\mathcal{T}_1\mathcal{T}_2$ | $(\psi,1,1)$ | $(1,1,-1)$ |

Table 4: Anomalies for all-fermion $\mathbb{Z}_2$ topological order with $\mathbb{Z}_2^T \times \mathbb{Z}_2^T$ symmetry, where $\mathcal{T}_1$ and $\mathcal{T}_1\mathcal{T}_2$ permute anyons, which is the reason for the subscripts for $e$ and $m$. The meanings of the other symbols are the same as in Table 2. All classes have $\mathcal{I}_0 = -1$ and hence the beyond-cohomology anomaly.

| Label | $\mathbf{t}(\mathcal{T}_1\mathcal{T}_2, \mathcal{T}_1\mathcal{T}_2), \mathbf{t}(\mathcal{T}_1, \mathcal{T}_1), \mathbf{t}(\mathcal{T}_2, \mathcal{T}_2)$ | $(\mathcal{I}_1, \mathcal{I}_2, \mathcal{I}_0\mathcal{I}_3)$ |
|---|---|---|
| $(ef\,mf)_{\mathcal{T}_1,\mathcal{T}_1\mathcal{T}_2}\psi f\,\mathcal{T}_1 C$ | $(1,1,1)$ | $(1,-1,-1)$ |
| $(ef\,\mathcal{T}_2mf\,\mathcal{T}_2)_{\mathcal{T}_1,\mathcal{T}_1\mathcal{T}_2}\psi f\,\mathcal{T}_1 C$ | $(1,1,\psi)$ | $(1,1,1)$ |

gauge choice $\mathbf{t}(\mathbf{1},\mathbf{g}) = \mathbf{t}(\mathbf{h},\mathbf{1}) = 1$, in both cases (a) and (b) $\mathbf{t}(\mathbf{g},\mathbf{h})$ is fully characterized by $\mathbf{t}(\mathbf{g},\mathbf{g})$ where $\mathbf{g}$ is the nontrivial group element that does not permute anyons. Now we discuss the two cases separately in detail.

(a) When $\mathcal{T}_1$ and $\mathcal{T}_2$ exchange $e$ and $m$, the representative cocycle $\mathbf{t}$ of the nontrivial element in $\mathcal{H}_\rho^2(\mathbb{Z}_2 \times \mathbb{Z}_2, \mathbb{Z}_2 \times \mathbb{Z}_2) \cong \mathbb{Z}_2$ can be chosen as

$$\mathbf{t}(\mathcal{T}_1\mathcal{T}_2, \mathcal{T}_1\mathcal{T}_2) = \psi, \quad \mathbf{t}(\mathcal{T}_1, \mathcal{T}_1) = \mathbf{t}(\mathcal{T}_2, \mathcal{T}_2) = 1. \tag{63}$$

The physical meaning of these symmetry fractionalization classes is as follows. In both classes $\psi$ is a Kramers doublet under both $\mathcal{T}_1$ and $\mathcal{T}_2$, and both $e$ and $m$ carry integer charge (half charge) under $\mathcal{T}_1\mathcal{T}_2$ in the class characterized by $\eta^{(1)}$ ($\eta^{(2)}$). So we denote the classes $\eta^{(1)}$ and $\eta^{(2)}$ by $(ef\,mf)_{\mathcal{T}_1,\mathcal{T}_2}\psi f\,\mathcal{T}_1\mathcal{T}_2$ and $(ef\,Cmf\,C)_{\mathcal{T}_1,\mathcal{T}_2}\psi f\,\mathcal{T}_1\mathcal{T}_2$, respectively. We see that $(\mathcal{I}_0, \mathcal{I}_1, \mathcal{I}_2, \mathcal{I}_3) = (-1, 1, 1, -1)$ and $(\mathcal{I}_0, \mathcal{I}_1, \mathcal{I}_2, \mathcal{I}_3) = (-1, 1, 1, 1)$ for $(ef\,mf)_{\mathcal{T}_1,\mathcal{T}_2}\psi f\,\mathcal{T}_1\mathcal{T}_2$ and $(ef\,Cmf\,C)_{\mathcal{T}_1,\mathcal{T}_2}\psi f\,\mathcal{T}_1\mathcal{T}_2$, respectively, as summarized in Table 3.

We mention that this result can also be achieved by considering the projection $p : \mathbb{Z}_2^T \times \mathbb{Z}_2^T \to \mathbb{Z}_2^{T0}$, where $\mathbb{Z}_2^{T0}$ is now an anti-unitary symmetry on $\mathcal{C}$ that permutes $e$ and $m$. Notice that $(ef\,mf)_{\mathcal{T}_1,\mathcal{T}_2}\psi f\,\mathcal{T}_1\mathcal{T}_2$ class of $\mathbb{Z}_2^T \times \mathbb{Z}_2^T$ symmetry is the pullback of $(ef\,mf)_{\mathcal{T}}\psi f\,\mathcal{T}$ of $\mathbb{Z}_2^{T0}$ symmetry (the meaning of this notation is similar to others), hence the anomaly of $(ef\,mf)_{\mathcal{T}_1,\mathcal{T}_2}\psi f\,\mathcal{T}_1\mathcal{T}_2$ for $\mathbb{Z}_2^T \times \mathbb{Z}_2^T$ is the pullback of the anomaly of $(ef\,mf)_{\mathcal{T}}\psi f\,\mathcal{T}$ for $\mathbb{Z}_2^{T0}$. From Eqs. (46) and (50), the latter anomaly is just $(w_2^{TM})^2$ hence the former anomaly is $(w_2^{TM})^2$ as well. Comparing this result with Eq. (56), we get the first line of Table 3. Based on the anomaly of this symmetry fractionalization class, the second line of Table 2 can be achieved from relative anomaly as in Ref. [37].

(b) When $\mathcal{T}_1$ and $\mathcal{T}_1\mathcal{T}_2$ exchange $e$ and $m$, the representative cocycle $\mathbf{t}$ of the nontrivial element in $\mathcal{H}_\rho^2(\mathbb{Z}_2 \times \mathbb{Z}_2, \mathbb{Z}_2 \times \mathbb{Z}_2) \cong \mathbb{Z}_2$ can be chosen as

$$\mathbf{t}(\mathcal{T}_2, \mathcal{T}_2) = \psi, \quad \mathbf{t}(\mathcal{T}_1, \mathcal{T}_1) = \mathbf{t}(\mathcal{T}_1\mathcal{T}_2, \mathcal{T}_1\mathcal{T}_2) = 1. \tag{64}$$

The physical meaning of these symmetry fractionalization classes is as follows. In both classes $\psi$ is a Kramers doublet under $\mathcal{T}_1$ and carries half charge under $\mathcal{T}_1\mathcal{T}_2$, and both $e$ and $m$ are Kramers singlets (doublets) under $\mathcal{T}_2$ in the class characterized by $\eta^{(1)}$ ($\eta^{(2)}$).

So we denote the classes $\eta^{(1)}$ and $\eta^{(2)}$ by $(ef\,mf)_{\mathcal{T}_1,\mathcal{T}_1\mathcal{T}_2}\psi f\,\mathcal{T}_1 C$ and $(ef\,\mathcal{T}_2 mf\,\mathcal{T}_2)_{\mathcal{T}_1,\mathcal{T}_1\mathcal{T}_2}\psi f\,\mathcal{T}_1 C$, respectively. We see that $(\mathcal{I}_0,\mathcal{I}_1,\mathcal{I}_2,\mathcal{I}_3) = (-1,1,-1,1)$ and $(\mathcal{I}_0,\mathcal{I}_1,\mathcal{I}_2,\mathcal{I}_3) = (-1,1,1,-1)$ for $(ef\,mf)_{\mathcal{T}_1,\mathcal{T}_1\mathcal{T}_2}\psi f\,\mathcal{T}_1 C$ and $(ef\,\mathcal{T}_2 mf\,\mathcal{T}_2)_{\mathcal{T}_1,\mathcal{T}_1\mathcal{T}_2}\psi f\,\mathcal{T}_1 C$, respectively, as summarized in Table 4.

For this particular case, because the unitary symmetry $\mathcal{T}_1\mathcal{T}_2$ exchanges $e$ and $m$, we are aware of no other method to get the anomaly besides the complete knowledge of anomaly indicators of $\mathbb{Z}_2^T \times \mathbb{Z}_2^T$.

# 5 Generalization to connected Lie group symmetry

We believe that the construction and recipe presented in Sec. 3 can be generalized to arbitrary group $G$. Comparing general group symmetry and finite group symmetry, what concerns us the most in the calculation of the partition function $\mathcal{Z}(\mathcal{M},\mathcal{G})$ is how to write down the $U$-factors and $\eta$-factors. Manifestly, given a $G$-bundle $\mathcal{G}$, there is an associated map $f : \mathcal{M} \to BG$, with $BG$ the classifying space of $G$. In particular, the map $f$ maps a 1-chain of $\mathcal{M}$ to a 1-chain of $BG$, and then assigns an element $\rho_{\mathbf{g}}$ to this 1-chain of $\mathcal{M}$, which in turn gives the desired $U$-factors. Moreover, the map $f$ maps a 2-chain of $\mathcal{M}$ to a 2-chain of $BG$, and then assigns an element $\rho_{\mathbf{g}} \circ \rho_{\mathbf{h}} \circ \dots$ to this 2-chain of $\mathcal{M}$, which gives the desired $\eta$-factors. This serves as a formal construction of the partition function of the TQFT with a general symmetry $G$. However, such a construction seems to be dependent on a specific choice of $f$ and $BG$. We believe that the partition function ultimately only depends on the $G$-bundle itself (but not the specific choice of $f$ and $BG$), although we are unable to prove it using arguments similar to what we present in Appendix C. Moreover, this construction is hard to work with for a general group $G$. Fortunately, for a connected Lie group $G$, there is a more operational method to write down the $\eta$-factors and eventually calculate $\mathcal{Z}(\mathcal{M},\mathcal{G})$. We discuss it in this section.

Specifically, for a connected Lie group $G$, to calculate the partition function $\mathcal{Z}(\mathcal{M},\mathcal{G})$ of the manifold $\mathcal{M}$ with a $G$-bundle structure $\mathcal{G}$ on $\mathcal{M}$, we can still start with a handle decomposition of the manifold $\mathcal{M}$. Since now $G$ cannot permute anyons, we can associate a single anyon $a$ to the $S^1$ boundary of each 2-handle. Moreover, no $U$-factors are involved. Now, given the prescribed labels, we need to calculate the correct $\eta$-factor, evaluate the anyon diagram from the Kirby diagram $\langle K \rangle$ of $\mathcal{M}$, and assemble the result in a way similar to Eq. (44):

$$\mathcal{Z}(\mathcal{M},\mathcal{G}) = D^{-\chi+2(N_4-N_3)}$$

$$\times \sum_{\text{labels}} \left( \frac{\displaystyle\prod_{\text{2 handle } i} d_{a_i}}{\displaystyle\prod_{\text{1 handle } x} \left( \prod_{\text{2 handle } j \text{ across } x} d_{a_j} \right)^{1/2}} \times \left( \prod_i (\eta\text{-factors})_i \right) \times \langle K \rangle \right). \quad (65)$$

Here $N_k$ is the number of $k$-handles of this handle decomposition, and $\chi \equiv N_0 - N_1 + N_2 - N_3 + N_4$ is the Euler number of $\mathcal{M}$.

The only nontrivial part compared with previous examples of finite group symmetry is the calculation of the $\eta$-factor for a 2-handle. In the presence of a connected Lie group symmetry $G$, we have the following prescription. For every 2-handle, there is an associated 2-chain $[h]$. The map $f : \mathcal{M} \to BG$ associated to the $G$-bundle $\mathcal{G}$ then gives $f_*[h]$, which is a 2-chain in $BG$. The symmetry fractionalization class is characterized by an element $\mathbf{w} \in \mathcal{H}^2(G,\mathcal{A}) \cong \mathcal{H}^2(BG,\mathcal{A})$, and pairing it with $f_*[h]$ gives an anyon $\mathbf{w}(f_*[h]) \in \mathcal{A}$. If we associate an anyon $a$ to the $S^1$ boundary of a 2-handle, the $\eta$-factor of this 2-handle is $M_{a,\mathbf{w}(f_*[h])}$, i.e., the double braid between $a$ and $\mathbf{w}(f_*[h])$. Intuitively, such a phase can be

viewed as the phase the anyon $a$ experiences when traveling along the $S^1$ boundary, given the nontrivial background $G$-bundle structure $\mathcal{G}$. Therefore, it can be written down in terms of the charge of $a$.

To illustrate this recipe regarding a connected Lie group symmetry $G$, now we go to the example of $SO(N)$. We will see that this recipe gives the correct partition function on manifolds with an $SO(N)$-bundle structure, and eventually provides us with the anomaly indicators, together with the $SO(N)$ Hall conductance.

## 5.1 Example: $SO(N)$

The relevant bordism group for symmetry group $SO(N)$ is [87]

$$
\Omega_4^{SO}(BSO(N)) = \begin{cases} (\mathbb{Z})^2, & N = 2, 3, \\ (\mathbb{Z})^3, & N = 4, \\ (\mathbb{Z})^2 \oplus \mathbb{Z}_2, & N \geqslant 5. \end{cases} \tag{66}
$$

The first generating manifold is $\mathbb{CP}^2$ with a trivial $SO(N)$ bundle. The second generating manifold is $\mathbb{CP}^2$ with a nontrivial $SO(N)$ bundle such that the associated map $f_1 : \mathbb{CP}^2 \to BSO(N)$ is given by

$$
f_1 : \quad \mathbb{CP}^2 \subset \mathbb{CP}^\infty \cong BU(1) \xrightarrow{B\tilde{f}_1} BSO(N), \tag{67}
$$

and $\tilde{f}_1 : U(1) \to SO(N)$ is

$$
e^{i\theta} \to \begin{pmatrix} \cos(\theta) & \sin(\theta) & \\ -\sin(\theta) & \cos(\theta) & \\ & & \mathrm{diag}(1, 1, \dots) \end{pmatrix}. \tag{68}
$$

Its associated vector bundle is simply the tautological line bundle of $\mathbb{CP}^2$ [88] (together with an $(N-2)$-dimensional trivial bundle), and thus we denote it by $\mathcal{A}_t$. When $N \geqslant 4$, there exists a third generating manifold, which is $\mathbb{CP}^2$ with another nontrivial $SO(N)$ bundle such that the associated map $f_2 : \mathbb{CP}^2 \to BSO(N)$ is given by

$$
f_2 : \quad \mathbb{CP}^2 \subset \mathbb{CP}^\infty \cong BU(1) \xrightarrow{B\tilde{f}_2} BSO(N), \tag{69}
$$

and $\tilde{f}_2 : U(1) \to SO(N)$ is

$$
e^{i\theta} \to \begin{pmatrix} \cos(\theta) & \sin(\theta) & & & \\ -\sin(\theta) & \cos(\theta) & & & \\ & & \cos(\theta) & \sin(\theta) & \\ & & -\sin(\theta) & \cos(\theta) & \\ & & & & \mathrm{diag}(1, \dots) \end{pmatrix}. \tag{70}
$$

Its associated vector bundle is the direct sum of two tautological line bundles of $\mathbb{CP}^2$ (together with an $(N-4)$-dimensional trivial bundle), and thus we denote it by $\mathcal{A}_t^{\oplus 2}$.

The partition function corresponds to an element in

$$
\Omega_{SO}^4(BSO(N), U(1)) \equiv \mathrm{Hom}(\Omega_4^{SO}(BSO(N)), U(1)) = \begin{cases} (U(1))^2, & N = 2, 3, \\ (U(1))^3, & N = 4, \\ (U(1))^2 \oplus \mathbb{Z}_2, & N \geqslant 5. \end{cases} \tag{71}
$$

Similar to the case in Sec. 4.1, because all elements corresponding to $U(1)$ are smoothly connected to the trivial element, the 't Hooft anomaly is absent when $N = 2, 3, 4$ and classified by

$\mathbb{Z}_2$ when $N \geqslant 5$. Still, the partition function is not completely trivial even for $N = 2, 3, 4$. To evaluate the partition function on an oriented 4-dimensional manifold with an $SO(N)$-bundle structure, we just need to evaluate it on the generating manifolds.

Since the underlying manifold is always $\mathbb{CP}^2$, compared with the calculation that leads to Eq. (46), the calculation of the partition function of $\mathbb{CP}^2$ with nontrivial bundle $\mathcal{A}_t$ or $\mathcal{A}_t^{\oplus 2}$ just requires us to add an appropriate $\eta$-factor for the 2-handle.

First consider $\mathcal{A}_t$. The extra $\eta$-factor can be seen as follows. The 2-handle $[h]$ here is the generator of $H_2(\mathbb{CP}^2, \mathbb{Z})$, the pushforward of which under $f_1$, i.e., $f_{1*}[h]$, gives the generator of $H_2(BSO(N), \mathbb{Z})$. According to the recipe, given the anyon label $a$ in Eq. (45), the correct $\eta$-factor should be $M_{a,\mathbf{w}([f_{1*}[h]])}$. Since $f_{1*}[h]$ is the generator of $H_2(BSO(N), \mathbb{Z})$, physically this phase factor is related to the $SO(N)$ charge $q_a$ of anyon $a$ by $e^{i2\pi q_a}$, where for $N = 2$ $q_a \in [0, 1)$ is the (fractional) $SO(2) \cong U(1)$ charge of $a$, and for $N \geqslant 3$ $q_a \in \{0, \frac{1}{2}\}$ labels whether $a$ carries linear ($q_a = 0$) or spinor ($q_a = \frac{1}{2}$) representation under $SO(N)$. Consequently, we have

$$\mathcal{Z}_{SO(N)}(\mathbb{CP}^2; \mathcal{A}_t) = \frac{1}{D} \sum_a d_a^2 \theta_a e^{i2\pi q_a}. \tag{72}$$

Secondly, for $\mathcal{A}_t^{\oplus 2}$, there is no extra $\eta$-factor. This is simply because $\tilde{f}_2$ defines a trivial element in $\pi_1(SO(N)) \cong \mathbb{Z}_2$, which suggests that $\mathcal{A}_t^{\oplus 2}$ can be constructed from attaching a 4-handle to lower handlebody with a trivial $SO(N)$ bundle on it. Therefore, the partition function of $\mathbb{CP}^2$ with the $SO(N)$ bundle $\mathcal{A}_t^{\oplus 2}$ is identical to the partition function of $\mathbb{CP}^2$ with a trivial $SO(N)$ bundle, i.e.,

$$\mathcal{Z}_{SO(N), N \geqslant 4}(\mathbb{CP}^2; \mathcal{A}_t^{\oplus 2}) = \frac{1}{D} \sum_a d_a^2 \theta_a. \tag{73}$$

When $N = 2, 3, 4$, even though there is no nontrivial 't Hooft anomaly, the partition function is still nontrivial and can be written down in terms of various theta terms. In particular, when $N = 2$, we have

$$\mathcal{Z}_{SO(2)}(\mathcal{M}; \mathcal{A}) = \mathcal{Z}(\mathbb{CP}^2)^{\sigma(\mathcal{M})} \cdot \left( \frac{\mathcal{Z}\left(\mathbb{CP}^2; \mathcal{A}_t\right)}{\mathcal{Z}(\mathbb{CP}^2)} \right)^{\left( C_1^{SO(2)} \right)^2}, \tag{74}$$

where $\sigma(\mathcal{M})$ is the intersection number of $\mathcal{M}$ and $C_1^{SO(2)} \in \mathcal{H}^2(SO(2), \mathbb{Z})$ is the first Chern class of $SO(2)$. When $N = 3$, we have

$$\mathcal{Z}_{SO(3)}(\mathcal{M}; \mathcal{A}) = \mathcal{Z}(\mathbb{CP}^2)^{\sigma(\mathcal{M})} \cdot \left( \frac{\mathcal{Z}\left(\mathbb{CP}^2; \mathcal{A}_t\right)}{\mathcal{Z}(\mathbb{CP}^2)} \right)^{p_1^{SO(3)}}, \tag{75}$$

where $p_1^{SO(3)} \in \mathcal{H}^4(SO(3), \mathbb{Z})$ is the first Pontryagin class of $SO(3)$. When $N = 4$, when writing down the term corresponding to the Euler class $e^{SO(4)}$, pay attention that given Eq. (69), the pullback of Pontryagin class $p_1^{SO(4)}$ should be twice the generator of $\mathcal{H}^4(\mathbb{CP}^2, \mathbb{Z})$. Therefore, we have

$$\mathcal{Z}_{SO(4)}(\mathcal{M}; \mathcal{A}) = \mathcal{Z}(\mathbb{CP}^2)^{\sigma(\mathcal{M})} \cdot \left( \frac{\mathcal{Z}\left(\mathbb{CP}^2; \mathcal{A}_t\right)}{\mathcal{Z}(\mathbb{CP}^2)} \right)^{p_1^{SO(4)}} \cdot \left( \left( \frac{\mathcal{Z}\left(\mathbb{CP}^2\right)}{\mathcal{Z}(\mathbb{CP}^2; \mathcal{A}_t)} \right)^2 \right)^{e^{SO(4)}}, \tag{76}$$

where $p_1^{SO(4)} \in \mathcal{H}^4(SO(4), \mathbb{Z})$ is the first Pontryagin class of $SO(4)$, and $e^{SO(4)} \in \mathcal{H}^4(SO(4), \mathbb{Z})$ is the Euler class of $SO(4)$. When $N \geqslant 5$, similarly we have

$$\mathcal{Z}_{SO(N), N \geqslant 5}(\mathcal{M}; \mathcal{A}) = \mathcal{Z}(\mathbb{CP}^2)^{\sigma(\mathcal{M})} \cdot \left( \frac{\mathcal{Z}\left(\mathbb{CP}^2; \mathcal{A}_t\right)}{\mathcal{Z}(\mathbb{CP}^2)} \right)^{p_1^{SO(N)}} \cdot \left( \left( \frac{\mathcal{Z}\left(\mathbb{CP}^2\right)}{\mathcal{Z}(\mathbb{CP}^2; \mathcal{A}_t)} \right)^2 \right)^{w_4^{SO(N)}}, \tag{77}$$

where $p_1^{SO(N)} \in \mathcal{H}^4(SO(N), \mathbb{Z})$ is the first Pontryagin class of $SO(N)$, and $w_4^{SO(N)} \in \mathcal{H}^4(SO(N), \mathbb{Z}_2)$ is the fourth Stiefel-Whitney class of $SO(N)$.

### 5.1.1 Anomaly indicator for $N \geqslant 5$

As discussed before, there is no nontrivial $SO(N)$ anomaly if $N < 5$. When $N \geqslant 5$, the $SO(N)$ anomalies are classified by $\mathbb{Z}_2$, whose anomaly indicator is given by

$$\mathcal{I} = \left( \frac{\mathcal{Z}(\mathbb{CP}^2)}{\mathcal{Z}(\mathbb{CP}^2; \mathcal{A}_t)} \right)^2 = \left( \frac{\sum_a d_a^2 \theta_a}{\sum_b d_b^2 \theta_b e^{i2\pi q_b}} \right)^2 . \tag{78}$$

As before, the general proof of the cobordism invariance and invertibility of this partition function indicates that this expression is $\pm 1$. The anomaly $\mathcal{O}$ can be written as

$$\mathcal{O} = (\mathcal{I})^{w_4^{SO(N)}} , \tag{79}$$

where $w_4^{SO(N)}$ is the fourth Stiefel-Whitney class belonging to $\mathcal{H}^4(SO(N), \mathbb{Z}_2)$.

### 5.1.2 $SO(N)$ Hall conductance

Besides giving the anomaly indicator, the above partition functions also encode various Hall conductance in a topological order with an $SO(N)$ symmetry. First, as discussed in Sec. 4.1, they reproduce the thermal Hall conductance from the chiral central charge (up to contributions from $(2+1)D$ invertible states). Moreover, they also yield the $SO(N)$ Hall conductance. Concretely, let us consider threading a $2\pi$ $SO(N)$ flux into the system, which breaks the $SO(N)$ symmetry to $SO(2) \times SO(N-2)$. The Hall conductance measures the charge under $SO(2) \times SO(N-2)$ that this flux attracts. For $N > 4$, the charge under $SO(N-2)$ means the representation under $SO(N-2)$. For $N = 4$, the flux breaks the $SO(4)$ symmetry to $SO(2) \times SO(2)'$, and it can attract charge under either $SO(2)$ or $SO(2)'$. We will use the unit where $\hbar = 1$ and the elementary charge of local excitations in the system under $SO(2)$ (and also under $SO(2)'$ when $N = 4$) is 1.

Let us first consider the amount of $SO(2)$ charge being attracted, denoted by $\sigma_{xy}$. We start with the case where $N = 2$. In the partition function, this can be read off from the factor $\left( \frac{\mathcal{Z}(\mathbb{CP}^2; \mathcal{A}_t)}{\mathcal{Z}(\mathbb{CP}^2)} \right)^{\left( C_1^{SO(2)} \right)^2}$. Denoting $e^{i\Theta} = \frac{\mathcal{Z}(\mathbb{CP}^2; \mathcal{A}_t)}{\mathcal{Z}(\mathbb{CP}^2)}$, and using $\left( C_1^{SO(2)} \right)^2 = \frac{1}{4\pi^2} dA \wedge dA$ where $A$ is the $SO(2)$ gauge field, this factor can be written as $e^{i \frac{\Theta}{\pi} \cdot \frac{1}{4\pi} d(A \wedge dA)}$. The standard argument (see, e.g., Ref. [84]) then shows that

$$\sigma_{xy} = \frac{\Theta}{\pi}, \quad e^{i\Theta} \equiv \frac{\mathcal{Z}(\mathbb{CP}^2; \mathcal{A}_t)}{\mathcal{Z}(\mathbb{CP}^2)} = \frac{\sum_a d_a^2 \theta_a e^{i2\pi q_a}}{\sum_b d_b^2 \theta_b} . \tag{80}$$

Notice that this formula only captures the "fractional" part of the Hall conductance, and there can be extra contributions to the Hall conductance from a $(2+1)D$ invertible state, which are integral multiples of 2.

For $N > 2$, $\sigma_{xy}$ can be extracted from the factor $\left( \frac{\mathcal{Z}(\mathbb{CP}^2; \mathcal{A}_t)}{\mathcal{Z}(\mathbb{CP}^2)} \right)^{p_1^{SO(N)}}$. Consider the inclusion map that maps $SO(2)$ into $SO(N)$, because $p_1^{SO(N)}$ becomes precisely $\left( C_1^{SO(2)} \right)^2$ under the pullback induced by this inclusion map, using the above result for $N = 2$ we get the $SO(N)$ Hall conductance for $N > 2$ with the same formula as Eq. (80).

For the special case of $N = 4$, there can be an additional $SO(2)'$ being attracted, whose amount $\sigma'_{xy}$ can be read off from the factor $\left( \left( \frac{\mathcal{Z}(\mathbb{CP}^2)}{\mathcal{Z}(\mathbb{CP}^2; \mathcal{A}_t)} \right)^2 \right)^{e^{SO(4)}}$. Consider the inclusion map

that maps $SO(2) \times SO(2)'$ into $SO(4)$. It turns out that $e^{SO(4)}$ becomes $\frac{dA \wedge dA'}{4\pi^2}$ under the pullback induced by this inclusion map, where $A$ and $A'$ are the gauge fields for $SO(2)$ and $SO(2)'$, respectively. Similar analysis as above then shows that the fractional part of this Hall conductance is

$$\sigma'_{xy} = \frac{\Theta'}{2\pi}, \quad e^{i\Theta'} = \left( \frac{\mathcal{Z}\left(\mathbb{CP}^2\right)}{\mathcal{Z}(\mathbb{CP}^2; \mathcal{A}_t)} \right)^2 = \left( \frac{\sum_a d_a^2 \theta_a}{\sum_b d_b^2 \theta_b e^{i2\pi q_b}} \right)^2. \tag{81}$$

There can be extra contributions to the Hall conductance from a $(2+1)D$ invertible state as well, which are integral multiples of 1.

For $N > 4$, the flux can also attract certain representation under $SO(N-2)$, which can be read off from the factor $\mathcal{I}^{w_4^{SO(4)}} = \left( \left( \frac{\mathcal{Z}\left(\mathbb{CP}^2\right)}{\mathcal{Z}(\mathbb{CP}^2; \mathcal{A}_t)} \right)^2 \right)^{w_4^{SO(N)}}$. Consider the inclusion map that maps $SO(2) \times SO(N-2)$ into $SO(N)$. It turns out that $w_4^{SO(N)}$ becomes $w_2^{SO(2)} \cup w_2^{SO(N-2)}$. So when the topological order is anomaly-free (anomalous), i.e., $\mathcal{I} = 1$ ($\mathcal{I} = -1$), the flux attracts a linear (spinor) representation under $SO(N-2)$.

Combining the above results of the Hall conductance and the fact that $\mathcal{I}$ takes values in $\pm 1$, we see that the possible values of the Hall conductance $\sigma_{xy}$ of an $SO(N)$ symmetric topological order with $N > 4$ is severely constrained. In particular, if this topological order is anomaly-free, then $\sigma_{xy} = 0$ or $\sigma_{xy} = 1$. If it is anomalous, then $\sigma_{xy} = \pm\frac{1}{2}$, which means that this topological order is incompatible with a further time reversal symmetry. This is related to the phenomenon of "symmetry-enforced gaplessness" [26, 36, 61, 62, 89, 90] discussed in Sec. 6.

# 6 Other symmetry groups

The examples presented in Sec. 4 and Sec. 5 contain many interesting and physically relevant examples. However, for some symmetries, the calculation of the anomaly indicators may be more technically involved, whose expressions may also be more complicated. Moreover, for disconnected Lie group $G$, the identification of $\eta$-factors and $U$-factors is not as straightforward, and the partition function appears to explicitly depend on a specific choice of the map $f : \mathcal{M} \to BG$ associated to a $G$-bundle $\mathcal{G}$ (although we believe that the partition function actually only depends on the homotopy class of $f$).

However, it turns out that even if we consider other symmetry groups, examples presented before can be very useful. Specifically, we discuss symmetries whose anomaly indicators nevertheless can be obtained by simply copying results that we have already derived without any need of further calculations. The common properties of these symmetries $G$ are that i) they have subgroups like $\mathbb{Z}_2^T$, $\mathbb{Z}_2 \times \mathbb{Z}_2$, $\mathbb{Z}_2^T \times \mathbb{Z}_2^T$ and/or $SO(N)$, whose anomaly indicators have already been obtained, and ii) by restricting $G$ to its various subgroups and considering the pullbacks of its anomaly, its anomaly can be uniquely determined. Such symmetries $G$ include $O(N)^T$, $SO(N) \times \mathbb{Z}_2^T$, $\mathbb{Z}_n \times \mathbb{Z}_2^T$, $\mathbb{Z}_n \rtimes \mathbb{Z}_2^T$, $\mathbb{Z}_n \rtimes \mathbb{Z}_2$, $O(N)$, etc. Here $O(N)^T$ means that the symmetry group is $O(N)$, and the superscript $^T$ denotes that elements in $O(N)$ with determinant $-1$ are anti-unitary. For an odd $N$, the groups $O(N)^T$ and $SO(N) \times \mathbb{Z}_2^T$ are actually the same.

In the following two subsections, we illustrate this strategy by calculating the anomaly indicators of $O(N)^T$ and $SO(N) \times \mathbb{Z}_2^T$. Especially, we demonstrate that for $O(N)^T, N \geqslant 5$ and $SO(N) \times \mathbb{Z}_2^T, N \geqslant 4$, certain 't Hooft anomaly cannot be realized by any symmetry-enriched topological order, illustrating the phenomenon of "symmetry-enforced gaplessness", first discussed in Ref. [26]. The anomaly indicators of $O(2)^T$ and $SO(2) \times \mathbb{Z}_2^T$ were first proposed in Ref. [35], while the anomaly indicators of $O(3)^T = SO(3) \times \mathbb{Z}_2^T$ were purposed in Ref. [36].

## 6.1 $O(N)^T$

The relevant bordism group for symmetry group $O(N)^T$ is [87]

$$
\Omega_4^{SO}\left((BO(N))^{q-1}\right) = \begin{cases} (\mathbb{Z}_2)^3, & N = 2, \\ (\mathbb{Z}_2)^4, & N = 3, \\ (\mathbb{Z}_2)^4 \oplus \mathbb{Z}, & N = 4, \\ (\mathbb{Z}_2)^5, & N \geqslant 5. \end{cases} \tag{82}
$$

First consider the case where $N = 2$ and the symmetry group is $O(2)^T$. The anomalies of $O(2)^T$ in (2+1)-dimension are classified by $(\mathbb{Z}_2)^3$, whose basis elements can be chosen as $(w_2^{TM})^2, (w_1^{O(2)})^4, (w_2^{O(2)})^2$, where $w_1^{O(2)}$ and $w_2^{O(2)}$ are the first and second Stiefel-Whitney class belonging to $\mathcal{H}^1(O(2)^T, \mathbb{Z}_2)$ and $\mathcal{H}^2(O(2)^T, \mathbb{Z}_2)$, respectively, and $(w_2^{TM})^2$ is the generator of the beyond-cohomology piece of anomaly. We can write down the anomaly/partition function as

$$
\mathcal{O} = (\mathcal{I}_0)^{(w_2^{TM})^2} \cdot (\mathcal{I}_1)^{\left(w_1^{O(2)}\right)^4} \cdot (\mathcal{I}_0 \mathcal{I}_2)^{\left(w_2^{O(2)}\right)^2}. \tag{83}
$$

Here the appearance of $\mathcal{I}_0 \mathcal{I}_2$ is just to make the final expression nicer and match with the known literature. Denote the anti-unitary element $\mathrm{diag}(-1, 1)$ by $\mathcal{T}$. When pulled back to the $\mathbb{Z}_2^T$ subgroup generated by $\mathcal{T}$, the anomaly becomes

$$
\tilde{\mathcal{O}} = (\mathcal{I}_0)^{(w_2^{TM})^2} \cdot (\mathcal{I}_1)^{t^4}. \tag{84}
$$

Therefore, compared with Eq. (51), we immediately have $\mathcal{I}_0 = \mathcal{Z}(\mathbb{CP}^2)$, given by Eq. (46), and $\mathcal{I}_1 = \mathcal{Z}(\mathbb{RP}^4; \mathcal{T})$, given by Eq. (50). When pulled back to the subgroup $SO(2)$, the anomaly becomes

$$
\tilde{\mathcal{O}} = (\mathcal{I}_0)^{\sigma(\mathcal{M})} \cdot (\mathcal{I}_0 \mathcal{I}_2)^{\left(c_1^{SO(2)}\right)^2}. \tag{85}
$$

Therefore, compared with Eq. (74), we have $\mathcal{I}_2 = \mathcal{Z}\left(\mathbb{CP}^2; \mathcal{A}_t\right)$, given by Eq. (72).

Next consider $N = 3$. The anomalies of $O(3)^T$ in (2+1)-dimension are classified by $(\mathbb{Z}_2)^4$, whose basis elements can be chosen as $(w_2^{TM})^2, (w_1^{O(3)})^4, (w_1^{O(3)})^2 w_2^{O(3)}, (w_2^{O(3)})^2$, where $w_1^{O(3)}$ and $w_2^{O(3)}$ are the first and second Stiefel-Whitney class belonging to $\mathcal{H}^1(O(3)^T, \mathbb{Z}_2)$ and $\mathcal{H}^2(O(3)^T, \mathbb{Z}_2)$, respectively, and $(w_2^{TM})^2$ is the generator of the beyond-cohomology piece of anomaly. We can write down the anomaly as

$$
\mathcal{O} = (\mathcal{I}_0)^{\left(w_2^{TM}\right)^2} \cdot (\mathcal{I}_1)^{\left(w_1^{O(N)}\right)^4} \cdot (\mathcal{I}_0 \mathcal{I}_1 \mathcal{I}_2 \mathcal{I}_3)^{w_2^{O(N)}\left(w_1^{O(N)}\right)^2} \cdot (\mathcal{I}_0 \mathcal{I}_3)^{\left(w_2^{O(N)}\right)^2}. \tag{86}
$$

Again, such a choice of coefficients is just to make the final expression nicer. Denote the anti-unitary element $\mathrm{diag}(-1, 1, 1)$ of $O(3)^T$ by $\mathcal{T}$, and another anti-unitary element $\mathrm{diag}(-1, -1, -1)$ of $O(3)^T$ by $\mathcal{T}' = \mathcal{T}U_\pi$, where $U_\pi$ is a $\pi$ rotation in the 2-3 plane. When pulled back to the subgroup generated by $\mathcal{T}$, the partition function becomes

$$
\tilde{\mathcal{O}} = (\mathcal{I}_0)^{(w_2^{TM})^2} \cdot (\mathcal{I}_1)^{t^4}. \tag{87}
$$

Therefore, compared with Eq. (51), we immediately have $\mathcal{I}_0 = \mathcal{Z}(\mathbb{CP}^2)$, given by Eq. (46), and $\mathcal{I}_1 = \mathcal{Z}(\mathbb{RP}^4; \mathcal{T})$, given by Eq. (50). When pulled back to the subgroup generatd by $\mathcal{T}'$, the partition function becomes

$$
\tilde{\mathcal{O}} = (\mathcal{I}_0)^{(w_2^{TM})^2} \cdot (\mathcal{I}_2)^{t^4}. \tag{88}
$$

Therefore, compared with Eq. (51), we have $\mathcal{I}_2 = \mathcal{Z}(\mathbb{RP}^4; \mathcal{T}')$, again given by Eq. (50), which also can be written in the following form

$$\mathcal{Z}\left(\mathbb{RP}^4; \mathcal{T}'\right) = \frac{1}{D} \sum_{\substack{a \\ \mathcal{T}'a=a}} d_a \theta_a \times \eta_a(\mathcal{T}', \mathcal{T}') = \frac{1}{D} \sum_{\substack{a \\ \mathcal{T}a=a}} d_a \theta_a \times \eta_a(\mathcal{T}, \mathcal{T}) e^{i2\pi q_a}, \qquad (89)$$

where $q_a \in \{0, \frac{1}{2}\}$ labels the symmetry fractionalization class of anyon $a$ under the $SO(N)$ symmetry. Finally, when pulled back to the subgroup generated by $SO(N)$, the partition function becomes

$$\tilde{\mathcal{O}} = (\mathcal{I}_0)^{\sigma(\mathcal{M})} \cdot (\mathcal{I}_0 \mathcal{I}_3)^{p_1^{SO(N)}}. \qquad (90)$$

Therefore, compared with Eq. (75), we have $\mathcal{I}_3 = \mathcal{Z}\left(\mathbb{CP}^2; \mathcal{A}_t\right)$, given by Eq. (72).

When $N = 4$, the anomalies of $O(4)^T$ in (2+1)-dimension are classified by $(\mathbb{Z}_2)^4$, whose basis elements can be chosen as $(w_2^{TM})^2$, $(w_1^{O(4)})^4$, $(w_1^{O(4)})^2 w_2^{O(4)}$, $(w_2^{O(4)})^2$, where $w_1^{O(N)}$ and $w_2^{O(N)}$ are the first and second Stiefel-Whitney class belonging to $\mathcal{H}^1(O(N)^T, \mathbb{Z}_2)$ and $\mathcal{H}^2(O(N)^T, \mathbb{Z}_2)$, and $(w_2^{TM})^2$ is the generator of the beyond-cohomology piece of anomaly. There is an extra U(1) piece in the cobordism group, which is associated to $e_4^{O(4)^T}$, i.e., the twisted Euler class belonging to $\mathcal{H}^4(O(N)^T, \mathbb{Z}_q)$. We can write down the partition function as

$$\mathcal{O} = (\mathcal{I}_0)^{\left(w_2^{TM}\right)^2} \cdot (\mathcal{I}_1)^{\left(w_1^{O(N)}\right)^4} \cdot (\mathcal{I}_0 \mathcal{I}_1 \mathcal{I}_2 \mathcal{I}_3)^{w_2^{O(N)}\left(w_1^{O(N)}\right)^2} \cdot (\mathcal{I}_0 \mathcal{I}_3)^{\left(w_2^{O(N)}\right)^2} \cdot (\tilde{\mathcal{I}})^{e_4^{O(N)^T}}. \qquad (91)$$

Denote the anti-unitary element $\mathrm{diag}(-1, 1, 1, 1)$ of $O(N)^T$ by $\mathcal{T}$, and another anti-unitary element $\mathrm{diag}(-1, -1, -1, 1)$ of $O(N)^T$ by $\mathcal{T}' = \mathcal{T}U_\pi$, where $U_\pi$ is a $\pi$ rotation in the 2-3 plane. From pullback to the subgroup generated by $\mathcal{T}$ and $\mathcal{T}'$, we still have $\mathcal{I}_0 = \mathcal{Z}\left(\mathbb{CP}^2\right)$, given by Eq. (46), $\mathcal{I}_1 = \mathcal{Z}\left(\mathbb{RP}^4; \mathcal{T}\right)$, given by Eq. (50), $\mathcal{I}_2 = \mathcal{Z}\left(\mathbb{RP}^4; \mathcal{T}'\right)$, given by Eq. (50) or (89). When pulled back to the subgroup generated by $SO(4)$, the partition function becomes

$$\tilde{\mathcal{O}} = (\mathcal{I}_0)^{\sigma(\mathcal{M})} \cdot (\mathcal{I}_0 \mathcal{I}_3)^{p_1^{SO(N)}} \cdot (\tilde{\mathcal{I}})^{e_4^{O(4)^T}}. \qquad (92)$$

Therefore, compared with Eq. (77), we have $\mathcal{I}_3 = \mathcal{Z}\left(\mathbb{CP}^2; \mathcal{A}_t\right)$, given by Eq. (72), and $\tilde{\mathcal{I}} = (\mathcal{I}_0/\mathcal{I}_3)^2$. Because both $\mathcal{I}_0$ and $\mathcal{I}_3$ here must take values only in $\pm 1$, $\tilde{\mathcal{I}}$ is always 1. Therefore, the (fractional part of) $SO(4)$ Hall conductance $\sigma'_{xy}$ as in Eq. (81) is always 0.

Finally, when $N \geq 5$, the anomalies of $O(N)^T$ in (2+1)-dimension are classified by $(\mathbb{Z}_2)^5$, whose basis elements can be chosen as $(w_2^{TM})^2$, $(w_1^{O(N)})^4$, $(w_1^{O(N)})^2 w_2^{O(N)}$, $(w_2^{O(N)})^2$ and $w_4^{O(N)}$, where $w_1^{O(N)}$, $w_2^{O(N)}$ and $w_4^{O(N)}$ are the first, second and fourth Stiefel-Whitney class belonging to $\mathcal{H}^1(O(N)^T, \mathbb{Z}_2)$, $\mathcal{H}^2(O(N)^T, \mathbb{Z}_2)$ and $\mathcal{H}^4(O(N)^T, \mathbb{Z}_2)$, respectively, and $(w_2^{TM})^2$ is the generator of the beyond-cohomology piece of anomaly. We can write down the anomaly as

$$\mathcal{O} = (\mathcal{I}_0)^{\left(w_2^{TM}\right)^2} \cdot (\mathcal{I}_1)^{\left(w_1^{O(N)}\right)^4} \cdot (\mathcal{I}_0 \mathcal{I}_1 \mathcal{I}_2 \mathcal{I}_3)^{w_2^{O(N)}\left(w_1^{O(N)}\right)^2} \cdot (\mathcal{I}_0 \mathcal{I}_3)^{\left(w_2^{O(N)}\right)^2} \cdot (\tilde{\mathcal{I}})^{w_4^{O(N)}}. \qquad (93)$$

Denote the anti-unitary element $\mathrm{diag}(-1, 1, 1, 1, \dots)$ of $O(N)^T$ by $\mathcal{T}$, and another anti-unitary element $\mathrm{diag}(-1, -1, -1, 1, \dots)$ of $O(N)^T$ by $\mathcal{T}' = \mathcal{T}U_\pi$, where $U_\pi$ is a $\pi$ rotation in the 2-3 plane. From pullback to the subgroup generated by $\mathcal{T}$ and $\mathcal{T}'$, we still have $\mathcal{I}_0 = \mathcal{Z}\left(\mathbb{CP}^2\right)$, given by Eq. (46), $\mathcal{I}_1 = \mathcal{Z}\left(\mathbb{RP}^4; \mathcal{T}\right)$, given by Eq. (50), $\mathcal{I}_2 = \mathcal{Z}\left(\mathbb{RP}^4; \mathcal{T}'\right)$, given by Eq. (50) or (89). When pulled back to the subgroup generated by $SO(N)$, the partition function becomes

$$\tilde{\mathcal{O}} = (\mathcal{I}_0)^{\sigma(\mathcal{M})} \cdot (\mathcal{I}_0 \mathcal{I}_3)^{p_1^{SO(N)}} \cdot (\tilde{\mathcal{I}})^{w_4^{SO(N)}}. \qquad (94)$$

Therefore, compared with Eq. (77), we have $\mathcal{I}_3 = \mathcal{Z}\left(\mathbb{CP}^2; \mathcal{A}_t\right)$, given by Eq. (72), and $\tilde{\mathcal{I}} = (\mathcal{I}_0/\mathcal{I}_3)^2$. Because both $\mathcal{I}_0$ and $\mathcal{I}_3$ here must take values only in $\pm 1$, we see that $\tilde{\mathcal{I}}$ is always 1. As a result, given a (3+1)-dimensional theory with global symmetry $O(N)^T$ and nontrivial 't Hooft anomaly involving $w_4^{O(N)}$, the boundary cannot be a topologically ordered state, i.e., it can either spontaneously break the $O(N)^T$ symmetry or be a gapless state. This phenomenon is called "symmetry-enforced gaplessness".[13]

As a summary, there are three anomaly indicators of $O(2)^T$:

$$\mathcal{I}_0 = \frac{1}{D}\sum_a d_a^2 \theta_a, \quad \mathcal{I}_1 = \frac{1}{D}\sum_{\substack{a \\ \mathcal{T}a=a}} d_a \theta_a \eta_a(\mathcal{T},\mathcal{T}), \quad \mathcal{I}_2 = \frac{1}{D}\sum_a d_a^2 \theta_a e^{2\pi i q_a}. \tag{95}$$

There are four anomaly indicators of $O(3)^T$ and $O(4)^T$:

$$
\begin{aligned}
\mathcal{I}_0 &= \frac{1}{D}\sum_a d_a^2 \theta_a, \quad \mathcal{I}_1 = \frac{1}{D}\sum_{\substack{a \\ \mathcal{T}a=a}} d_a \theta_a \eta_a(\mathcal{T},\mathcal{T}), \\
\mathcal{I}_2 &= \frac{1}{D}\sum_{\substack{a \\ \mathcal{T}a=a}} d_a \theta_a \eta_a(\mathcal{T},\mathcal{T}) e^{2\pi i q_a}, \quad \mathcal{I}_3 = \frac{1}{D}\sum_a d_a^2 \theta_a e^{2\pi i q_a}.
\end{aligned}
\tag{96}
$$

Here $\mathcal{T}$ denotes the anti-unitary element $\mathrm{diag}(-1,1,\dots)$. For $N \geqslant 5$, these four expressions still give the anomaly indicators for $O(N)^T$, and there is one more anomaly indicator, $\tilde{\mathcal{I}} = (\mathcal{I}_0/\mathcal{I}_3)^2$, which is always 1 and indicates that this anomaly cannot be realized by any topological order. The full anomaly of the topological order can be written in the form of Eqs. (83), (86), (91) and (93), for $N = 2,3,4$ and $N \geqslant 5$, respectively.

## 6.2 $SO(N) \times \mathbb{Z}_2^T$

The relevant bordism group for symmetry group $SO(N) \times \mathbb{Z}_2^T$ is [87]

$$\Omega_4^{SO}\left((B(SO(N) \times \mathbb{Z}_2))^{q-1}\right) = \begin{cases} (\mathbb{Z}_2)^4, & N = 2,3, \\ (\mathbb{Z}_2)^5, & N \geqslant 4. \end{cases} \tag{97}$$

When $N = 2,3$, and the anomalies are classified by $(\mathbb{Z}_2)^4$, whose basis elements can be chosen as $(w_2^{TM})^2, t^4, t^2 w_2^{SO(N)}, (w_2^{SO(N)})^2$, where $t$ is the generator of $\mathcal{H}^1(\mathbb{Z}_2^T, \mathbb{Z}_2)$ and $w_2^{SO(N)}$ is the second Stiefel-Witney class or the generator of $\mathcal{H}^2(SO(N), \mathbb{Z}_2)$. We can write down the anomaly/partition function as

$$\mathcal{O} = (\mathcal{I}_0)^{(w_2^{TM})^2} \cdot (\mathcal{I}_1)^{t^4} \cdot (\mathcal{I}_0 \mathcal{I}_1 \mathcal{I}_2 \mathcal{I}_3)^{w_2^{SO(N)} t^2} \cdot (\mathcal{I}_0 \mathcal{I}_3)^{\left(w_2^{SO(N)}\right)^2}. \tag{98}$$

Denote the anti-unitary generator of $\mathbb{Z}_2^T$ as $\mathcal{T}$ and a $\pi$-rotation of $SO(N)$ as $U_\pi$. When pulled back to the subgroup generated by $\mathcal{T}$, the anomaly becomes

$$\tilde{\mathcal{O}} = (\mathcal{I}_0)^{(w_2^{TM})^2} \cdot (\mathcal{I}_1)^{t^4}. \tag{99}$$

---

[13]For the reader's convenience, we repeat the argument in Refs. [10, 11] of "symmetry-enforced gaplessness" discussed here. Consider an $SO(N)$ monopole, represented as a unit $SO(2) \subset SO(N)$ monopole in the first two components. The $w_4^{O(N)}$ anomaly requires the monopole to carry spinor representation for the remaining $SO(N-2)$. For a gapped topologically ordered state, this condition can be satisfied only by attaching a gapped anyon excitation to the monopole, with the anyon carrying spinor representation under $SO(N-2)$. But an anyon should carry irreducible representation under the entire $SO(N)$, which means that the $SO(N-2)$ spinor anyon should also carry $SO(2)$ charge 1/2. This leads to a nontrivial Hall conductance for the $SO(2)$, which necessarily breaks time-reversal symmetry. This argument can also be carried over to the symmetry group $SO(N) \times \mathbb{Z}_2^T, N \geqslant 4$ with anomaly involving $w_4^{SO(N)}$. However, it does not apply to $O(4)^T$, because in that case the charge of the $SO(4)$ monopole under $SO(2)'$ is not quantized by time reversal.

Therefore, compared with Eq. (51), we immediately have $\mathcal{I}_0 = \mathcal{Z}(\mathbb{CP}^2)$, given by Eq. (46), $\mathcal{I}_1 = \mathcal{Z}(\mathbb{RP}^4; \mathcal{T})$, given by Eq. (50). When pulled back to the subgroup generated by $\mathcal{T}U_\pi$, the anomaly becomes

$$\tilde{\mathcal{O}} = (\mathcal{I}_0)^{(w_2^{TM})^2} \cdot (\mathcal{I}_2)^{t^4} \,. \tag{100}$$

Therefore, compared with Eq. (51), we have $\mathcal{I}_2 = \mathcal{Z}(\mathbb{RP}^4; \mathcal{T}')$, again given by Eq. (50), which can also be written in the following form

$$\mathcal{Z}\left(\mathbb{RP}^4; \mathcal{T}'\right) = \frac{1}{D} \sum_{\substack{a \\ \mathcal{T}'a=a}} d_a \theta_a \times \eta_a(\mathcal{T}', \mathcal{T}') = \frac{1}{D} \sum_{\substack{a \\ \mathcal{T}a=a}} d_a \theta_a \times \eta_a(\mathcal{T}, \mathcal{T}) e^{i2\pi q_a} \,, \tag{101}$$

where again $q_a \in \{0, \frac{1}{2}\}$ labels the symmetry fractionalization class of anyon $a$ under $SO(N)$ symmetry (even for $N = 2$, here $q_a$ can only take values from $\{0, \frac{1}{2}\}$). When pulled back to the subgroup $SO(N)$, the anomaly becomes for $N = 2$

$$\tilde{\mathcal{O}} = (\mathcal{I}_0)^{\sigma(\mathcal{M})} \cdot (\mathcal{I}_0 \mathcal{I}_3)^{\left(c_1^{SO(2)}\right)^2} \,, \tag{102}$$

or for $N = 3$

$$\tilde{\mathcal{O}} = (\mathcal{I}_0)^{\sigma(\mathcal{M})} \cdot (\mathcal{I}_0 \mathcal{I}_3)^{p_1^{SO(N)}} \,. \tag{103}$$

Therefore, compared with Eq. (74) or Eq. (75), we have $\mathcal{I}_3 = \mathcal{Z}\left(\mathbb{CP}^2; \mathcal{A}_t\right)$, given by Eq. (72).

When $N \geqslant 4$, the anomalies of $SO(N) \times \mathbb{Z}_2^T$ in (2+1)-dimension are classified by $(\mathbb{Z}_2)^5$, whose basis elements can be chosen as $(w_2^{TM})^2$, $t^4$, $t^2 w_2^{SO(N)}$, $(w_2^{SO(N)})^2$ and $w_4^{SO(N)}$, where $t$ is the generator of $\mathcal{H}^1(\mathbb{Z}_2^T, \mathbb{Z}_2)$, and $w_2^{SO(N)}$ ($w_4^{SO(N)}$) is the second (fourth) Stiefel-Whitney class or the generator of $\mathcal{H}^2(SO(N), \mathbb{Z}_2)$ ($\mathcal{H}^4(SO(N), \mathbb{Z}_2)$). We can write down the anomaly/partition function as

$$\mathcal{O} = (\mathcal{I}_0)^{(w_2^{TM})^2} \cdot (\mathcal{I}_1)^{t^4} \cdot (\mathcal{I}_0 \mathcal{I}_1 \mathcal{I}_2 \mathcal{I}_3)^{w_2^{SO(N)} t^2} \cdot (\mathcal{I}_0 \mathcal{I}_3)^{\left(w_2^{SO(N)}\right)^2} \cdot (\tilde{I})^{w_4^{SO(N)}} \,. \tag{104}$$

Denote the anti-unitary generator of $\mathbb{Z}_2^T$ as $\mathcal{T}$ and a $\pi$-rotation of $SO(N)$ as $U_\pi$. From pullback to the subgroup generated by $\mathcal{T}$ and $\mathcal{T}U_\pi$, we still have $\mathcal{I}_0 = \mathcal{Z}\left(\mathbb{CP}^2\right)$, given by Eq. (46), $\mathcal{I}_1 = \mathcal{Z}\left(\mathbb{RP}^4; \mathcal{T}\right)$, given by Eq. (50), $\mathcal{I}_2 = \mathcal{Z}\left(\mathbb{RP}^4; \mathcal{T}'\right)$, given by Eq. (50) or (101). When pulled back to the subgroup generated by $SO(N)$, the partition function becomes

$$\tilde{\mathcal{O}} = (\mathcal{I}_0)^{\sigma(\mathcal{M})} \cdot (\mathcal{I}_0 \mathcal{I}_3)^{p_1^{SO(N)}} \cdot (\tilde{I})^{w_4^{SO(N)}} \,. \tag{105}$$

Therefore, compared with Eq. (76) or (77), we have $\mathcal{I}_3 = \mathcal{Z}\left(\mathbb{CP}^2; \mathcal{A}_t\right)$, given by Eq. (72), and $\tilde{\mathcal{I}} = (\mathcal{I}_0/\mathcal{I}_3)^2$. Again, because both $\mathcal{I}_0$ and $\mathcal{I}_3$ here must take values only in $\pm 1$, we see that $\tilde{\mathcal{I}}$ is identically 1. Consequently, given a (3+1)-dimensional theory with global symmetry $SO(N) \times \mathbb{Z}_2^T$ and nontrivial 't Hooft anomaly involving $w_4^{SO(N)}$, the boundary cannot be a topologically ordered state, i.e., it can either spontaneously break the $SO(N) \times \mathbb{Z}_2^T$ symmetry or be a gapless state. We again discover the phenomenon of "symmetry-enforced gaplessness".

In summary, there are four anomaly indicators of $SO(N) \times \mathbb{Z}_2^T$:

$$\begin{aligned} \mathcal{I}_0 &= \frac{1}{D} \sum_a d_a^2 \theta_a \,, \quad \mathcal{I}_1 = \frac{1}{D} \sum_{\substack{a \\ \mathcal{T}a=a}} d_a \theta_a \eta_a(\mathcal{T}, \mathcal{T}) \,, \\[2mm] \mathcal{I}_2 &= \frac{1}{D} \sum_{\substack{a \\ \mathcal{T}a=a}} d_a \theta_a \eta_a(\mathcal{T}, \mathcal{T}) e^{2\pi i q_a} \,, \quad \mathcal{I}_3 = \frac{1}{D} \sum_a d_a^2 \theta_a e^{2\pi i q_a} \,. \end{aligned} \tag{106}$$

Here $\mathcal{T}$ denotes the generator of $\mathbb{Z}_2^T$. For $N = 2, 3$, these are all the anomaly indicators. For $N \geqslant 4$, besides these four anomaly indicators, there is another one $\tilde{\mathcal{I}} = (\mathcal{I}_0/\mathcal{I}_3)^2$, which is always 1 and implies that such anomaly cannot be realized by any topological order. The full anomaly of the topological order can be written in the form of Eq. (98) and (104), for $N = 2, 3$ and $N \geqslant 4$, respectively.

# 7 Discussion

In summary, we have constructed a $(3+1)D$ TQFT given the data of a UMTC and $G$-action on the UMTC. The partition functions of this TQFT on certain representative manifolds equipped with appropriate $G$ bundles give the anomaly indicators of $(2+1)D$ bosonic topological orders enriched with a finite group symmetry $G$, which may be Abelian or non-Abelian, contain anti-unitary elements and permute anyons. Via this framework, besides reproducing the known anomaly indicators of $G = \mathbb{Z}_2^T$, we have calculated the anomaly indicators of $G = \mathbb{Z}_2 \times \mathbb{Z}_2$ and $G = \mathbb{Z}_2^T \times \mathbb{Z}_2^T$, which have not been previously derived as far as we know. The usage of these anomaly indicators have been demonstrated in the example of all-fermion $\mathbb{Z}_2$ topological orders. This framework is generalized to the case where the relevant symmetry is a connected Lie group, and we use it to derive the anomaly indicator for $SO(N)$. As a byproduct, we also obtain the expressions of the Hall conductance of an $SO(N)$ symmetric topological order, written in terms of data characterizing this symmetry-enriched topological order. We explain how to use these results to calculate the anomaly indicators for some other symmetry groups without the need of further calculation, and explicitly derive the anomaly indicators for symmetry groups $O(N)^T$ and $SO(N) \times \mathbb{Z}_2^T$. In particular, we show that certain anomalies associated with these symmetries cannot be realized by any topological order.

Being able to calculate the anomaly is extremely useful, because the anomaly is powerful in constraining the possible low-energy dynamics of a strongly interacting field theory, which is often challenging to understand by other means. For example, if a strongly interacting field theory with some symmetry has an anomaly different from the ones we calculate for a symmetry-enriched topological order, this field theory cannot flow to this symmetry-enriched topological order at low energies under renormalization group. Moreover, according to the hypothesis of emergibility, the ability to calculate the anomaly of a quantum phase or phase transition is crucial to understand whether this phase or transition can emerge in a given quantum many-body system, whose robust microscopic properties are compactly encoded in their Lieb-Schultz-Mattis-type anomalies [11, 18]. Going one step further, this hypothesis provides a possible route to solve the open problem of classifying topological orders with lattice symmetries, in a way similar to the classification of various symmetry-enriched quantum critical states in Ref. [18]. We believe that this work is an important step towards these goals.

From the mathematical side, our work spells out in detail how to deal with $G$-bundle structure in real calculation of the partition function of TQFT. In particular, on each 1-handle the $G$-bundle structure is mapped to a functor acting on the vector space (topological state space), while on each 2-handle it is mapped to a natural isomorphism acting on the object (anyon). This is consistent with the general treatment of the $G$-bundle structure in Ref. [69], and serves as a nice demonstration of the real computational power of TQFT porposed therein.

Below we briefly comment on some future directions.

1. It is natural to generalize the calculation to other groups and obtain the anomaly indicators of these groups. For example, it is easy to generalize the calculation in Sec. 4.3 to the group $\mathbb{Z}_m \times \mathbb{Z}_n$. The manifolds that we should consider are $L(m, 1) \times S^1$, with a $\mathbb{Z}_m$ defect on the noncontractible cycle of $L(m, 1)$ and a $\mathbb{Z}_n$ defect on $S^1$, and $L(n, 1) \times S^1$, with a $\mathbb{Z}_n$

defect on the noncontractible cycle of $L(n,1)$ and a $\mathbb{Z}_m$ defect on $S^1$. The handle decomposition of these manifolds are straightforward generalizations of $L(2,1)\times S^1 = \mathbb{RP}^3\times S^1$, whose Kirby diagram is already shown in Fig. 8, and they are also explained in detail in Ref. [80]. We can also consider the group $(\mathbb{Z}_2)^4$ and one of the representative manifolds relevant to the calculation of anomaly indicators is $(S^1)^4$, with one different $\mathbb{Z}_2$ defect on each $S^1$ cycle. See Ref. [81] for a detailed explanation of the handle decomposition and the Kirby diagram of $(S^1)^4$. It is also natural to consider other Lie group symmetries, as in the discussion in Ref. [46]. We defer them to future study.

2. We have described in detail how to deal with the (3+1)$D$ TQFT equipped with a finite group symmetry or a connected Lie group symmetry. However, for the most general symmetry $G$, a $G$-bundle structure on a manifold $\mathcal{M}$ is still specified by a map $f : \mathcal{M} \to BG$. In this case, our recipe outlined in Sec. 5 is tedious, and we have not shown that the recipe gives a partition function that is indeed topological, in the sense that it only depends on the homotopy class of $f$ (but not the specific choice of $f$ in each homotopy class). It will be nice in the future to rigorously prove the topological nature of the partition function, possibly in a more abstract level following the most general treatment of Ref. [69].

3. It is natural to extend our formalism to fermionic systems and calculate the anomalies of (2+1)$D$ fermionic topological orders from (3+1)$D$ spin TQFTs, similar to Ref. [39]. In particular, according to Ref. [39, 91], the partition function of a (3+1)$D$ fermionic SPT can be obtained by combining a so-called "bosonic shadow" part and another part that originates from the fermionic nature of the system, denoted by $z_c$ in Ref. [39]. Our framework allows us to obtain the bosonic shadow much easier, given a handle decomposition of the manifold. Moreover, the prescription to obtain $z_c$ given a handle decomposition is also relatively straightforward. So our framework can be generalized to fermionic systems, and we believe the calculation will be greatly simplified as well. We defer the full details to future work.

4. Such calculation may shed light on the phenomenon of symmetry-enforced gaplessness [26, 61, 62], and possibly even generate a necessary and sufficient condition for certain element of 't Hooft anomaly being not realized by any symmetry-enriched topological order. There have already been a lot of attempts in this direction, including Refs. [75, 76, 89, 90, 92], and we wish to push it further to more general situations.

5. We have been focusing on the case where $G$ is an internal symmetry, and it is interesting and important to generalize the framework to incorporate lattice symmetries, which are important in many condensed matter systems. Ref. [34] already derived the anomaly indicators for the reflection symmetry, but the anomaly indicators for a generic lattice symmetry have not been derived. Based on the crystalline equivalence principle [93], which roughly states that the classifications of the anomalies associated with an internal symmetry $G$ and anomalies associated with a lattice symmetry $G$ are the same, we expect that the final results of the anomaly indicators for lattice symmetries take a similar form as the ones for internal symmetries.

6. The anomaly of symmetry-enriched topological order with symmetry group $G$, at least when $G$ contains unitary symmetry only, can also be interpreted as an obstruction of extending some UMTC $\mathcal{C}$ to a $G$-crossed braided fusion category $\mathcal{C}_G$ with compatible $G$ action, where a $G$-crossed braided fusion category is a tensor category whose objects are graded by elements $\mathbf{g} \in G$, and objects graded by $\mathbf{1}$ form a subcategory that is precisely the original UMTC $\mathcal{C}$ [27, 29]. Therefore, our paper gives a well-defined procedure to

calculate this obstruction as well. However, it is still nice to see the connection between our calculation and the obstruction in the context of category theory more directly, and perhaps even rederive our formula purely from cateogry theory, similar to the analysis in Refs. [27,30]. In the presence of anti-unitary symmetry, it is also nice to see how the "beyond-cohomology" anomaly comes into play, given a suitable generalization of the notion of $G$-crossed braided fusion category (potentially via a proper generalization of the 3-functor $BG \to \mathcal{C}$ defined in Ref. [94]).

7. It is intriguing to see how the anomalies associated with the exact 0-form symmetries discussed in this paper are related to the anomalies associated with the generalized emergent symmetries of a topological order. It is already known that when the exact 0-form symmetry $G$ is unitary and does not permute anyons, the anomaly of $G$ is the pullback of the anomaly of the 1-form symmetry $\mathcal{A}$ of the topological order [45]. Here $\mathcal{A}$ is precisely the Abelian group reviewed in Sec. 2, whose group elements correspond to the Abelian anyons in this UMTC and the group multiplication corresponds to the fusion of these Abelian anyons. The symmetry fractionalization class given by an element in $\mathcal{H}^2(G, \mathcal{A})$ is interpreted as a map from $BG$ to $B^2\mathcal{A}$. When $G$ contains anti-unitary symmetry and/or does permute anyons, it is natural to think that the anomaly of $G$ is still the pullback of the anomaly of some emergent 2-group symmetry, as discussed in Refs. [16, 17, 27, 45]. We believe our result can shed light on both the anomaly of this 2-group symmetry and the calculation of the pullback in relevant contexts.

*Note added*: In the first arXiv version (v1), only a finite group symmetry is incorporated into the $(3+1)D$ TQFT. In this new version, we have generalized the framework to the case with a continuous symmetry. Also, in v1 a method based on pulling back the topological symmetry and relative anomaly was proposed to calculate the anomaly of a given symmetry-enriched topological order. This method actually does not apply to the general case. For example, it does not apply to the case where the topological symmetry suffers from a nontrivial $H^3$ obstruction.

## Acknowledgements

We thank John W. Barret, Daniel Bulmash, Meng Guo, Yin-Chen He, Chao-Ming Jian, Tian Lan, Ruochen Ma, David Reutter, Kevin Walker, Chong Wang, Matthew Yu, Keyou Zeng, Zhi-Hao Zhang and Yehao Zhou for helpful discussion. WY would like to especially thank Matthew Yu for the discussion of anomaly indicators in general, Keyou Zeng for the discussion regarding category theory, and Chong Wang and Kevin Walker for correcting some mistakes I made during calculation.

**Funding information** WY acknowledges supports from the Natural Sciences and Engineering Research Council of Canada(NSERC) through Discovery Grants. Research at Perimeter Institute is supported in part by the Government of Canada through the Department of Innovation, Science and Industry Canada and by the Province of Ontario through the Ministry of Colleges and Universities.

# A Derivation of Eq. (44)

For the reader's convenience, in this appendix we repeat some explicit computations of various factors in Eq. (42), including partition functions of various handles and inner products, which ultimately lead to the main formula Eq. (44). The presentation here follows Ref. [34], see also Ref. [48] for the calculation from a higher-category point of view.

## A.1 Vector Spaces

First of all, in this sub-appendix, we write down $\mathcal{V}(\mathcal{N})$, the vector space associated to some 3-dimensional manifold $\mathcal{N}$ which will be defined as the attaching region of some $k$-handle, following the diagrammatic definition in Eq. (38). This will serve as the starting point of our diagrammatic treatment and calculation.

A 4-handle is attached to lower handles along $S^3$, and it is clear that

$$\mathcal{V}(S^3) \simeq \mathbb{C} \tag{A.1}$$

is one-dimensional, spanned by the empty diagram in $S^3$, as all closed anyon diagrams in $S^3$ can be reduced via local moves to a multiple of the empty diagram.

Similarly, a 3-handle is attached to lower handles along $S^2 \times D^1$, and we have

$$\mathcal{V}(S^2 \times D^1; \emptyset) \simeq \mathbb{C}. \tag{A.2}$$

We use $\emptyset$ to denote that we put only trivial anyon on the boundary.

A 2-handle is attached to lower handles along $S^1 \times D^2$. It is also clear that

$$\mathcal{V}(S^1 \times D^2; \emptyset) \simeq \mathbb{C}^{|\mathcal{C}|}. \tag{A.3}$$

Here, $|\mathcal{C}|$ denotes the number of simple anyons in $\mathcal{C}$. The basis vector in $\mathcal{V}(S^1 \times D^2; \emptyset)$ associated to an anyon $a \in \mathcal{C}$ corresponds to putting the anyon loop with label $a$ along $S^1 \times \{\text{pt}\} \subset S^1 \times D^2$, where $\{\text{pt}\}$ denotes a point in $D^2$.

Finally, a 1-handle is attached to lower handles along two copies of $D^3$, and we have

$$\mathcal{V}\left(D^3; (a_1, \dots; b_1, \dots)\right) \simeq V^{a_1, \dots}_{b_1, \dots}. \tag{A.4}$$

Here $a_1, \dots$ and $b_1, \dots$ are used to denote anyons associated to 2-handles running out of and into the 1-handle along the boundary of $D^3$. And $V^{a_1, \dots}_{b_1, \dots}$ is the fusion space of $a_1, \dots$ into $b_1, \dots$. This is illustrated in the upper plane of Fig. 12.

## A.2 Partition functions

In this sub-appendix, we compute the partition functions for different handles. Suppose for $D^4$ we have

$$\mathcal{Z}(D^4; \emptyset) = \lambda, \tag{A.5}$$

where $\lambda$ is a parameter to be fixed later and $\emptyset$ denotes the empty diagram in $\partial D^4 = S^3$. Then if there is some anyon diagram $K$ on $S^3$, we have

$$\mathcal{Z}(D^4; K) = \lambda \langle K \rangle, \tag{A.6}$$

where $\langle K \rangle$ denotes the evaluation of the anyon diagram $K$.

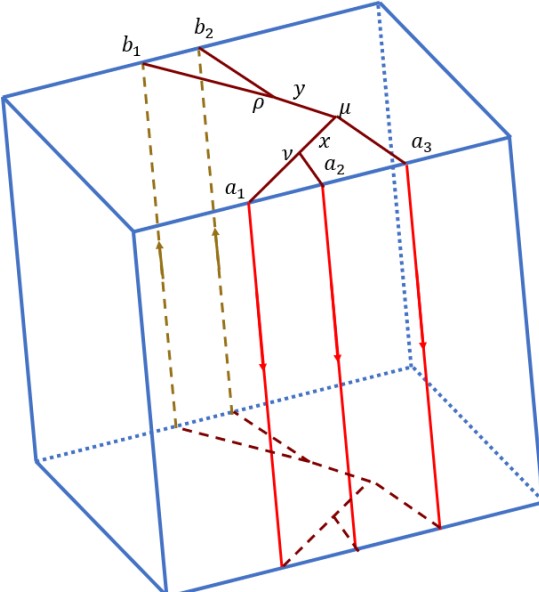

Figure 12: Illustration of the 1-handle, with no defect present. The 1-handle has topology of a $D^4$ but we draw it as a $D^3$ for illustration. The lower plane displays a vector $(x, \nu, \mu)(y, \mu, \rho)$ that lives in the vector space associated to $D^3$, i.e., $\mathcal{V}\left(D^3; (a_1, \dots; b_1, \dots)\right) \in V^{a_1, a_2, a_3}_{b_1, b_2}$, while the upper plane hosts a dual vector.

Specifically, first consider the situation where no defect is present. For a 2-handle, there is a loop $l_a$ of anyon $a$ on $S^1$ and we have

$$\mathcal{Z}(D^4; l_a) = \lambda d_a. \tag{A.7}$$

For a 1-handle there is a $\Theta$-diagram as in Fig. 12, and if no defect is present the evaluation of the diagram gives

$$\mathcal{Z}(D^4; \Theta_{a_1, \dots; b_1, \dots}) = \lambda \sqrt{\prod_i d_{a_i} \prod_j d_{b_j}}. \tag{A.8}$$

For a 0-handle the anyon diagram $K$ on the boundary $S^3$ is precisely the Kirby diagram of the manifold $\mathcal{M}$, with correct labels on the attaching regions of 1-handles and 2-handles.

In the presence of defects, for a 2-handle the associated anyon $a$ is acted on by successive defects, but the combination of all defects along the $S^1$ line of a 2-handle is still a trivial defect, since this $S^1$ is contractible. Nevertheless, the functor of successive symmetry actions is not the same as the identity functor, and they are connected to each other by some natural isomorphism, which when acting on $a$ gives the desired $\eta$-factor. This is explained in detail in Sec. 3.4.

For a 1-handle we just need to take account of the symmetry action on the vector assigned to the boundary, and then calculate the $\Theta$-diagram. In particular, from the symmetry action we get the desired $U$-factor in Eq. (43).

## A.3 Inner Products

Next, in this sub-appendix, we calculate the inner products in the vector spaces described in Appendix A.1.

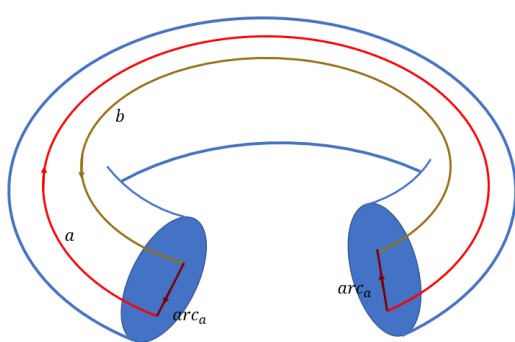

Figure 13: Illustration of the calculation of $\langle l_a | l_b \rangle_{\mathcal{V}(S^1 \times D^2; \emptyset)}$ through $\mathcal{Z}(S^1 \times D^3)[l_{\bar{a}} \cup l_b]$.

For a vector in $|\beta\rangle \in \mathcal{V}\big(D^3; (a_1, \dots; b_1, \dots)\big)$ representing the label on the boundary of Fig. 12, from Eq. (39) we see that

$$\langle \beta | \beta \rangle = \mathcal{Z}(D^4; \Theta_{a_1, \dots; b_1, \dots}) = \lambda \sqrt{\prod_i d_{a_i} \prod_j d_{b_j}}. \tag{A.9}$$

Specifically, in the presence of only 2 anyons, $\dim \mathcal{V}\big(D^3; (a; b)\big) = \delta_{ab}$, and when $a = b$, $\mathcal{V}\big(D^3; (a; b)\big)$ is 1-dimensional and spanned by an arc that we denote as $\text{arc}_a$ connecting the two anyons. The inner product is

$$\langle \text{arc}_a | \text{arc}_b \rangle_{\mathcal{V}(D^3; (a;b))} = \mathcal{Z}(D^4; l_a) \delta_{ab} = d_a \lambda \delta_{ab}. \tag{A.10}$$

Then consider the inner product in $\mathcal{V}(S^1 \times D^2; \emptyset)$. Let $|l_a\rangle$ denote the basis vector in $\mathcal{V}(S^1 \times D^2; \emptyset)$ corresponding to anyon loop $a$ along $S^1$. From Eq. (39), we have

$$\langle l_a | l_b \rangle_{\mathcal{V}(S^1 \times D^2; \emptyset)} = \mathcal{Z}(S^1 \times D^3; l_{\bar{a}} \cup l_b). \tag{A.11}$$

From the gluing formula Eq. (40),

$$
\begin{aligned}
\mathcal{Z}(S^1 \times D^3; l_{\bar{a}} \cup l_b) &= \sum_{|\beta\rangle \in \mathcal{V}(D^3; (a;b))} \frac{\mathcal{Z}(D^4; \text{arc}_b \cup \overline{\beta} \cup \text{arc}_a \cup \beta)}{\langle \beta | \beta \rangle_{\mathcal{V}(D^3; (a;b))}} \\
&= \delta_{ab} \frac{\mathcal{Z}(D^4; \text{arc}_a \cup \text{arc}_a \cup \text{arc}_a \cup \text{arc}_a)}{\langle \text{arc}_a | \text{arc}_a \rangle_{\mathcal{V}(D^3; (a;b))}} \\
&= \delta_{ab} \frac{\mathcal{Z}(D^4; l_a)}{\langle \text{arc}_a | \text{arc}_a \rangle_{\mathcal{V}(D^3; (a;b))}} \\
&= \delta_{ab}. \tag{A.12}
\end{aligned}
$$

Here we have used the fact that cutting $l_{\bar{a}} \cup l_b$ gives rise to two arcs, $\text{arc}_b$ and $\text{arc}_a$, while $\mathcal{V}(D^3; (a; b))$ is spanned by an arc connecting $a$ and $b$. Thus the combination $\text{arc}_a \cup \text{arc}_a \cup \text{arc}_a \cup \text{arc}_a = l_a$. This is illustrated in Fig. 13.

Then let us consider the inner product in $\mathcal{V}(S^2 \times D^1; \emptyset)$. For $|\emptyset\rangle \in \mathcal{V}(S^2 \times D^1; \emptyset)$ denoting the empty diagram, we have

$$
\begin{aligned}
\langle \emptyset | \emptyset \rangle_{\mathcal{V}(S^2 \times D^1; \emptyset)} &= \mathcal{Z}(S^2 \times D^2; \emptyset) \\
&= \sum_{|l_a\rangle \in \mathcal{V}(S^1 \times D^2)} \frac{\mathcal{Z}(D^4; l_a) \mathcal{Z}(D^4; l_a)}{\langle l_a | l_a \rangle_{\mathcal{V}(S^1 \times D^2; \emptyset)}} \\
&= \sum_a d_a^2 \lambda^2 = D^2 \lambda^2, \tag{A.13}
\end{aligned}
$$

where $D$ is the total dimension.

Finally, for $\mathcal{V}(S^3)$ and a basis vector $|\emptyset\rangle$ denoting the empty diagram, we have

$$\langle\emptyset|\emptyset\rangle_{\mathcal{V}(S^3)} = \mathcal{Z}(D^1 \times S^3; \emptyset)$$
$$= \frac{\mathcal{Z}(D^4; \emptyset)\mathcal{Z}(D^4; \emptyset)}{\langle\emptyset|\emptyset\rangle_{\mathcal{V}(S^2 \times D^1; \emptyset)}}$$
$$= \frac{1}{D^2}. \tag{A.14}$$

## A.4 Requirement from Invertibility

A further constraint comes from our wish to define an invertible TQFT [71,72], given a suitable choice of $\lambda$. This means that on every closed 3-manifold $\mathcal{N}$ the associated vector space $\mathcal{V}(\mathcal{N})$ is one-dimensional, and on every closed 4-manifold the partition function is a pure phase factor.

Consider $\mathcal{Z}(S^4)$, the gluing formula Eq. (40) gives

$$\mathcal{Z}(S^4) = \frac{\mathcal{Z}(D^4; \emptyset)\mathcal{Z}(D^4; \emptyset)}{\langle\emptyset|\emptyset\rangle_{\mathcal{V}(S^3)}} = \lambda^2 D^2. \tag{A.15}$$

In order for $|\mathcal{Z}(S^4)| = 1$, we must choose $|\lambda| = \frac{1}{D}$.

Furthermore, we have found in Eq. (A.10) that the norm of the state $|\text{arc}_0\rangle$ is $\lambda$. In a unitary TQFT, norms are always positive definite, so $\lambda > 0$. Therefore we have determined

$$\lambda = \frac{1}{D}. \tag{A.16}$$

As a result we also have $\mathcal{Z}(S^4) = 1$.

Assembling all these factors together, we finally arrive at Eq. (44).

## B An explicit expression of the $\eta$-factor

In Remark g of Sec. 3.4, we have explained that, for a finite group symmetry $G$, the $\eta$-factor associated with a 2-handle comes from the natural isomorphism connecting the functor $\rho_{\mathbf{g}_1}^{s_1} \circ \rho_{\mathbf{g}_2}^{s_2} \circ \cdots$ and the identity functor, where $\mathbf{g}_{1,2,\ldots}$ are defects the $S^1$ line of this 2-handle crosses, starting from a segment with anyon label $a$, and $s_{1,2,\ldots}$ are determined by whether the $S^1$ crosses the defect upward or downward, according to the convention in Remark f. In this appendix, we give an explicit expression of this $\eta$-factor. We stress again that the expression of this $\eta$-factor is not unique, and different expressions can be converted into each other via Eq. (24). The expression presented here is obtained by "combining from left to right" of the functor $\rho_{\mathbf{g}_1}^{s_1} \circ \rho_{\mathbf{g}_2}^{s_2} \circ \cdots$.

To describe this expression, we first write down the $\eta$-factor we get after connecting $\rho_{\mathbf{g}_1}^{s_1} \circ \rho_{\mathbf{g}_2}^{s_2}$ to a single functor $\rho_{\mathbf{g}_{12}}^{s_{12}}$, i.e.,

$$H_{12}: \quad \rho_{\mathbf{g}_1}^{s_1} \circ \rho_{\mathbf{g}_2}^{s_2} \Longrightarrow \rho_{\mathbf{g}_{12}}^{s_{12}}, \tag{B.1}$$

where $\mathbf{g}_{12}$ and $s_{12}$ are defined as follows

$$\mathbf{g}_{12} \equiv \begin{cases} \mathbf{g}_2\mathbf{g}_1, & s_1 = s_2 = -1, \\ \mathbf{g}_1^{s_1}\mathbf{g}_2^{s_2}, & \text{else}, \end{cases} \tag{B.2}$$

$$s_{12} \equiv \begin{cases} -1, & s_1 = s_2 = -1, \\ 1, & \text{else}, \end{cases} \tag{B.3}$$

and $H_{12}$ acting on anyon $a$ gives the following $\eta$-factor

$$(H_{12})_a = \begin{cases} \eta_a(\mathbf{g}_1, \mathbf{g}_2), & s_1 = s_2 = 1, \\ \eta_{\mathbf{g}_2\mathbf{g}_1 a}(\mathbf{g}_2, \mathbf{g}_1)^{-\sigma(\mathbf{g}_2\mathbf{g}_1)}, & s_1 = s_2 = -1, \\ \eta_a(\mathbf{g}_1\mathbf{g}_2^{-1}, \mathbf{g}_2)^{-1}, & s_1 = 1, s_2 = -1, \\ \eta_{\mathbf{g}_1 a}(\mathbf{g}_1, \mathbf{g}_1^{-1}\mathbf{g}_2)^{-\sigma(\mathbf{g}_1)}, & s_1 = -1, s_2 = 1. \end{cases} \tag{B.4}$$

Then we have an expression of the form $\rho_{\mathbf{g}_{12}}^{\mathbf{s}_{12}} \circ \rho_{\mathbf{g}_3}^{s_3} \circ \cdots$, and we can iterate the above process until we get the identity functor. Finally, simply multiplying all individual $\eta$-factors we get the $\eta$-factor associated with the 2-handle,

$$\left(H_{1,2}\right)_a \cdot \left(H_{12,3}\right)_a \cdot \left(H_{123,4}\right)_a \cdot \ldots \tag{B.5}$$

# C  Consistency check of TQFT

There are multiple consistency checks that we need to perform in order to confirm that, for finite group symmetry $G$, the recipe in Sec. 3.4, especially Eq. (44), indeed gives rise to a well-defined partition function $\mathcal{Z}(\mathcal{M}, \mathcal{G})$, defined on a target manifold $\mathcal{M}$ together with a $G$-bundle structure $\mathcal{G}$ on it. In this appendix we explicitly perform the consistency checks and prove that the recipe in Sec. 3.4 does give rise to a well-defined partition function, in the sense that we will make explicit in the following subsections. Most of our exposition will utilize similar proofs for the Crane-Yetter model, as in Refs. [34,48,52,54] for example. However, we need to understand the roles played by symmetry defects. These checks provide further evidence that the partition function $\mathcal{Z}(\mathcal{M}, \mathcal{G})$ constructed in Sec. 3.4 is indeed exactly the same partition function of the TQFT decribed in Sec. 3.2.

The checks we perform include:

1. Independence of the partition function on the handle decomposition in Appendix C.1.

2. Invariance of the partition function under changes of defects in Appendix C.2.

3. Gauge invariance of the partition function in Appendix C.3.

4. Cobordism invariance of the partition function in Appendix C.4.

5. Invertibility of the partition function in Appendix C.5.

For connected Lie group symmetry $G$, we also need to prove that Eq. (65) gives rise to a well-definied partition function. Such a proof follows closely the proof for a finite group symmetry $G$ but is much easier, since we just need to focus on $\eta$-factors. This proof is presented in Appendix C.6.

## C.1  Independence on the handle decomposition

First of all, our construction explicitly uses a handle decomposition of the target manifold $\mathcal{M}$. In this sub-appendix, we prove that the partition function $\mathcal{Z}(\mathcal{M}, \mathcal{G})$ we get in Eq. (44) is in fact independent of the handle decomposition.

Two different handle decompositions of a given manifold are related to each other by the following handle moves: isotopies, handle slides and creating/annihilating cancelling handle pairs [80]. In order to prove the independence of the partition function on the handle decomposition, we just need to show its invariance under all handle moves. Fortunately, most handle moves do not involve $G$-defects, and therefore the partition function is automatically invariant

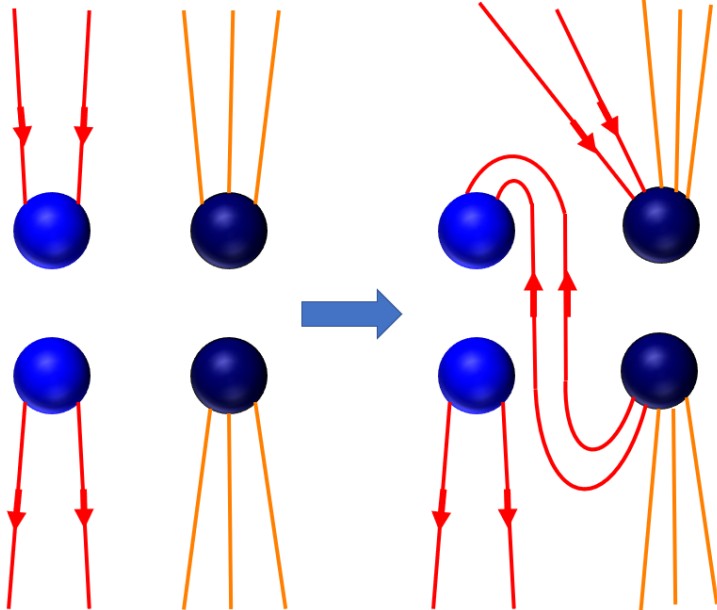

Figure 14: An illustration of the effect of 1-1 handle slide on the Kirby diagram, where the blue 1-handle slides past the darkblue 1-handle. On the anyon diagram the two red lines running upward become a bubble.

under these handle moves, according to our knowledge of the Crane-Yetter model [48, 52]. Here we just need to analyze handle moves which do explicitly involve $G$-defects, and they are either 1-1 handle slides (see Fig. 14), isotopies where some 2-handles cross some defects (see Fig. 15) or 2-2 handle slides (see Fig. 16).

- 1-1 handle slide.

  The effect of a 1-1 handle slide on a Kirby diagram is explicitly shown in Fig. 14. Suppose that before the handle slide, a **g**-defect is present across the blue 1-handle, and an **h**-defect is present across the darkblue 1-handle. Then after the handle slide, an $\mathbf{h}^{-1}\mathbf{g}$ defect is present across the blue 1-handle while an **h** defect is still present across the darkblue 1-handle. We wish to prove the invariance of the partition function $\mathcal{Z}(\mathcal{M}, \mathcal{G})$ by proving the invariance of each individual summand of Eq. (44), which has a given set of labels $\beta$.

  Suppose anyons labeled by $\{a_i\}$ cross the blue 1-handle before the handle slide. Without loss of generality, suppose that they are running downward in the blue 1-handle of the Kirby diagram as in Fig. 14. After the handle slide, anyons $\{a_i\}$ cross both the blue 1-handle and the darkblue 1-handle. Accordingly, the prefactor of quantum dimension in the individual summand of Eq. (44) is modified by an extra $1/\sqrt{\prod_i d_{a_i}}$ after the handle slide. This is canceled by the contribution from the extra bubble in the Kirby diagram, formed e.g., by the two red lines running upward in Fig. 14, as can be seen by using Eq. (4). After accounting for this extra bubble, the contribution of the Kirby diagram is invariant before and after the handle move. Then we just need to analyze the change of the $\eta$-factors and $U$-factors

  For the sake of presentation, let us first assume that all symmetry defects involved are unitary. Now consider the change of the $\eta$-factors. Before the handle slide, $\{a_i\}$ are acted upon by $\rho_{\mathbf{g}}^{-1}$ while after the handle slide, $\{a_i\}$ are acted upon by $\rho_{\mathbf{h}^{-1}\mathbf{g}}^{-1} \circ \rho_{\mathbf{h}}^{-1}$. This

gives an extra $\eta$-factor

$$\prod_i \left(\eta_{a_i}\left(\mathbf{h}, \mathbf{h}^{-1}\mathbf{g}\right)\right)^{-1} . \tag{C.1}$$

Next we consider the change of $U$-factors. Before the handle slide, the vector and the dual vector assigned to the two disconnected $D^3$ balls of the blue 1-handle are $|a_i, \ldots; 1\rangle_{\tilde{\mu}\ldots}$ and $\langle\overline{\mathbf{g}}a_i, \ldots; 1|_{\mu\ldots}$, respectively, which give the $U$-factor from the red lines

$$\langle\overline{\mathbf{g}}a_i, \ldots; 1|_{\mu\ldots}\rho_{\mathbf{g}}^{-1}|a_i, \ldots; 1\rangle_{\tilde{\mu}\ldots} = U_{\mathbf{g}}^{-1}(a_i, \ldots; 1)_{\tilde{\mu}\ldots,\mu\ldots} . \tag{C.2}$$

After the handle slide, the vector assigned to the upper ball of the darkblue 1-handle is the tensor product of $|a_i, \ldots; 1\rangle_{\tilde{\mu}\ldots}$ and the original vector corresponding to the orange lines. The dual vector assigned to the lower ball of the darkblue 1-handle is the tensor product of $\langle\overline{\mathbf{h}}a_i, \ldots; 1|_{\tilde{\tilde{\mu}}\ldots}$ and the original dual vector corresponding to orange lines. The vector assigned to the upper ball of the blue 1-handle is $|\overline{\mathbf{h}}a_i, \ldots; 1\rangle_{\tilde{\tilde{\mu}}\ldots}$, and finally the dual vector assigned to the lower ball of the blue 1-handle is still $\langle\overline{\mathbf{g}}a_i, \ldots; 1|_{\mu\ldots}$. Then the $U$-factor relevant to the red lines after the handle slide becomes

$$\sum_{\tilde{\tilde{\mu}}\ldots}\langle\overline{\mathbf{g}}a_i, \ldots; 1|_{\mu\ldots}\rho_{\mathbf{h}^{-1}\mathbf{g}}^{-1}|\overline{\mathbf{h}}a_i, \ldots; 1\rangle_{\tilde{\tilde{\mu}}\ldots} \cdot \langle\overline{\mathbf{h}}a_i, \ldots; 1|_{\tilde{\tilde{\mu}}\ldots}\rho_{\mathbf{h}}^{-1}|a_i, \ldots; 1\rangle_{\tilde{\mu}\ldots}$$

$$= \sum_{\tilde{\tilde{\mu}}\ldots}U_{\mathbf{h}}^{-1}(a_i, \ldots; 1)_{\tilde{\mu}\ldots,\tilde{\tilde{\mu}}\ldots} \cdot U_{\mathbf{h}^{-1}\mathbf{g}}^{-1}\left(\overline{\mathbf{h}}a_i, \ldots; 1\right)_{\tilde{\tilde{\mu}}\ldots,\mu\ldots} . \tag{C.3}$$

According to Eq. (23), the product of Eq. (C.1) and Eq. (C.3) is precisely Eq. (C.2), which means that the changes of the $\eta$-factors and $U$-factors cancel each other, and each individual summand in Eq. (44) is invariant under 1-1 handle slides.

To account for anti-unitary symmetry, we need to pay attention to two special effects: i) some anyons will change their directions of flow compared with the Kirby diagram; ii) we need to add proper factors of $K^{q(\mathbf{h})}$ in front of some vectors to account for $\mathbb{C}$-anti-linear functor. Without loss of generality, we can still suppose that anyons labeled by $\{a_i\}$ are running downward in the blue 1-handle of the Kirby diagram. Yet we should give these anyons an extra label $\{s_i\}$, according to whether the segment corresponding to labels $\{a_i\}$ has flipped the direction of the flow or not:

$$s_i = \begin{cases} +1, & \text{if } a_i \text{ does not flip}, \\ -1, & \text{if } a_i \text{ does flip}. \end{cases} \tag{C.4}$$

Then after carefully counting the extra contribution from anti-unitary symmetry, the change of $\eta$-factor becomes

$$\prod_i \left(\eta_{a_i}\left(\mathbf{h}, \mathbf{h}^{-1}\mathbf{g}\right)\right)^{-s_i} . \tag{C.5}$$

Before the handle slide, the vector and the dual vector assigned to the two disconnected $D^3$ balls of the blue 1-handle are $|a_i, \ldots, \{s_i = +1\}; a_{\tilde{\imath}}, \ldots, \{s_{\tilde{\imath}} = -1\}\rangle_{\tilde{\mu}\ldots}$ and $\langle\overline{\mathbf{g}}a_i, \ldots, \{s_i = +1\}; \overline{\mathbf{g}}a_{\tilde{\imath}}, \ldots, \{s_{\tilde{\imath}} = -1\}|_{\mu\ldots}K^{q(\mathbf{g})}$, which gives the $U$-factor from the red lines

$$\langle\overline{\mathbf{g}}a_i, \ldots; \overline{\mathbf{g}}a_{\tilde{\imath}}, \ldots\ldots|_{\mu\ldots}K^{q(\mathbf{g})}\rho_{\mathbf{g}}^{-1}|a_i, \ldots; a_{\tilde{\imath}}, \ldots\rangle_{\tilde{\mu}\ldots} = U_{\mathbf{g}}^{-1}(a_i, \ldots; a_{\tilde{\imath}}, \ldots)_{\tilde{\mu}\ldots,\mu\ldots} , \tag{C.6}$$

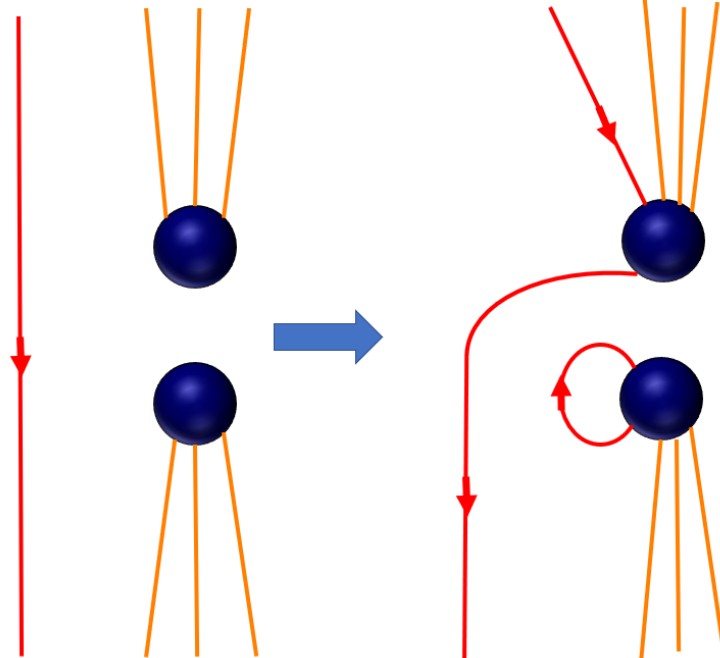

Figure 15: An illustration of the effect of isotopies where the red 2-handle crosses some defect on the darkblue 1-handle. On the anyon diagram the red line connecting the lower darkblue ball to itself becomes a red bubble.

where $i$ and $\tilde{i}$ are indices to label anyons with $s_i = +1$ or $s_i = -1$, respectively. After the handle slide, the $U$-factor relevant to the red lines after the handle slide becomes

$$
\sum_{\tilde{\mu}\ldots} \langle {}^{\overline{g}}a_i, \ldots; {}^{\overline{g}}a_{\tilde{i}}, \ldots |_{\mu\ldots} K^{q(\mathbf{g})} \rho_{\mathbf{h}^{-1}\mathbf{g}}^{-1} K^{q(\mathbf{h})} | {}^{\overline{h}}a_i, \ldots; {}^{\overline{h}}a_{\tilde{i}}, \ldots \rangle_{\tilde{\mu}\ldots}
$$

$$
\times \langle {}^{\overline{h}}a_i, \ldots; {}^{\overline{h}}a_{\tilde{i}}, \ldots |_{\tilde{\mu}\ldots} K^{q(\mathbf{h})} \rho_{\mathbf{h}}^{-1} | a_i, \ldots; a_{\tilde{i}}, \ldots \rangle_{\tilde{\mu}\ldots}
$$

$$
= \sum_{\tilde{\mu}\ldots} U_{\mathbf{h}}^{-1} \left( a_i, \ldots; a_{\tilde{i}}, \ldots \right)_{\tilde{\mu}\ldots, \tilde{\mu}\ldots} \cdot U_{\mathbf{h}^{-1}\mathbf{g}}^{-1} \left( {}^{\overline{h}}a_i, \ldots; {}^{\overline{h}}a_{\tilde{i}}, \ldots \right)_{\tilde{\mu}\ldots, \mu\ldots}^{\sigma(\mathbf{h})}. \tag{C.7}
$$

Again, according to Eq. (23), the product of Eq. (C.5) and Eq. (C.7) is precisely Eq. (C.6), which means that the changes of the $\eta$-factors and $U$-factors cancel each other, and each individual summand in Eq. (44) is invariant under 1-1 handle slides.

Therefore, we have established that the partition function $\mathcal{Z}(\mathcal{M}, \mathcal{G})$ is invariant under the 1-1 handle slide.

- Isotopy.

  The invariance of the partition function $\mathcal{Z}(\mathcal{M}, \mathcal{G})$ under an isotopy where a 2-handle crosses a defect is relatively easy. Suppose that a $\mathbf{g}$-defect is present across the darkblue 1-handle (see Fig. 15). We wish to prove the invariance of the partition function $\mathcal{Z}(\mathcal{M}, \mathcal{G})$ by proving the invariance of each individual summand of Eq. (44). Label the red 2-handle by an anyon $a$. After this isotopy, the prefactor of quantum dimension in the individual summand of Eq. (44) is modified by an extra $1/d_a$. This is canceled by the contribution from the extra bubble in the Kirby diagram. There is no change in $\eta$-factors and $U$-factors. Therefore, we see that $\mathcal{Z}(\mathcal{M}, \mathcal{G})$ is invariant under the isotopy.

- 2-2 handle slide.

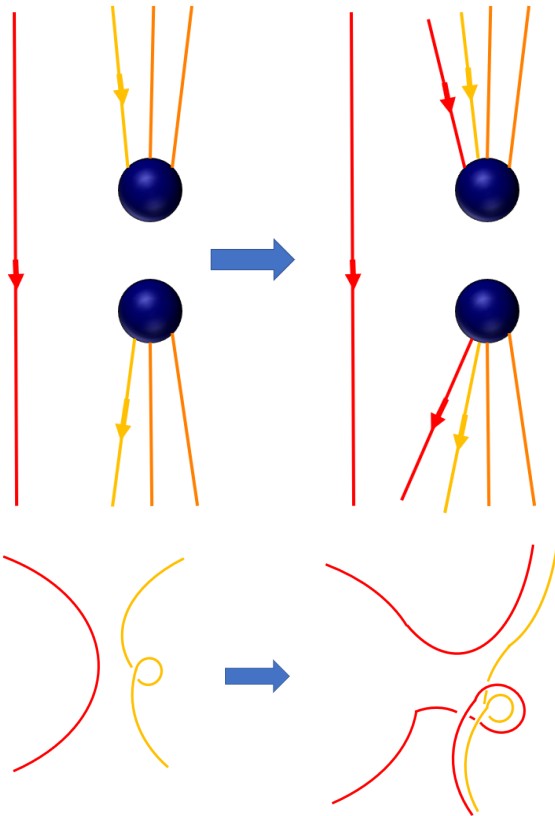

Figure 16: Upper: An illustration of the effect of a 2-2 handle slide on the Kirby diagram, where the red 2-handle slides past the yellow 2-handle. Lower: An illustration of the change of framing after 2-2 handle slide.

The effect of a 2-2 handle slide on a Kirby diagram is explicitly shown in Fig. 16. We wish to prove the invariance of the partition function $\mathcal{Z}(\mathcal{M}, \mathcal{G})$ by proving the invariance of each individual summand of Eq. (44).

Let us put anyon $a$ on (a segment of) the red 2-handle and anyon $b$ on the yellow 2-handle. The strategy is to fuse $a$ and $b$ into another anyon $c$, such that the expression involving $a$ and $b$ can be transformed to an expression involving $a$ and $c$, which turns out to be manifestly the same as the expression before the 2-2 handle slide.

First of all, at every 1-handle that the yellow 2-handle crosses, the relevant prefactor involving quantum dimensions is shifted from $\frac{1}{\sqrt{d_b}}$ to $\frac{1}{\sqrt{d_a d_b}}$. Yet when we fuse $a$ and $b$ into another anyon $c$, there is an extra bubble as in Eq. (4) which gives $\sqrt{\frac{d_a d_b}{d_c}}$. Multiplying them together gives $\frac{1}{\sqrt{d_c}}$. Therefore, the factors regrading quantum dimensions match before and after the handle slide.

Next, we should consider the effect of the change of framing, as illustrated in the lower figure of Fig. 16. Suppose that before the handle slide $b$ has framing $n$. This contributes a $\theta_b^n$ term in $\mathcal{Z}(\mathcal{M}, \mathcal{G})$. After the handle slide, the red 2-handle should have an additional framing $n$, the yellow 2-handle should still have framing $n$, and the red 2-handle should wind around the yellow 2-handle $(-n)$-times, contributing an extra factor of

$$\theta_a^n \theta_b^n \times \frac{\theta_c^n}{\theta_a^n \theta_b^n} = \theta_c^n, \tag{C.8}$$

consistent with the expression before the handle slide.

Now we consider $U$- and $\eta$-factors. Suppose that starting with the segment of the yellow 2-handle labeled by $b$, the yellow 2-handle begins crossing some 1-handles, and the symmetry defects these 1-handles host are $\mathbf{g}, \mathbf{h}, \dots$. Moreover, without loss of generality suppose that on the Kirby diagram the yellow 2-handle runs downward in these 1-handles. Then consider two consecutive 1-handles that the yellow 2-handle crosses with $\mathbf{g}$, $\mathbf{h}$-defects on them, the extra $U$-factor involved is

$$U_{\mathbf{g}}^{-1}(a, b; c)_{\mu\mu''} U_{\mathbf{h}}^{-1}(\overline{\mathbf{g}}a, \overline{\mathbf{g}}b; \overline{\mathbf{g}}c)_{\mu''\mu'}^{\sigma(\mathbf{g})}. \tag{C.9}$$

The $\eta$-factor coming from composing $\rho_{\mathbf{h}}^{-1}$ and $\rho_{\mathbf{g}}^{-1}$ to $\rho_{\mathbf{gh}}^{-1}$, is

$$\eta_a(\mathbf{g}, \mathbf{h})^{-1} \eta_b(\mathbf{g}, \mathbf{h})^{-1}. \tag{C.10}$$

Multiplying Eq. (C.9) and Eq. (C.10) and using Eq. (23), we get

$$\eta_c(\mathbf{g}, \mathbf{h})^{-1} U_{\mathbf{gh}}^{-1}(a, b; c)_{\mu\mu'}. \tag{C.11}$$

It is then straightforward to see that, after accounting for all 1-handles that the yellow 2-handle crosses, such manipulation will cancel all extra $U$-factors while reproducing the correct $\eta$-factor from the natural isomorphism for anyon $c$.

Therefore, we have established that the partition function $\mathcal{Z}(\mathcal{M}, \mathcal{G})$ is invariant under the 2-2 handle slide.

## C.2 Invariance under change of defects

The partition function $\mathcal{Z}(\mathcal{M}, \mathcal{G})$ is also dependent on a specific choice of the defect network defined on $\mathcal{M}$ to reflect the $G$-bundle structure $\mathcal{G}$. There are two important choices that we have made during calculation in Eq. (44). The first choice is that, for a 1-handle hosting a $\mathbf{g}$-defect, there are two $D^3$ balls on the attaching region, and we need to choose one $D^3$ ball out of the two to be "above" another one, as in Remark e in Sec. 3.4. It amounts to choosing an orientation of the defect, i.e., whether this defect is $\mathbf{g}$ or $\overline{\mathbf{g}}$. Another choice is that even for the same $G$-bundle $\mathcal{G}$ on $\mathcal{M}$, we can choose a different set of defects put across 1-handles by changing each defect $\mathbf{g}$ to $\mathbf{hgh}^{-1}$, where $\mathbf{h}$ is an arbitrary fixed element in $G$. Remember that $G$-bundles on $\mathcal{M}$ are classified by

$$\text{Hom}(\pi_1(\mathcal{M}), G)/G. \tag{C.12}$$

Here $\text{Hom}(\pi_1(\mathcal{M}), G)$ is nothing but identifying the holonomy we put on noncontractible cycles, yet there is an equivalence relation due to $G$-action by conjugation on the holonomy. Therefore, we need to prove that the partition function $\mathcal{Z}(\mathcal{M}, \mathcal{G})$ is the same if the defect we put on 1-handles are conjugated by elements in $G$. This amounts to showing the gauge invariance of the partition function under gauge transformation of $G$.

Let us start by considering the first choice, i.e., the choice of the orientation of the defect. Again, we wish to prove the invariance of the partition function $\mathcal{Z}(\mathcal{M}, \mathcal{G})$ by proving the invariance of each individual summand of Eq. (44). Suppose anyons labeled by $\{a_i\}$ cross the blue 1-handle which hosts a defect labeled by $\mathbf{g}$. Without loss of generality suppose that at the beginning they are all running downward in the blue 1-handle of the Kirby diagram. Now we flip the relative position of the two balls. Then anyons crossing the 1-handles upwards are acted upon by $\rho_{\overline{\mathbf{g}}}$ instead of $\rho_{\mathbf{g}}^{-1}$, which gives the extra $\eta$-factor

$$\prod_i \left( \eta_{a_i}(\mathbf{g}, \overline{\mathbf{g}}) \right)^{s_i}. \tag{C.13}$$

Again, to account for the fact that some anyons will flip the direction of the flow, we introduce an extra factor $s_i$ as in Eq. (C.4). Before the flip, the vector and the dual vector assigned to the two disconnected $D^3$ balls of the blue 1-handle are $|a_i, \ldots, \{s_i = +1\}; a_{\tilde{i}}, \ldots, \{s_{\tilde{i}} = -1\}\rangle_{\tilde{\mu}\ldots}$ and $\langle {}^{\bar{g}}a_i, \ldots, \{s_i = +1\}; {}^{\bar{g}}a_{\tilde{i}}, \ldots, \{s_{\tilde{i}} = -1\}|_{\mu\ldots} K^{q(g)}$, which gives the $U$-factor

$$\langle {}^{\bar{g}}a_i, \ldots; {}^{\bar{g}}a_{\tilde{i}}, \ldots \ldots |_{\mu\ldots} K^{q(g)} \rho_g^{-1} |a_i, \ldots; a_{\tilde{i}}, \ldots\rangle_{\tilde{\mu}\ldots} = U_g^{-1}\left(a_i, \ldots; a_{\tilde{i}}, \ldots\right)_{\tilde{\mu}\ldots,\mu\ldots}, \qquad (C.14)$$

where again $i$ and $\tilde{i}$ are indices to label anyons with $s_i = +1$ or $s_{\tilde{i}} = -1$, respectively.

After the flip, the vector and the dual vector assigned to the two disconnected $D^3$ balls of the blue 1-handle are $K^{q(g)}|{}^{\bar{g}}a_{\tilde{i}}, \ldots, \{s_{\tilde{i}} = -1\}; {}^{\bar{g}}a_i, \ldots, \{s_i = +1\}\rangle_{\mu\ldots}$ and $\langle a_{\tilde{i}}, \ldots, \{s_{\tilde{i}} = -1\}; a_i, \ldots, \{s_i = +1\}|_{\tilde{\mu}\ldots}$, which gives the $U$-factor

$$\langle a_{\tilde{i}}, \ldots; a_i, \ldots \ldots |_{\tilde{\mu}\ldots} \rho_{\bar{g}}^{-1} K^{q(g)} |{}^{\bar{g}}a_{\tilde{i}}, \ldots; {}^{\bar{g}}a_i, \ldots\rangle_{\mu\ldots} = U_{\bar{g}}^{-1}\left({}^{\bar{g}}a_{\tilde{i}}, \ldots; {}^{\bar{g}}a_i, \ldots\right)_{\mu\ldots,\tilde{\mu}\ldots}^{\sigma(g)}$$
$$= U_{\bar{g}}\left({}^{\bar{g}}a_i, \ldots; {}^{\bar{g}}a_{\tilde{i}}, \ldots\right)_{\tilde{\mu}\ldots,\mu\ldots}^{\sigma(g)}. \qquad (C.15)$$

According to Eq. (23), the product of Eq. (C.13) and Eq. (C.15) is precisely Eq. (C.14). Therefore, we have established that the partition function $\mathcal{Z}(\mathcal{M}, \mathcal{G})$ is independent of the first choice.

Now consider the second choice. Suppose that all defects are conjugated by an element $\mathbf{h}$ in $G$, i.e., all $\mathbf{g}$-defects become $\mathbf{hgh}^{-1}$. We need to consider the case where $\mathbf{h}$ is unitary or anti-unitary separately.

Suppose that $\mathbf{h}$ is unitary. First we change the labels $\{a_i\}$ of all anyons to $\{{}^{\mathbf{h}}a_i\}$. Then we wish to prove the invariance of the partition function $\mathcal{Z}(\mathcal{M}, \mathcal{G})$ by proving the invariance of each individual summand of Eq. (44). First consider the Kirby diagram, whose evaluation schematically takes the form

$$\left(F \cdot R \cdot F \cdot R \cdots\right)_{\mu_1 \ldots \mu_2 \ldots, \tilde{\mu}_1 \ldots \tilde{\mu}_2 \ldots \cdots}, \qquad (C.16)$$

where $\mu_1 \ldots$ and $\tilde{\mu}_1 \ldots$ are indices corresponding to vectors and dual vectors on the anyon diagram associated to the Kirby diagram, respectively. After relabeling, according to Eq. (19) the Kirby diagram changes to

$$\sum_{\mu_1' \ldots \tilde{\mu}_1' \ldots} U_{\mu_1 \mu_1'}^{-1} \ldots U_{\mu_2 \mu_2'}^{-1} \cdots \cdot \left(F \cdot R \cdot F \cdot R \cdots\right)_{\mu_1' \ldots \mu_2' \ldots \cdots, \tilde{\mu}_1' \ldots \tilde{\mu}_2' \ldots \cdots} \cdot U_{\tilde{\mu}_1' \tilde{\mu}_1} \ldots U_{\tilde{\mu}_2' \tilde{\mu}_2} \ldots \qquad (C.17)$$

We have suppressed all anyon labels, but pay attention that in the above formula anyon labels in $F$- and $R$- symbols are from $\{a_i\}$ while anyon labels in $U$-symbols are from $\{{}^{\mathbf{h}}a_i\}$.

Now we focus on a single 1-handle. Suppose anyons labeled by $\{a_i\}$ cross the 1-handle which hosts a defect labeled by $\mathbf{g}$, and without loss of generality suppose that they are all running downward in the Kirby diagram. Before the change of defects, the vector and the dual vector assigned to the two disconnected $D^3$ balls are $|a_i, \ldots, \{s_i = +1\}; a_{\tilde{i}}, \ldots, \{s_{\tilde{i}} = -1\}\rangle_{\tilde{\mu}\ldots}$ and $\langle {}^{\bar{g}}a_i, \ldots, \{s_i = +1\}; {}^{\bar{g}}a_{\tilde{i}}, \ldots, \{s_{\tilde{i}} = -1\}|_{\mu\ldots} K^{q(g)}$, which gives the $U$-factor

$$\langle {}^{\bar{g}}a_i, \ldots; {}^{\bar{g}}a_{\tilde{i}}, \ldots \ldots |_{\mu\ldots} K^{q(g)} \rho_g^{-1} |a_i, \ldots; a_{\tilde{i}}, \ldots\rangle_{\tilde{\mu}\ldots} = U_g^{-1}\left(a_i, \ldots; a_{\tilde{i}}, \ldots\right)_{\tilde{\mu}\ldots,\mu\ldots}, \qquad (C.18)$$

where again $i$ and $\tilde{i}$ are indices to label anyons with $s_i = +1$ or $s_{\tilde{i}} = -1$, respectively. After the change of defects (and relabeling), the vector and the dual vector assigned to the two disconnected $D^3$ balls are $|{}^{\mathbf{h}}a_i, \ldots, \{s_i = +1\}; {}^{\mathbf{h}}a_{\tilde{i}}, \ldots, \{s_{\tilde{i}} = -1\}\rangle_{\tilde{\mu}\ldots}$ and $\langle {}^{\mathbf{h}\bar{g}}a_i, \ldots, \{s_i = +1\}; {}^{\mathbf{h}\bar{g}}a_{\tilde{i}}, \ldots, \{s_{\tilde{i}} = -1\}|_{\mu\ldots} K^{q(g)}$, which gives the $U$-factor

$$\langle \, ^{\mathbf{h\bar{g}}}a_i, \dots; \, ^{\mathbf{h\bar{g}}}a_{\tilde{i}}, \dots \dots |_{\mu \dots} K^{q(\mathbf{g})} \rho^{-1}_{\mathbf{hgh}^{-1}} | \, ^{\mathbf{h}}a_i, \dots; \, ^{\mathbf{h}}a_{\tilde{i}}, \dots \rangle_{\tilde{\mu} \dots}$$
$$= U^{-1}_{\mathbf{hgh}^{-1}} \left( \, ^{\mathbf{h}}a_i, \dots; \, ^{\mathbf{h}}a_{\tilde{i}}, \dots \right)_{\tilde{\mu} \dots, \mu \dots} . \tag{C.19}$$

Together with the extra $U$-factor in Eq. (C.17), we have

$$\sum_{\mu' \tilde{\mu}'} U_{\mathbf{h}}(\, ^{\mathbf{h}}a_i, \dots; \, ^{\mathbf{h}}a_{\tilde{i}}, \dots)_{\tilde{\mu}\tilde{\mu}'} \cdot U^{-1}_{\mathbf{hgh}^{-1}}(\, ^{\mathbf{h}}a_i, \dots; \, ^{\mathbf{h}}a_{\tilde{i}}, \dots)_{\tilde{\mu}'\mu'} \cdot U^{-1}_{\mathbf{h}}(\, ^{\mathbf{h\bar{g}}}a_i, \dots; \, ^{\mathbf{h\bar{g}}}a_{\tilde{i}}, \dots)^{\sigma(\mathbf{g})}_{\mu'\mu} . \tag{C.20}$$

Finally, after conjugation by $\mathbf{h}$, anyons are now labeled by $\{\, ^{\mathbf{h}}a_i\}$, and acted by $\rho^{-1}_{\mathbf{hgh}^{-1}}$. Comparing with $\rho_{\mathbf{h}} \circ \rho^{-1}_{\mathbf{g}} \circ \rho^{-1}_{\mathbf{h}}$, this gives an extra $\eta$-factor to be

$$\prod_i \left( \frac{\eta_{\, ^{\mathbf{h}}a_i}(\mathbf{h}, \mathbf{g})}{\eta_{\, ^{\mathbf{h}}a_i}(\mathbf{hgh}^{-1}, \mathbf{h})} \right)^{s_i} . \tag{C.21}$$

According to Eq. (23), the product of Eq. (C.21) and Eq. (C.20) is precisely Eq. (C.18), which means that each individual summand in Eq. (44) is invariant.

Finally, suppose that $\mathbf{h}$ is anti-unitary. Then conjugation by $\mathbf{h}$ needs to be accompanied by change of orientation of the manifold $\mathcal{M}$.[14] The partition function $\mathcal{Z}(\mathcal{M}, \mathcal{G})$ is complex conjugated under the change of orientation. The rest analysis is similar to the case where $\mathbf{h}$ is unitary. Compared with Eq. (C.16) and according to Eq. (19), after relabeling the Kirby diagram changes to

$$\sum_{\mu'_1 \dots \tilde{\mu}'_1 \dots} \left( U^{-1}_{\mu_1 \mu'_1} \dots U^{-1}_{\mu_2 \mu'_2} \dots \right)^* \cdot \left( F \cdot R \cdot F \cdot R \cdots \right)_{\mu'_1 \dots \mu'_2 \dots \cdots, \tilde{\mu}'_1 \dots \tilde{\mu}'_2 \dots \cdots} \cdot \left( U_{\tilde{\mu}'_1 \tilde{\mu}_1} \dots U_{\tilde{\mu}'_2 \tilde{\mu}_2} \dots \right)^* . \tag{C.22}$$

Again pay attention that in the above formula anyon labels in $F$- and $R$- symbols are from $\{a_i\}$ while anyon labels in $U$- symbols are from $\{\, ^{\mathbf{h}}a_i\}$. Then focus on a single 1-handle. Again suppose anyons labeled by $\{a_i\}$ cross the 1-handle which hosts a defect labeled by $\mathbf{g}$, and without loss of generality suppose that they are all running downward in the Kirby diagram. After the change of defects (and relabeling), together with the extra $U$-factor in Eq. (C.22), the $U$-factor relevant to the 1-handle becomes

$$\sum_{\mu' \tilde{\mu}'} U_{\mathbf{h}}(\, ^{\mathbf{h}}a_i, \dots; \, ^{\mathbf{h}}a_{\tilde{i}}, \dots)^*_{\tilde{\mu}\tilde{\mu}'} \cdot U^{-1}_{\mathbf{hgh}^{-1}}(\, ^{\mathbf{h}}a_i, \dots; \, ^{\mathbf{h}}a_{\tilde{i}}, \dots)^*_{\tilde{\mu}'\mu'} \cdot U^{-1}_{\mathbf{h}}(\, ^{\mathbf{h\bar{g}}}a_i, \dots; \, ^{\mathbf{h\bar{g}}}a_{\tilde{i}}, \dots)^{*\sigma(\mathbf{g})}_{\mu'\mu} . \tag{C.23}$$

Finally, after conjugation by $\mathbf{h}$, anyons are now labeled by $\{\, ^{\mathbf{h}}a_i\}$, and acted by $\rho^{-1}_{\mathbf{hgh}^{-1}}$. Comparing with $\rho_{\mathbf{h}} \circ \rho^{-1}_{\mathbf{g}} \circ \rho^{-1}_{\mathbf{h}}$, this gives an extra $\eta$-factor to be

$$\prod_i \left( \frac{\eta_{\, ^{\mathbf{h}}a_i}(\mathbf{h}, \mathbf{g})}{\eta_{\, ^{\mathbf{h}}a_i}(\mathbf{hgh}^{-1}, \mathbf{h})} \right)^{-s_i} . \tag{C.24}$$

According to Eq. (23), the product of Eq. (C.24) and Eq. (C.23) is precisely Eq. (C.18), which means that each individual summand in Eq. (44) is invariant.

Therefore, we have established that the partition function $\mathcal{Z}(\mathcal{M}, \mathcal{G})$ is invariant under the change of defects.

---

[14]For $\mathcal{M}$ orientable the meaning is clear. For $\mathcal{M}$ non-orientable we need to choose an orientation of $T\mathcal{M} \oplus \xi$, where $T\mathcal{M}$ is the tangent bundle of $\mathcal{M}$, and $\xi$ is the associated 1-dimensional vector bundle $\xi$ of $\mathcal{G}$, as explained in Appendix D.

## C.3   Gauge invariance

Another important check we need to perform is the "gauge invariance" of Eq. (44). Specifically, we need to prove that Eq. (44) is invariant under vertex basis transformation Eqs. (10) and (18), as well as symmetry action gauge transformation Eq. (26). In this sub-appendix, we explicitly perform this check.

To show the invariance under vertex basis transformation, we can think of the result of the Kirby diagram as a giant matrix, which schematically takes the form

$$\left( F \cdot R \cdot F \cdot R \cdots \right)_{\mu_1...\mu_2...\cdots,\tilde{\mu}_1...\tilde{\mu}_2...\cdots} , \tag{C.25}$$

where $\mu_1 \ldots$ and $\tilde{\mu}_1 \ldots$ are indices corresponding to vectors and dual vectors on the anyon diagram associated to the Kirby diagram, respectively. Then we can schematically write down what the giant matrix Eq. (C.25) becomes after vertex basis transformation Eq. (10), which is

$$\left(\Gamma_{\cdot}^{\cdot\cdot}\right)_{\mu_1\mu_1'} \cdots \left(\Gamma_{\cdot}^{\cdot\cdot}\right)_{\mu_2\mu_2'} \cdots \times \left( F \cdot R \cdot F \cdot R \right)_{\mu_1'...\mu_2'...\cdots,\tilde{\mu}_1'...\tilde{\mu}_2'...\cdots} \times \left(\Gamma_{\cdot}^{\cdot\cdot}\right)_{\tilde{\mu}_1'\tilde{\mu}_1}^{\dagger} \cdots \left(\Gamma_{\cdot}^{\cdot\cdot}\right)_{\tilde{\mu}_2'\tilde{\mu}_2}^{\dagger} \cdots \tag{C.26}$$

On the 1-handles, we have $U$-factors $U_{\mathbf{g}}^{-1}(\ldots)_{\tilde{\mu}_1...,\mu_1...}$, which under vertex basis transformation transforms according to Eq. (18), i.e., it becomes

$$\left(\Gamma_{\cdot}^{\cdot\cdot}\right)_{\tilde{\mu}_1\tilde{\mu}_1''} \cdots \times U_{\mathbf{g}}^{-1}(\ldots)_{\tilde{\mu}_1''...,\mu_1''...} \times \left(\left(\Gamma_{\cdot}^{\cdot\cdot}\right)_{\mu_1''\mu_1}^{\dagger}\right)^* . \tag{C.27}$$

Here we substitute all anyon labels by $\cdot$ and hopefully they will be clear in specific contexts. Now we immediately see that after multiplying $\Gamma$-matrices and summing over $\mu$, $\tilde{\mu}$ indices, we have $\delta_{\mu_1'\mu_1''} \cdots \delta_{\tilde{\mu}_1'\tilde{\mu}_1''} \cdots$ and the expression becomes the original expression.

Next consider symmetry action gauge transformation. Again let us focus on a single 1-handle, and without loss of generality suppose that all anyons $\{a_i\}$ crossing the 1-handle are running downward in the Kirby diagram. Again, to account for the fact that some anyons will flip the direction of the flow, we introduce an extra factor $s_i$ as in Eq. (C.4). The vector and the dual vector assigned to the two disconnected $D^3$ balls are $|a_i, \ldots, \{s_i = +1\}; a_{\tilde{i}}, \ldots, \{s_{\tilde{i}} = -1\}\rangle_{\tilde{\mu}...}$ and $^{\overline{g}}\langle a_i, \ldots, \{s_i = +1\}; {}^{\overline{g}}a_{\tilde{i}}, \ldots, \{s_{\tilde{i}} = -1\}|_{\mu...}K^{q(\mathbf{g})}$, which gives the $U$-factor

$$\langle {}^{\overline{g}}a_i, \ldots; {}^{\overline{g}}a_{\tilde{i}}, \ldots \ldots |_{\mu...}K^{q(\mathbf{g})}\rho_{\mathbf{g}}^{-1}|a_i, \ldots; a_{\tilde{i}}, \ldots\rangle_{\tilde{\mu}...} = U_{\mathbf{g}}^{-1}\left(a_i, \ldots; a_{\tilde{i}}, \ldots\right)_{\tilde{\mu}...,\mu...} , \tag{C.28}$$

where $i$ and $\tilde{i}$ are indices to label anyons with $s_i = +1$ or $s_i = -1$, respectively. Under the transformation in Eq. (26), it becomes

$$\prod_i \left(\gamma_{a_i}(\mathbf{g})\right)^{-s_i} U_{\mathbf{g}}^{-1}\left(a_i, \ldots; a_{\tilde{i}}, \ldots\right)_{\tilde{\mu}...,\mu...} . \tag{C.29}$$

All $\{a_i\}$ crossing the 1-handle are acted by $\rho_{\mathbf{g}}^{-1}$, and therefore following Eq. (25) under symmetry action gauge transformation we have extra $\gamma$ parts:

$$\prod_i \left(\gamma_{a_i}(\mathbf{g})\right)^{s_i} . \tag{C.30}$$

Then we immediately see that the extra $\gamma$ part in Eq. (C.30) exactly cancels the extra $\gamma$ part in Eq. (C.29).

Therefore, we have established that the partition function $\mathcal{Z}(\mathcal{M}, \mathcal{G})$ is invariant under vertex basis transformation Eqs. (10),(18) and symmetry action gauge transformation Eq. (26).

## C.4  Cobordism invariance

In order to demonstrate that the construction gives the TQFT that reflects the anomaly of the symmetry-enriched topological order, in this sub-appendix we prove that the partition function in Eq. (44) on a closed 4-manifold $\mathcal{M}$ with $G$-bundle structure $\mathcal{G}$ is in fact a cobordism invariant.

Two closed, oriented $n$-dimensional manifolds $\mathcal{M}$ and $\tilde{\mathcal{M}}$ are cobordant if and only if they are related to each other by a sequence of surgeries [80]. In 4 dimensions, cobordisms of 4-manifolds can be generated by the following types of surgery moves:

- Removing or adding an $S^4$.

- Replacing an $S^1 \times D^3$ by $S^2 \times D^2$ and vice versa. Note that they have the same boundary $S^1 \times S^2$.

- Replacing $S^0 \times D^4$ by $D^1 \times S^3$ and vice versa. Note that they have the same boundary $S^0 \times S^3$.

In the Crane-Yetter model, in order to prove that the partition function is a cobordism invariant, we need to prove that the pratition function $\mathcal{Z}(\mathcal{M})$ is invariant under these three surgery moves.

Now in the presence of $G$-bundle structure $\mathcal{G}$, we need to consider $G$-bordism [71,88], and therefore we need to pay special attention to whether defects can be extended or not during the surgery. Let us enumerate the effect of the three surgery moves one by one.

- Removing or adding an $S^4$.

  This surgery move will not involve $G$-defects because $S^4$ is simply connected (i.e., $\pi_1(S^4) = 0$) and a $G$-bundle on it must be trivial. Then the partition function is invariant because we can directly see from Eq. (44) that $\mathcal{Z}(S^4) = 1$.

- Replacing an $S^1 \times D^3$ by $S^2 \times D^2$ and vice versa.

  We can interpret this surgery move as "trading" a 1-handle with a 2-handle as follows. Decompose $S^1$ into $S^1_+$ and $S^1_-$ which are both homeomorphic to $D^1$. Interpret the $S^1_+ \times D^3$ part as a 1-handle that is attached to $S^1_- \times D^3$ which is interpreted as a 0-handle. (Now we do not assume that there is only 1 0-handle.) Similarly, decompose $S^2$ into $S^2_+$ and $S^2_-$ which are all homeomorphic to $D^2$. Interpret the $S^2_+ \times D^2$ part as a 2-handle that is attached to $S^2_- \times D^2$ which is interpreted as a 0-handle. Then before and after the surgery move that replaces an $S^1 \times D^3$ by $S^2 \times D^2$, a 1-handle is removed and a 2-handle is added.

  An important observation is that for a $G$-bordism, there can be no $G$-defect that is put on the 1-handle before the trading. This fact can be proven as follows. Consider $S^1 \times \{pt\} \subset S^1 \times \partial(D^3) \cong S^1 \times S^2$, which is a loop that survives before and after the trading. Even though it can be a noncontractible loop before the trading, it is a contractible loop after the trading, because we can shrink it to a point as it lives on the boundary of the 2-handle $D^2 \times D^2$. Therefore, $G$-bordism demands that no **g**-defect can be present on such $S^1$ loop.

  It is immediate that now we can carry over the proof of the invariance in the Crane-Yetter model [48] to prove the invariance of $\mathcal{Z}(\mathcal{M}, \mathcal{G})$ under the surgery move.

- Replacing $S^0 \times D^4$ by $D^1 \times S^3$ and vice versa.

  We can interpret this surgery move as adding a 1-handle and removing a 4-handle. Note that we can introduce some $G$-defect as well, including crosscap, to the newly introduced non-contractible cycle.

To consider the effect of this surgery move, we can choose a handle decomposition such that $S^0 \times D^4$ before the surgery move are two 4-handles. Then $D^1 \times S^3$ can be thought of as a 1-handle and a 4-handle. In particular, no 2-handle is attached to the newly introduced 1-handle. We can also directly see this by noting that the cycle corresponding to the newly introduced 1-handle is a free generator in $\pi_1(\tilde{\mathcal{M}})$, therefore we can make handle moves such that no 2-handle touches the 1-handle. It is then straightforward to see that the partition function is invariant under the handle move just by inspecting Eq. (44). Namely, after replacing $S^0 \times D^4$ by $D^1 \times S^3$, $N_4$ is decreased by 1 while $N_1$ is increased by 1. Moreover, all other factors do not change. Therefore, the partition function $\mathcal{Z}(\mathcal{M}, \mathcal{G})$ is invariant under the surgery move.

Therefore, we have established that the partition function $\mathcal{Z}(\mathcal{M}, \mathcal{G})$ is invariant under $G$-bordism.

## C.5  Invertibility

A TQFT is invertible if on every closed 4-manifold the partition function is a pure phase factor, and on every closed 3-manifold $\mathcal{N}$ the associated vector space $\mathcal{V}(\mathcal{N})$ is one-dimensional. We prove that Eq. (44) is indeed a pure phase factor on closed 4-manifolds, due to the cobordism invariance proved previously. To see it, first note that there is a $\mathbb{Z}$ piece in $\Omega_4^{SO}(BG)$ when $G$ contains unitary symmetry only, the generator of which is $\mathbb{CP}^2$ with trivial $G$-bundle structure on it. Moreover, the partition function $\mathcal{Z}(\mathbb{CP}^2)$ is a pure phase factor (see Eq. (47)). If $G$ is finite, besides the $\mathbb{Z}$ piece in $\Omega_4^{SO}(BG)$ when $G$ contains unitary symmetry only, all elements in the relevant bordism group are torsion elements. Accordingly, several copies of $\mathcal{M}$ together with $G$-bundle structure $\mathcal{G}$ on it have to be bordant to $S^4$ with trivial $G$-bundle structure on it. Since $\mathcal{Z}\left(S^4\right) = 1$ from Appendix A.4, the norm of $\mathcal{Z}(\mathcal{M}, \mathcal{G})$ has to be 1. This further means that the anomaly indicators we calculate have norm 1, which is not at all obvious from the explicit formulae, as given by, for example, Eqs. (50),(53),(55). Now we see that they actually take values only in $\pm 1$.

As a side remark, according to a theorem by Freed and Teleman from Ref. [95] (see footnote 10 therein), in order for a fully-extended TQFT to be invertible, we just need to prove that $\mathcal{Z}(S^4)$ is nonzero and $\mathcal{V}(S^1 \times S^2)$ as well as $\mathcal{V}(S^3)$ are all 1-dimensional. They are all straightforward to check for the theory proposed in Sec. 3.2.

## C.6  Generalization to connected Lie groups

In this sub-appendix, we generalize the consideration to connected Lie groups $G$, and prove that the partition function $\mathcal{Z}(\mathcal{M}, \mathcal{G})$ in Eq. (65), defined on a target manifold $\mathcal{M}$ together with a $G$-bundle structure $\mathcal{G}$ on it with associated map $f : \mathcal{M} \to BG$, is a well-defined partition function as well. Given the results from the Crane-Yetter model, we just need to focus on $\eta$-factors, which greatly simplifies the analysis.

- Independence on the handle decomposition

  In order to prove the independence of the partition function $\mathcal{Z}(\mathcal{M}, \mathcal{G})$ on the handle decomposition, again we just need to show its invariance under all handle moves. Moreover, we also just need to focus on the handle moves that explicitly alter $\eta$-factors. Such handle moves contain 2-2 handle slides only, which are illustrated in Fig. 16.

  Let us put anyon $a$ on the red 2-handle and anyon $b$ on the yellow 2-handle. Suppose that the yellow 2-handle corresponds to a 2-chain $[h]$ of $\mathcal{M}$. Then if we fuse $a$ and $b$ into another anyon $c$, the extra $\eta$-factors for the red 2-handle and the yellow 2-handle

become

$$M_{a,\mathbf{w}(f_*[h])}M_{b,\mathbf{w}(f_*[h])} = M_{c,\mathbf{w}(f_*[h])}\,, \tag{C.31}$$

consistent with the expression before the handle slide.

- Independence on the choice of $f : \mathcal{M} \to BG$

  The prescription to write down the $\eta$-factors explicitly uses a specific choice of $f : \mathcal{M} \to BG$. Yet two maps $f : \mathcal{M} \to BG$ and $\tilde{f} : \mathcal{M} \to BG$ that are homotopic to each other should give rise to a topologically equivalent bundle $\mathcal{G}$. Because $f$ and $\tilde{f}$ are homotopic to each other, for a 2-handle with its associated 2-chain $[h]$, $f_*[h]$ and $\tilde{f}_*[h]$ should be related to each other by some 3-chain $v$, i.e., $f_*[h] = \tilde{f}_*[h] + \partial v$. Therefore, given the symmetry fractionalization class $\mathbf{w} \in \mathcal{H}^2(G, \mathcal{A})$, $\mathbf{w}(f_*[h]) = \mathbf{w}(\tilde{f}_*[h])$ and $\eta$-factors are indeed independent of the specific choice of $f : \mathcal{M} \to BG$.

- Cobordism invariance

  To prove that the partition function is a cobordism invariant, we also need to prove that the partition function $\mathcal{Z}(\mathcal{M},\mathcal{G})$ is invariant under the three surgery moves. Especially, the first surgery move, i.e., removing or adding an $S^4$, does not change the partition function because we can directly see from Eq. (65) that $\mathcal{Z}(S^4,\mathcal{G}) = 1$, no matter what $G$-bundle $\mathcal{G}$ we put on $S^4$. The third surgery move does not involve any 2-handles. The second surgery move, i.e., replacing an $S^1 \times D^3$ by $S^2 \times D^2$ and vice versa, involves a 2-handle. Now consider $S^2 \times \{pt\} \subset S^2 \times \partial(D^2) \cong S^2 \times S^1$. Such an $S^2$ lives on the boundary of $D^3$ before the surgery, hence the $G$-bundle on this $S^2$ can be thought of as trivial. Denoting the 2-chain associated to the 2-handle by $[h]$, we then see that $f_*[h]$ is accordingly also trivial and thus no $\eta$-factor is involved. This argument is in a similar spirit to the argument presented in Appendix C.4. Now we can carry over the proof of cobordism invariance in the Crane-Yetter model to prove cobordism invariance of $\mathcal{Z}(\mathcal{M},\mathcal{G})$ under the surgery moves.

- Invertibility

  For torsion elements in $\Omega_4^{SO}((BG)^{q-1})$, cobordism invariance and the fact that $\mathcal{Z}(S^4) = 1$ from Eq. (65) also dictate that the norm of the partition function $\mathcal{Z}(\mathcal{M},\mathcal{G})$ on such manifolds is 1. However, for a connected Lie group $G$, there can be many $\mathbb{Z}$ pieces in $\Omega_4^{SO}((BG)^{q-1})$ which do not correspond to $\mathbb{CP}^2$ with trivial $G$-bundle structure. We conjecture that partition functions of these manifolds according to the construction in Eq. (65) still have norm 1, but we do not have a direct proof (but see Appendix C.5 for an argument based on properties of a fully-extended TQFT).

# D   Identifying the manifold $\mathcal{M}$ from bordism

In this appendix, we say more about which $(3+1)D$ manifolds $\mathcal{M}$ concern us, given a symmetry group $G$ equipped with a $\mathbb{Z}_2$ grading $q : G \to \mathbb{Z}_2$ to denote anti-unitary elements as in Eq. (15).

First of all, the manifolds $\mathcal{M}$ should be the generating manifolds of $\Omega_4^{SO}((BG)^{q-1})$ [70,88], which we define in detail below by identifying its tangential structure.

Let $H$ be the tangential structure that concerns us, given the symmetry group $G$ together with a $\mathbb{Z}_2$ grading $q$ to denote anti-unitary symmetries. Then for any integer $n$ we have

$$
\begin{array}{ccccccccc}
1 & \longrightarrow & G/\mathbb{Z}_2 & \longrightarrow & H_n & \longrightarrow & O_n & \longrightarrow & 1 \\
& & \downarrow{\scriptstyle\cong} & & \downarrow & & \downarrow{\scriptstyle\det} & & \\
1 & \longrightarrow & G/\mathbb{Z}_2 & \longrightarrow & G & \xrightarrow{q} & \mathbb{Z}_2 & \longrightarrow & 1
\end{array}
\tag{D.1}
$$

where det denotes the determinant map. Here $H$ is a nontrivial extension of $O$ by $G/\mathbb{Z}_2$, and $H_n \to G$ is the pullback of the determinant map $\det : O_n \to \mathbb{Z}_2$ by the $\mathbb{Z}_2$ grading $q : G \to \mathbb{Z}_2$. In this paper we use $\Omega_4^{SO}((BG)^{q-1})$ to denote the bordism group with this tangential strcutrure $H$.

An informative example is when $G$ is $\mathbb{Z}_4^T$ with the generator of $\mathbb{Z}_4$ an anti-unitary symmetry. Then $q : \mathbb{Z}_4^T \to \mathbb{Z}_2$ is just the projection from $\mathbb{Z}_4$ to $\mathbb{Z}_2$. According to Eq. (D.1), we see that $H$ is a nontrivial extension of $O$ by $\mathbb{Z}_2$. But pay attention that $H$ is not the same as Pin$^+$ or Pin$^-$, because the extension for $H$ corresponds to $w_1^2$ in $\mathcal{H}^2(O_n, \mathbb{Z}_2)$, while the extension for Pin$^+$ or Pin$^-$ corresponds to $w_2$ or $w_2 + w_1^2$ in $\mathcal{H}^2(O_n, \mathbb{Z}_2)$, respectively. Accordingly, for $G = \mathbb{Z}_4^T$ the manifold $\mathcal{M}$ as the generator of the bordism group $\Omega_4^{SO}((BG)^{q-1})$ should have $w_1^2 = 0$. It is also straightforward to see that when $G$ is $\mathbb{Z}_2^T \times \mathbb{Z}_2^T$, $H$ is a trivial extension of $O$ by $\mathbb{Z}_2$. Given $\mathcal{H}^2(O_n, \mathbb{Z}_2) \cong \mathbb{Z}_2 \times \mathbb{Z}_2$, we have listed all tangential structures associated to the four extensions of $O_n$ by $\mathbb{Z}_2$.

From the right box in Eq. (D.1), we also see that $H$-tangential structure is equivalent to a $(BG, q)$-twisted orientation of $\mathcal{M}$, hence the notation $\Omega_4^{SO}((BG)^{q-1})$. Namely, to identify some $H$-structure on $\mathcal{M}$, we can first put a principal $G$ bundle $\mathcal{G}$ on $\mathcal{M}$. The map $q : G \to \mathbb{Z}_2$ induces an associated 1-dimensional line bundle on $\mathcal{M}$ that we denote by $\xi$, then we must have $w_1(\xi) = w_1(\mathcal{M})$ and we need to choose an orientation of $\xi \oplus T\mathcal{M}$, where $T\mathcal{M}$ is the tangent bundle of $\mathcal{M}$.

Secondly, it turns out that most elements in the group are "in-cohomology" in the following sense. When $G$ only contains unitary symmetry, $\Omega_4^{SO}(BG)$ contains a special $\mathbb{Z}$ piece, generated by $\mathbb{CP}^2$. When $G$ also contains anti-unitary symmetry, $\Omega_4^{SO}((BG)^{q-1})$ contains a special $\mathbb{Z}_2$ piece, also generated by $\mathbb{CP}^2$. The rest elements are (Pontraygin) dual to the image of the natural map from group cohomology to cobordism group, i.e.,

$$\mathcal{H}^4\left(BG, U(1)_q\right) \longrightarrow \Omega_{SO}^4\left((BG)^{q-1}\right), \tag{D.2}$$

where $q$ as subscript of $U(1)$ denotes the nontrivial $G$ action on $U(1)$ associated with $q$. Therefore, we call the $\mathbb{Z}$ piece or $\mathbb{Z}_2$ piece "beyond-cohomology", while the rest piece "in-cohomology" [8].

Therefore, as an easier step to identify a complete list of $(3+1)D$ manifolds needed for the calculation, first we calculate $\mathcal{H}^4(BG, U(1)_q)$ and identify a set of generators $\mathcal{O}_i$. For $G$ containing unitary symmetry only, we proceed by searching for some oriented manifold $\mathcal{M}_i$ together with a map $f_i : \mathcal{M}_i \to BG$ corresponding to some $G$-bundle for each $i$, such that $f^*(\mathcal{O}_i)$ is dual to the fundamental cycle $[\mathcal{M}_i] \in H_4(\mathcal{M}_i, \mathbb{Z})$.

For $G$ containing anti-unitary symmetry, we also need to search for some manifold $\mathcal{M}_i$ together with a map $f_i : \mathcal{M}_i \to BG$ corresponding to some $G$-bundle for each $i$, with the following two constraints:

1. The following diagram commute

$$\begin{array}{ccc} \mathcal{M}_i & \xrightarrow{f_i} & BG \\ & \searrow{\scriptstyle w} & \downarrow{\scriptstyle q} \\ & & B\mathbb{Z}_2 \end{array} \tag{D.3}$$

   where $w$ is the map corresponding to the orientation bundle. In particular, we allow $\mathcal{M}_i$ to be non-orientable.

2. $f^*(\mathcal{O}_i)$ is dual to the twisted fundamental cycle $[\mathcal{M}_i] \in H_4(\mathcal{M}_i, \mathbb{Z}_w)$ twisted by the orientation character $w$ [88].

We call such a manifold $\mathcal{M}_i$ (together with a $G$-bundle structure $\mathcal{G}_i$ on it) a *representative manifold* of $\mathcal{O}_i$. Moreover, we also need $\mathbb{CP}^2$ for the "beyond-cohomology" piece of the bordism group.

Finally, we refer the reader to Ref. [38] for an algorithm to get the cellulation of the representative manifolds given a finite symmetry group $G$.

# E  More information about handle decomposition of manifolds

In this appendix, we give more information about the handle decomposition of various manifolds that appear in the main text, i.e., those manifolds listed in Table 1. More information about them can be found in Refs. [80, 81].

## E.1  $\mathbb{CP}^2$

Let us start with $\mathbb{CP}^2$. The manifolds $\mathbb{CP}^n$ have a handle decomposition with $n+1$ handles. There is one handle of each even index from 0 to $2n$. Such a decomposition for $\mathbb{CP}^n$ can be constructed as follows. Recall that each point $p \in \mathbb{CP}^n$ has homogeneous coordinates $[z_0 : \cdots : z_n]$, $z_i \in \mathbb{C}$, which we can normalize so that $\max_i |z_i| = 1$. Let $\mathcal{D}$ be the closed unit disk in $\mathbb{C}$, which is homeomorphic to $D^2 = [-1, 1]^2$. Then $\mathbb{CP}^n$ can be covered by $n+1$ balls $\mathcal{D}^n$ through the following map

$$\psi_i : \mathcal{D}^n \to \mathbb{CP}^n, \quad i = 0, \ldots, n, \tag{E.1}$$

where

$$\psi_i(z_1, \ldots, z_n) = [z_1 : \cdots : z_i : 1 : z_{i+1} : \cdots : z_n]. \tag{E.2}$$

Let the image of $\mathcal{D}^n$ under the map $\psi_i$ be $B_{2i}$. Then $p \in B_{2i}$ if and only if $|z_i| = 1$, and $p \in \text{int } B_{2i}$ if and only if $|z_j| < 1$ for all $j \neq i$. It follows immediately that the balls $B_{2i}$ cover $\mathbb{CP}^n$, and that they only intersect along their boundaries. Moreover, $B_{2k}$ intersects $\cup_{i<k} B_{2i}$ precisely on $\psi_k(\partial(\mathcal{D}^k) \times \mathcal{D}^{n-k})$. Therefore, we can interpret $B_{2k}$ as a $2k$-handle attached to $\cup_{i<k} B_{2i}$, exhibiting the required handle decomposition.

Now specialize to $\mathbb{CP}^2$. To draw the Kirby diagram as in Eq. (45) we just need to understand the appearance of the topological twist reflecting the self-intersection number $+1$. We can see the fact from the intersection form of $\mathbb{CP}^2$, which is $[+1]$. We can also directly determine the attaching region of the 2-handle. A point $p$ in $B_0 \cap B_2$ can be written in two ways: $p = \psi_0(w_1, w_2) = [1 : w_1 : w_2]$ and $p = \psi_1(z_1, z_2) = [z_1 : 1 : z_2]$. Comparing homogeneous coordinates, we find that $w_1 = z_1^{-1}$ and $w_2 = z_1^{-1} z_2$, so $\varphi(z_1, z_2) = (z_1^{-1}, z_1^{-1} z_2)$ defines the attaching map $\varphi : \partial \mathcal{D} \times \mathcal{D} \to \partial \mathcal{D} \times \mathcal{D} \subset \partial B_0$. Parametrize $z_1 = e^{2\pi i t}$, $0 \leq t \leq 1$, as we travel once around $\partial \mathcal{D}$, $t$ goes from 0 to 1 while the identification of the fibers ($z_2 \mapsto e^{-2\pi i t} z_2$) rotates once, realizing a generator of $\pi_1(O(2)) \cong \mathbb{Z}$. As a result, there is a $+1$ framing of the 2-handle, reflecting the self-intersection number $+1$.

## E.2  $\mathbb{RP}^4$

The handle decomposition of manifolds $\mathbb{RP}^n$ is very similar to $\mathbb{CP}^n$. The manifolds $\mathbb{RP}^n$ have a handle decomposition with $n+1$ handles. There is one handle of each index from 0 to $n$. A decomposition for $\mathbb{RP}^n$ can be constructed from the construction of $\mathbb{CP}^n$ simply by changing $\mathbb{C}$ to $\mathbb{R}$ and $\mathcal{D}$ to $D$. More specifically, recall that each point $p \in \mathbb{RP}^n$ has homogeneous coordinates $[x_0 : \cdots : x_n]$, $x_i \in \mathbb{R}$, which we can normalize so that $\max_i |x_i| = 1$. Then $\mathbb{RP}^n$ can be covered by $n+1$ balls $D^n$ through the following map

$$\psi_i : D^n \to \mathbb{RP}^n, \quad i = 0, \ldots, n, \tag{E.3}$$

where

$$\psi_i(x_1, \ldots, x_n) = [x_1 : \cdots : x_i : 1 : x_{i+1} : \cdots : x_n]. \tag{E.4}$$

Let the image of $D^n$ under the map $\psi_i$ be $B_i$. Then we see that $B_i$ as an $i$-handle is the required handle decomposition.

Now specialize to $\mathbb{RP}^4$. To draw the Kirby diagram as in Fig. 7 we need to determine the self-intersection number of the line reflecting the 2-handle. We can see this from the mod-2 intersection form of $\mathbb{RP}^4$, which is $[+1]$. We can also directly determine the attaching region of the 2-handle. A point $p$ in $\partial(D^2) \times D^2 \subset \partial B_2$ can be written as: $p = \psi_2(x_1, x_2, x_3, x_4) = [x_1 : x_2 : 1 : x_3 : x_4]$ and either $|x_1| = 1$, $p \in \partial B_0$ or $|x_2| = 1$, $p \in \partial B_1$. Comparing the fibre $(x_3, x_4)$, we see that when we travel along the boundary $\partial(D^2)$, $(x_3, x_4)$ changes sign twice after the identification. As a result, there is a +1 framing of the 2-handle as well.

Notice that when attaching a 2-handle to a 1-handle, the framing may not be a well-defined integer because some isotopy involving the 1-handle may change the framing. But it is a well-defined integer mod-2 [81].

### E.3 $\quad \mathbb{RP}^3 \times S^1$

The handle decomposition of a product manifold $\mathcal{A} \times \mathcal{B}$ is easy to achieve if we know the handle decomposition of $\mathcal{A}$ and $\mathcal{B}$ individually. In this way, we can get the handle decomposition of $\mathbb{RP}^3 \times S^1$ and $\mathbb{RP}^2 \times \mathbb{RP}^2$ that concerns us in this paper.

For $\mathbb{RP}^3 \times S^1$, the handle decomposition of $\mathbb{RP}^3$ has been worked out in Appendix E.2, which consists of 1 0-handle, 1 1-handle, 1 2-handle and 1 3-handle, and the handle decomposition of $S^1$ can just consist of 1 0-handle $D^1$ and 1 1-handle $D^1$ attached along the two end points. Therefore, the handle decomposition of $\mathbb{RP}^3 \times S^1$ consists of 1 0-handle, 2 1-handle, 2 2-handle, 2 3-handle and 1 4-handle, and the Kirby diagram is drawn in Fig. 7. The blue 1-handle comes from the product of 0-handle of $S^1$ and 1-handle of $\mathbb{RP}^3$, and the darkblue 1-handle comes from the product of 0-handle of $\mathbb{RP}^3$ and 1-handle of $S^1$. The orange 2-handle comes from the product of 0-handle of $S^1$ and 2-handle of $\mathbb{RP}^3$, and we can determine its framing either by mod-2 intersection form or from the Heegard diagram of $\mathbb{RP}^3$. Finally, the red 2-handle comes from the product of 1-handle of $S^1$ and 1-handle of $\mathbb{RP}^1$. The explicit ways of drawing the 2-handles on the Kirby diagram can be worked out by following closely the construction of the handle decomposition.

### E.4 $\quad \mathbb{RP}^2 \times \mathbb{RP}^2$

The handle decomposition of $\mathbb{RP}^2 \times \mathbb{RP}^2$ can be achieved in a similar manner to $\mathbb{RP}^3 \times S^1$. Specifically, the handle decomposition of $\mathbb{RP}^2$ has been worked out in Appendix E.2, which consists of 1 0-handle, 1 1-handle and 1 2-handle. Therefore, the handle decomposition of $\mathbb{RP}^2 \times \mathbb{RP}^2$ consists of 1 0-handle, 2 1-handle, 3 2-handle, 2 1-handle and 1 4-handle, and the Kirby diagram is drawn in Fig. 10. The blue 1-handle and the red 2-handle come from one $\mathbb{RP}^2$ piece while the darkblue 1-handle and the orange 2-handle come from the other $\mathbb{RP}^2$ piece. There is another sanddune 2-handle, coming from the product of 2 1-handles of two $\mathbb{RP}^2$ pieces. The explicit ways of drawing the 2-handles on the Kirby diagram can be worked out by following closely the construction of the handle decomposition.

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
