# Peer review of "Anomaly of $(2+1)$-Dimensional Symmetry-Enriched Topological Order from $(3+1)$-Dimensional Topological Quantum Field Theory"

_SciPost Physics, doi:SciPost Phys. 15, 004 (2023)_

## Round 1 · Referee Report · Anonymous (Referee 1) · 2023-1-15

Strengths
1- The paper gives a pedagogical introduction to anomaly indicators and the recipe for writing one down in bosonic theories.
2-The authors compute the relevant bordism groups associated to the anomaly and also give the manifold generators.
3-The authors were able to come up with intricate formulas for the anomaly indicators of different symmetries by studying the handle body decomposition of the manifold generators. They checked that the formula they give is gauge invariant, and also vertex basis transformations.
Weaknesses
Report
How applicable are anomaly indicators for other continuous groups, or perhaps nonsimply connected Lie groups?
If I have a TQFT of the form U(N)_{N,2N} with T^2 = (-1)^F symmetry, can I use anomaly indicators to find the anomaly that this theory takes? It should be \pm 2 mod 16, as was shown in the work here: https://arxiv.org/pdf/1610.07010.pdf
It was known how to detect the anomalies for theories with T^2=(-1)^F, and then the anomaly indicator was reproduced on the generating manifold of Pin^+ bordism in degree 4, i.e. RP^4. Since this is also a spin theory, can the methods used here to obtain the anomaly indicator be used to find the anomaly indicators of other fermionic theories? By this, I mean a theory the couples to spin structure, and has a "fermionic symmetry" of the form of an extension of some bosonic symmetry by Z_2^F. If not, then what are the obstructions to writing an anomaly indicator for fermionic theories in general?
Requested changes
1- In appendix D, the relevant bordism group should be \Omega^{\SO}((BG)^{\sigma-1}) as the symmetry structure is a \sigma-twisted orientation of BG. The map q: Z_4-->Z_2 induces a line bundle \sigma-->BZ_4 by pulling the tautological line bundle back from BO_1. If P --> X is a principal Z_4-bundle, we let \sigma_R --> X be the associated line bundle; then w1(\sigma_P ) = x(P).
Author: Weicheng Ye on 2023-03-03 [id 3434]
(in reply to Report 1 on 2023-01-15)
We thank the referee for thinking highly of our paper and the recommendation of publication. We especially thank the referee for the detailed explanation of the bordism group. Below we address the referee's comments.
-
In the presence of non-simply connected Lie groups, following our logic we outline a similar construction at the beginning of Section V. However, we find that such construction more cumbersome and is manifestly dependent on some unnecessary details, i.e., the explicit choice of $f:~\mathcal{M}\rightarrow BG$ instead of just its homotopy class $[f]$, which we believe is an artifact of the formalism and can be avoided in principle.
Nevertheless, here we want to demonstrate that our construction does give a construction of the partition function in many cases as follows. We can decompose a non-simply connected Lie group $G$ using the following short exact sequence, \begin{equation} 1\rightarrow G_c\rightarrow G\rightarrow G/G_c\rightarrow 1 \end{equation} where $G_c$ is the subgroup which contains elements that are smoothly connected to the identity element of $G$, and $G/G_c$ is the finite part of $G$. When $G/G_c$ has a section in $G$, i.e. when there is an injection $i:~G/G_c\rightarrow G$, we can show that the partition function of any $G$ bundle $\mathcal{G}$ of $\mathcal{M}$ can be constructed by demanding that $\eta$-factors relevant for some 2-handle can be achieved by multiplying the $\eta$-factors we got from $G_c$ as in Sec. V and $\eta$-factors we got from $G/G_c$ as in Sec. IV.
However, in the most general situation, we cannot write down or prove a more operational procedure to calculate the partition function like what we discussed in the main text. That said, we do believe such a procedure is possible, and we defer it to future work.
-
The anomaly indicator of fermionic topological order for time-reversal symmetry with $T^2=(-1)^F$ has been given in https://arxiv.org/abs/1610.04624 (Eq. (10) thereof) and https://arxiv.org/abs/2104.14567 (Eq. (4) thereof), and it certainly can be used to get the anomaly of $U(N)_{N,2N}$ by explicitly plugging the anyon data into the anomaly indicator. The detailed calculation for $N=1$, i.e., the semion-fermion theory, is carried out in the paragraph surrounding Eq. (13) of https://arxiv.org/abs/1610.04624. And we believe that the calculation for other $N$ parallels the calculation of $N=1$.
-
We believe that our construction utilizing handle-body decomposition instead of cell decomposition or triangulation can be straightforwardly generalized to fermionic topological order and constructing anomaly indicators for fermionic symmetry. To address this point, we expand our discussion point 3 in the Discussion section of our paper and we repeat the statement here. According to https://arxiv.org/abs/1505.05856 and https://arxiv.org/abs/2104.14567, the partition function of a (3+1)$D$ fermionic SPT can be obtained by combining a so-called ``bosonic shadow" part, denoted by $Z_b$, and another part that originates from the fermionic nature of the system, denoted by $z_c$, i.e., \begin{equation} \mathcal{Z}(\mathcal{M}, \mathcal{G}, \xi) = \frac{1}{\sqrt{|\mathcal{H}^2(\mathcal{M}, \mathbb{Z}_2)|}} \sum Z_b(\mathcal{M}, \mathcal{G}, f)z_c(\mathcal{M}, f, \xi) \end{equation} Our framework allows us to obtain the bosonic shadow $Z_b$ much easier, given a handle decomposition of the manifold. Here the sum is over $[f]\in H_1(\mathcal{M},\mathbb{Z}_2)$ which suggests that we need to insert an extra fermion loop into the cycle $f$. Moreover, the prescription to obtain $z_c$ given a handle decomposition is also relatively straightforward. So our framework can be generalized to fermionic systems, and we believe the calculation will be greatly simplified as well. We defer the full details to future work.
Author: Weicheng Ye on 2023-03-03 [id 3433]
(in reply to Report 2 on 2023-02-01)We thank the referee for carefully reading our manuscript, for the recommendation of publication and for the constructive comments. Below we address the referee's comments.
This confusion is due to the difference in the definitions of anomaly indicators between us and the referee. In our definition (stated informally in the Introduction and more formally at the end of Sec. III A), the anomaly indicators are numbers which serve as coefficients of an element in the cohomology or cobordism group, defined under a certain basis. On the other hand, the referee's definition of the anomaly indicators seems to be the set of basis of the cohomology or cobordism group. These two definitions are of course equivalent, but we would like to continue using ours because in our definition the anomaly indicators are directly expressed in terms of data characterizing the symmetry-enriched topological order, without explicitly referring to the cohomology or cobordism group.
In the context of Eq. (51), we can write $\mathcal{I}_0=\exp(\pi i \tilde{I}_0)$ and $\mathcal{I}_1=\exp(\pi i \tilde{I}_1)$, then the partition function corresponding to a certain topological order with $\mathbb{Z}_2^T$ symmetry should be \begin{equation} \mathcal{O} = \exp\left(\pi i \left(\tilde{I}_0 \int w_2\cup w_2 + \tilde{I}_1 \int w_1^4 \right)\right) \end{equation} For example, when $\mathcal{I}_0=1$ and $\mathcal{I}_1=-1$, the partition function above becomes $\exp\left(\pi i\int w_1^4\right)$; while when $\mathcal{I}_0=\mathcal{I}_1=-1$, the partition function above becomes $\exp\left(\pi i\int w_2\cup w_2 + w_1^4\right)$.
In Eq. (51) we omit the integral sign (to ease the notation) as well as regroup $\exp(\pi i \tilde{I}_0)$ and $\exp(\pi i \tilde{I}_1)$ into $\mathcal{I}_0$ and $\mathcal{I}_1$.
There is indeed an additional factor of $d_a$ coming from the product over 2-handles, but it is canceled by another factor of $d_a$ coming from the contribution of 1-handles, as suggested by Eq. (44).
Similar to point 1 above.
We do not neglect contributions coming from the curvature of the $G$-bundle. In fact, the procedure outlined at the beginning of Sec. V is specifically dedicated to dealing with the curvature of the $G$-bundle. In various places of our calculation, e.g. Eq. (74), the Chern class of the bundle appears explicitly.
The bordism group and the cobordism group are the same whenever $N\geqslant 5$, for symmetry groups $SO(N)$, $O(N)^T$ and $SO(N)\times \mathbb{Z}_2^T$ that we consider in our text.

---

## Round 1 · Referee Report · Anonymous (Referee 2) · 2023-2-1

Strengths
1) The work studies the interesting problem of diagnosing the ’t Hooft anomalies of G-symmetric 2+1d topological orders by constructing a 3+1d TQFT built from the data of the 2+1d topological order along with the G-action on it.
2) Several interesting and illustrative computations of the partition functions of the 3+1d anomaly TQFT have been described.
3) While various aspects of the work have appeared previously in the literature, in particular in Ref 34 and 39 cited in the paper, the paper under review makes the technical contribution of using handle-body decomposition of the 4-manifold in computing the partition function thereof.
4) Additionally the authors discuss anomalies of Lie group global symmetries which haven’t been explored much in the literature particularly, using the methods of the present work. The authors also show that certain anomalies of Lie groups can’t be saturated by any topological order.
Weaknesses
Report
Requested changes
-
There seems to be a typo in Equation (51). The anomaly action of $\mathbb Z_2^T$ global symmetry should be $I_0= (-1)^{\int w_2 \cup w_2}$ and $I_1= (-1)^{\int w_1^4}$. It is unclear, what is meant by $I_0^{w_2^2}$ and $ I_1^{t^4}$.
-
In Equation (50), should there be an additional factor of $d_a$ as in the expression for $I_0$, coming from the product over 2-handles in Equation (44)?
-
Similar to point 1 above, in (54), should the anomaly action for 2+1d topological orders with $\mathbb Z_2 \times \mathbb Z_2$ global symmetry be simply $I_1 \times I_2$. For instance, $I_1= (-1)^{ \int A_1^3 \cup A_2} = (-1)^{c_1^3 c_2}$, where $(A_1,A_2)$ is the $\mathbb Z_2\times \mathbb Z_2$ background gauge field etc.
-
(66) is obtained by generalising (44) to connected Lie groups. In general, as opposed to the case of finite groups, network of symmetry defects does not fully capture a G bundle for a Lie group G. Can the authors clarify how one can neglect possible contributions to the anomaly coming from the curvature of the G-bundle. For instance, from the Chern class of the bundle.
-
Similar to (70), what is the reason there aren’t additional SO(N) line bundles for N >= 6,8, etc.?

---

## Round 2 · Referee Report · Anonymous (Referee 2) · 2023-3-6

Report

The authors have satisfactorily addressed all the comments in my previous report. I recommend this paper for publication on SciPost.

---

## Round 2 · Author Response

We thank the editor for dealing with our draft, and all referees for their constructive comments and suggestions. We respond to the referees' reports in the comments. We list changes we made in our new manuscript below.

---

## Round 2 · List of Changes

1. As suggested by Referee 2, we change the notation of the bordism group when $G$ contains anti-unitary symmetry from $\Omega_4^{O}(BG, q)$ to $\Omega_4^{O}((BG)^{q-1})$, to emphasize the choice of choosing a $q$-twisted orientation of $\mc{M}$.

  2. In Sec. IIIA and Sec. IVB, we emphasize that the anomaly indicators are numbers which serve as coefficients in front of a certain basis of the relevant cohomology or cobordism group.

  3. We expand the point 3 in the discussion section to explain in more detail how our formalism can be generalized to fermionic systems and obtain partition functions and anomaly indicators thereof.

---

## Editorial Decision

published